# PARAMETER-EFFICIENT FINE-TUNING OF STATE SPACE MODELS

## ABSTRACT

Deep State Space Models (SSMs), such as Mamba (Gu & Dao, 2024), have emerged as powerful tools for language modeling, offering high performance with efficient inference and linear scaling in sequence length. However, the application of parameter-efficient fine-tuning (PEFT) methods to SSM-based models remains largely unexplored. This paper aims to systematically study two key questions: (i) How do existing PEFT methods perform on SSM-based models? (ii) Which modules are most effective for fine-tuning? We conduct an empirical benchmark of four basic PEFT methods on SSM-based models. Our findings reveal that prompt-based methods (e.g., prefix-tuning) are no longer effective, an empirical result further supported by theoretical analysis. In contrast, LoRA remains effective for SSM-based models. We further investigate the optimal application of LoRA within these models, demonstrating both theoretically and experimentally that applying LoRA to linear projection matrices without modifying SSM modules yields the best results, as LoRA is not effective at tuning SSM modules. To further improve performance, we introduce LoRA with Selective Dimension tuning (SDLoRA), which selectively updates certain channels and states on SSM modules while applying LoRA to linear projection matrices. Extensive experimental results show that this approach outperforms standard LoRA.

## 1 INTRODUCTION

Over the past two years, Large Language Models (LLMs) such as ChatGPT (Achiam et al., 2023; Brown et al., 2020) have achieved groundbreaking performance and are now widely used in daily life. Many models use the Transformer architecture (Vaswani et al., 2017), with its attention mechanism essential in predicting subsequent tokens based on context. Each token computes attention scores with every preceding one, selectively focusing only on the most relevant during processing. This, however, creates quadratic time complexity, posing challenges when dealing with long sequences. In response, various alternative architectures like Linear Attention (Katharopoulos et al., 2020), RWKV (Peng et al., 2023), RetNet (Sun et al., 2023), and Mamba (Gu & Dao, 2024) have been developed to operate with subquadratic time complexity.

As the most popular subquadratic-time architecture currently serving as an alternative to Transformers, SSMs (Gu et al., 2021; 2022b;a; Gu & Dao, 2024) achieve efficient training and inference. SSMs are closely related to linear RNNs, which maintain a hidden state to encapsulate the information of previous tokens. When a new input token is introduced, the prediction of the next token involves only operations on this hidden state and the new token, which enhances inference efficiency. To overcome the limitation of RNNs, which cannot be trained in parallel, S4 (Gu et al., 2022b;a) leverages its linearity, enabling it to adopt a convolutional form during training, facilitating parallel computation. Consequently, SSMs are highly efficient and have demonstrated success in numerous long-sequence tasks (Gu et al., 2022b;a). Recently, a new series of SSM models, Mamba (Mamba-I (Gu & Dao, 2024) and Mamba-II (Dao & Gu, 2024)), have achieved Transformer-level performance in language modeling. In the main paper, we primarily focus on the deep S4 model and Mamba-I, while deferring experiments involving Mamba-II and the hybrid model, Jamba (Lieber et al., 2024), to the appendix. Unless otherwise specified, "Mamba" refers to Mamba-I for simplicity of notation. The deep S4 model, serving as the foundational architecture, readily extends its properties to other variants, while Mamba has emerged as one of the most popular SSM-based models.

Consequently, we expect fine-tuning these pretrained SSMs for downstream tasks will become a crucial problem in the near future. While fine-tuning the entire model is expensive and inefficient,

numerous Parameter-Efficient Fine-Tuning (PEFT) methods (Houlsby et al., 2019; Hu et al., 2021; He et al., 2021; Li & Liang, 2021; Lester et al., 2021; Zaken et al., 2022; Liu et al., 2021; 2022) have been developed for efficient adaptation under resource constraints. Notably, most popular PEFT methods fall into two categories: (i) prompt-based tuning, which involves modifying the input sequence (Lester et al., 2021) or tuning the sequence at each layer (Li & Liang, 2021); and (ii) parameter-based tuning, which directly updates the model parameters, such as LoRA (Hu et al., 2021), which modifies the weight matrices, and BitFit (Zaken et al., 2022), which updates only the bias terms.

Despite the success that existing PEFT methods have achieved in adapting Transformer-based models, their efficacy in adapting SSM-based models remains largely unexplored, leaving many interesting questions open. For instance, are existing popular PEFT methods still effective for SSM-based models? If they are applicable, what is the optimal way to apply these methods to SSM-based models, and which parameters should be updated? If not, can we develop variants specifically tailored for SSMs that perform better? To answer these questions, to the best of our knowledge, we conduct the first comprehensive study of PEFT on SSM-based models, both theoretically and empirically.

To the best of our knowledge, we are the first to benchmark existing PEFT methods on SSM-based models. Through extensive experiments, we demonstrate that **(Finding 1)** prompt-based PEFT methods are no longer effective for SSM-based models, and **(Finding 2)** LoRA remains effective on SSM-based models. Meanwhile, the two major components of SSM-based models are the SSM module, which functions analogously to attention in Transformers, and linear projection matrices, which are similar to feed-forward layers. We next investigate which part of the model is more effective for applying PEFT. We empirically find that **(Finding 3)** applying LoRA to linear projection matrices without modifying the SSM module is already effective, while the most effective linear projection matrices differ depending on the dataset. Notably, Findings 1 and 3 are supported by our theoretical analysis. While LoRA is not effective for tuning SSM modules, theoretically, tuning additional SSM modules increases expressivity. Finally, we analyze the architecture of SSM-based models using the theoretical framework of Giannou et al. (2023) and Zeng & Lee (2024). We show that, in addition to applying LoRA to linear projection matrices, *S*electively updating the channel and state *D*imensions of SSM modules further enhances performance. We dub this method as SDLoRA, the first PEFT method tailored for SSM-based models. Through extensive experiments, we observe that **(Finding 4)** SDLoRA outperforms LoRA alone in fine-tuning SSM-based models.

## 2 RELATED WORKS

**State Space Models (SSMs).** Linear State-Space Layers (LSSL) represent one of the earliest SSM layers utilized in deep learning, functioning as continuous-time, recurrent, and convolutional models (Gu et al., 2021). LSSL employs HiPPO theory (Gu et al., 2020) to initialize the state matrix $A$, enabling the capture of long dependencies. However, LSSL is computationally expensive, limiting its practical application. Gu et al. (2022b) introduced Structured State Space Models (S4), which optimize computation efficiency by employing a structured state matrix $A$. Gupta et al. (2022) proposed DSS, which simplifies the model by using a diagonal matrix for $A$ and empirically demonstrated that it suffices to achieve performance comparable to S4. Further, Gu et al. (2022a) provided a theoretical explanation for the effectiveness of the diagonal state matrix $A$ in DSS and introduced S4D, which offers various initialization methods for $A$. Subsequently, the diagonal structure of the state matrix $A$ has been adopted in follow-up methods (Gu & Dao, 2024). Despite differences in optimization algorithms, we refer to S4 and its close variants, including DSS and S4D, collectively as S4. This terminology encompasses models that maintain the standard discrete-time SSM form with a diagonal state matrix.

Despite of the remarkable performance of SSMs on certain tasks of sequence modeling, SSMs still showed worse performance than Transformers on language modeling. Fu et al. (2022) transitioned from synthetic language modeling tasks to real language modeling tasks with SSMs. They proposed H3, which is inspired by Linear Attention (Katharopoulos et al., 2020), introducing both diagonal SSM and shift SSM. Recently, Mamba (Gu & Dao, 2024; Dao & Gu, 2024) escaped from linear time invariance (LTI) modeling by introducing input-dependent terms and achieved better performance than Transformer on language modeling. Furthermore, several hybrid models (Lieber et al., 2024; Park et al., 2024) tried to exploit the advantages of both SSMs and Transformers.

**Parameter-Efficient Fine-Tuning (PEFT).** Due to the increase in model size, PEFT methods have gained increasing popularity as they achieve good performance while being much more efficient compared to full model updating (Houlsby et al., 2019; Hu et al., 2021; He et al., 2021; Li & Liang, 2021; Lester et al., 2021; Zaken et al., 2022; Liu et al., 2021; 2022). Most of the existing popular PEFT methods fall into two categories: (i) prompt-based methods (Li & Liang, 2021; Lester et al., 2021; Liu et al., 2021; 2022), and (ii) parameter tuning methods (Donahue et al., 2014; Yosinski et al., 2014; Hu et al., 2021; Zaken et al., 2022). Common prompt-based methods include prompt tuning (Lester et al., 2021), and prefix-tuning (Li & Liang, 2021). Prompt tuning prepends a sequence of learnable virtual tokens, which are continuous vectors. Prefix-tuning further expands on Prompt tuning by prepending tokens across the model's depth, making it more powerful. Therefore, our analysis of prompt-based methods' limitations will focus on prefix-tuning, with the findings also applicable to the other prompt-based methods. Conversely, parameter tuning methods, which originated from traditional transfer learning practices, typically involve freezing the initial layers and only tuning the last few layers (Donahue et al., 2014; Yosinski et al., 2014). In recent years, more effective and innovative parameter tuning approaches have emerged (Hu et al., 2021; Zaken et al., 2022). The widely used Low-Rank Adaptation (LoRA) updates a subset of parameters (e.g., attention layers of a Transformer) in a low-rank manner. BitFit (Zaken et al., 2022), focuses on tuning only the bias terms of a pretrained model. In Sec. A, we provide a more detailed description of these baseline methods.

Numerous efforts have been made to theoretically understand existing PEFT methods. For prompt-based methods, Wang et al. (2023b), Petrov et al. (2024), and Oymak et al. (2023) have theoretically analyzed the effectiveness and limitations of prompt tuning and prefix-tuning for Transformer-based models. For LoRA, Zeng & Lee (2024) explored its expressive power by demonstrating that even a randomly initialized model can be adapted to match any smaller target model using LoRA. Some of our theoretical analysis draws upon the framework established by Zeng & Lee (2024). Jang et al. (2024) conducted a theoretical exploration of LoRA within the neural tangent kernel (NTK) regime.

## 3 PRELIMINARIES OF STATE SPACE MODELS

**Scalar-input Scalar-output SSM.** The initial SSM is derived from a specific continuous system that maps a one-dimensional function or signal $x(t) \in \mathbb{R}$ to $y(t) \in \mathbb{R}$ via an $H$-dimensional latent state $\boldsymbol{h}(t) \in \mathbb{R}^H$, as described in (1). In (1), input transition vector $\boldsymbol{B} \in \mathbb{R}^{H \times 1}$ indicates the input's impact on the state of the system, state matrix $\boldsymbol{A} \in \mathbb{R}^{H \times H}$ characterizes the system's internal state dynamics, and the output mapping vector $\boldsymbol{C} \in \mathbb{R}^{1 \times H}$ relates the state to the output $y(t)$.[1]

$$\begin{aligned} \boldsymbol{h}'(t) &= \boldsymbol{A}\boldsymbol{h}(t) + \boldsymbol{B}x(t) \\ y(t) &= \boldsymbol{C}\boldsymbol{h}(t) \end{aligned} \quad (1) \qquad \begin{aligned} \boldsymbol{h}_t &= \overline{\boldsymbol{A}}\boldsymbol{h}_{t-1} + \overline{\boldsymbol{B}}x_t, \\ y_t &= \boldsymbol{C}\boldsymbol{h}_t \end{aligned} \quad (2) \qquad \begin{aligned} \overline{\boldsymbol{K}} &= (\boldsymbol{C}\overline{\boldsymbol{B}}, \boldsymbol{C}\overline{\boldsymbol{A}}\,\overline{\boldsymbol{B}}, \dots, \boldsymbol{C}\overline{\boldsymbol{A}}^{t-1}\overline{\boldsymbol{B}}), \\ (y_1, \dots, y_t) &= (x_1, \dots, x_t) * \overline{\boldsymbol{K}} \end{aligned} \quad (3)$$

To adapt SSMs for deep learning, the continuous parameters $(\boldsymbol{A}, \boldsymbol{B})$ are transformed into discrete counterparts $(\overline{\boldsymbol{A}}, \overline{\boldsymbol{B}})$ using a learnable step size $\Delta \in \mathbb{R}$. An example of a discretization rule is the zero-order hold, which defines $\overline{\boldsymbol{A}} = \exp(\Delta\boldsymbol{A}), \overline{\boldsymbol{B}} = (\Delta\boldsymbol{A})^{-1}(\exp(\Delta\boldsymbol{A}) - \boldsymbol{I}) \cdot \Delta\boldsymbol{B}$.

The discrete-time SSM is formulated as (2). For efficient and parallelizable training, the output $\boldsymbol{y}$ of a length-$N$ input $\boldsymbol{x}$ in the discrete-time SSM can be computed with a long convolution, as detailed in (3). This convolution operation can be efficiently computed in the frequency domain with FFT.

**Vector-input Vector-output SSM.** Many deep learning tasks, such as language modeling, often use multi-channel inputs. When the input and output are vectors, denoted as $\boldsymbol{x}, \boldsymbol{y} \in \mathbb{R}^D$, separate SSMs are used for each of the $D$ input channels. As such, a superscript $(d)$ is introduced to indicate parameters specific to each channel when necessary. This notation may be omitted for simplicity.

**Structured State Space Sequence Model (S4).** S4 introduced by Gu et al. (2022b) represents one of the earliest applications of SSMs in deep learning. It features a **diagonal** structure for the state matrix $\boldsymbol{A}$, a design theoretically validated by Gu et al. (2022b) and practically implemented through its subsequent variants, DSS (Gupta et al., 2022) and S4D (Gu et al., 2022a).

---

[1]Note that $\boldsymbol{B}, \boldsymbol{C}$ are vectors. We use bold capital letters to remain consistent with existing works (Gu et al., 2022b; Gu & Dao, 2024).

**Deep S4 Layer.** Since S4 lacks non-linearity and operates with independent channels, a position-wise linear layer and a non-linear activation function are integrated into the deep S4 layer, facilitating information mixing across channels and introducing non-linearity. Furthermore, a residual connection from the input to the output of S4 is introduced. Let $\otimes$ represent the element-wise product, and $S4(\cdot)$ denote the S4 mechanism, where the output of each channel is computed according to (3) using its own convolutional kernel $\overline{\boldsymbol{K}}^{(d)}$. While the subtle details such as the activation functions may vary slightly from the previous studies (Gu et al., 2022b;a), for the theoretical analysis in this paper, we define the deep S4 layer as below. The output of a deep S4 layer is then formulated as:

$$\boldsymbol{y}_t = \text{ReLU}(\boldsymbol{W} \cdot S4_t(\boldsymbol{x}_1, \ldots, \boldsymbol{x}_t) + \boldsymbol{\beta} + \boldsymbol{u} \otimes \boldsymbol{x}_t), \tag{4}$$

where $\boldsymbol{W} \in \mathbb{R}^{D \times D}$ and $\boldsymbol{\beta} \in \mathbb{R}^D$ represent the linear projection matrix and bias, respectively, and $\boldsymbol{u} \in \mathbb{R}^D$ is the coefficient of the residual connection. Note that in a deep S4 layer, the trainable parameters are SSM parameters $(\boldsymbol{A}^{(d)}, \boldsymbol{B}^{(d)}, \boldsymbol{C}^{(d)}, \Delta^{(d)})$ across $D$ channels with $\boldsymbol{A}^{(d)}$ being diagonal and the parameters $(\boldsymbol{W}, \boldsymbol{\beta})$ for the linear layer and $\boldsymbol{u}$ for the residual connection.

**Selective State Space Models (S6).** A key property of all SSMs mentioned above is linear time invariance (LTI), where model dynamics remain constant over time. However, LTI models face significant limitations: their constant dynamics fail to selectively extract relevant information from the context or influence the hidden state in an input-dependent manner. The S6 model, proposed by Gu & Dao (2024), addresses these limitations by making its parameters input-dependent.

In particular, at each time step $t$, given the input $\boldsymbol{x}_t \in \mathbb{R}^D$, they introduce input-dependency to step size $\boldsymbol{\Delta}_t = (\Delta_t^{(1)}, \ldots, \Delta_t^{(D)})^\top \in \mathbb{R}^D$, input transition vectors $\boldsymbol{B}_t \in \mathbb{R}^{H \times 1}$ and the output mapping vectors $\boldsymbol{C}_t \in \mathbb{R}^{1 \times H}$ via linear projection:

$$\boldsymbol{\Delta}_t = \text{softplus}(\boldsymbol{W}_{\boldsymbol{\Delta}} \boldsymbol{x}_t + \boldsymbol{\beta}_{\boldsymbol{\Delta}}), \quad \boldsymbol{B}_t = \boldsymbol{W}_{\boldsymbol{B}} \boldsymbol{x}_t, \quad \boldsymbol{C}_t = \boldsymbol{W}_{\boldsymbol{C}} \boldsymbol{x}_t,$$

whereas the diagonal state matrices $\boldsymbol{A}^{(1)}, \ldots, \boldsymbol{A}^{(D)}$ remain input-independent. Note that $\boldsymbol{W}_{\boldsymbol{\Delta}} \in \mathbb{R}^{D \times D}$ is implemented via a rank-$r$ low-rank parameterization, denoted by $\boldsymbol{W}_{\boldsymbol{\Delta}} = \boldsymbol{W}_{\boldsymbol{\Delta}, \uparrow} \boldsymbol{W}_{\boldsymbol{\Delta}, \downarrow}$, where $\boldsymbol{W}_{\boldsymbol{\Delta}, \uparrow} \in \mathbb{R}^{D \times r}$ and $\boldsymbol{W}_{\boldsymbol{\Delta}, \downarrow} \in \mathbb{R}^{r \times D}$, which is a common method for reducing compute overheads (Wang et al., 2021; 2023a). To summarize, the trainable parameters in S6 include state matrices $\boldsymbol{A}^{(d)}$ across $D$ channels, parameters $\boldsymbol{W}_{\boldsymbol{\Delta}, \uparrow}, \boldsymbol{W}_{\boldsymbol{\Delta}, \downarrow}$ and $\boldsymbol{\beta}_{\boldsymbol{\Delta}}$ for computing $\boldsymbol{\Delta}_t$, and weight matrices $\boldsymbol{W}_{\boldsymbol{B}}, \boldsymbol{W}_{\boldsymbol{C}} \in \mathbb{R}^{H \times D}$ for computing $\boldsymbol{B}_t, \boldsymbol{C}_t$. The state matrices and the input transition vectors of S6 are then discretized according to $\overline{\boldsymbol{A}}_t^{(d)} = \exp(\Delta_t^{(d)} \boldsymbol{A}^{(d)}), \overline{\boldsymbol{B}}_t^{(d)} = \Delta_t^{(d)} \boldsymbol{B}_t$. In contrast to S4, where $\overline{\boldsymbol{B}}^{(d)}$ varies independently across channels, the differences in $\overline{\boldsymbol{B}}^{(d)}$ in S6 are solely due to the scalar $\Delta_t^{(d)}$. Additionally, S6 uses the same $\boldsymbol{C}_t$ for all channels at each time step $t$, unlike S4, which has unique $\boldsymbol{C}^{(d)}$ for each channel.

**Mamba.** Similar to the Transformer block, which consists of attention and linear layers, the Mamba block proposed by Gu & Dao (2024) features an S6 module, a point-wise 1D causal convolution layer (Conv1d) for token mixing, linear layers — including input ($\boldsymbol{W}_{\text{in}}$) and output ($\boldsymbol{W}_{\text{out}}$) projection layers and a gated MLP. Mamba, primarily allocating its parameters in $\boldsymbol{W}_{\text{in}}$ and $\boldsymbol{W}_{\text{out}}$, is inspired by H3 (Fu et al., 2022).

## 4 BENCHMARKING PEFT METHODS ON SSM-BASED MODELS

In this section, we examine the effectiveness of popular PEFT methods when applied naively to SSM-based models, specifically Mamba-I (130M and 1.4B). Further evaluation on other models, including Mamba-II (Dao & Gu, 2024) and Jamba (Lieber et al., 2024), is deferred to Sec. C.5.

**Experiment Setup.** We consider two main categories of PEFT methods: parameter-based and prompt-based. From each category, we evaluate two representative methods. For parameter-based methods, we select BitFit (Zaken et al., 2022) and LoRA (Hu et al., 2021). For prompt-based methods, we choose prefix-tuning (Li & Liang, 2021) and prompt tuning (Lester et al., 2021). For BitFit, fine-tuning is performed on all bias terms present in the Mamba architecture, specifically the biases of the Conv1d and the linear projection layer of step size $\boldsymbol{\Delta}$. For prefix-tuning, we

adopted the huggingface implementation (Mangrulkar et al., 2022) to construct a MLP, employing the overparameterization technique to ensure stable optimization.

We consider five datasets spanning diverse domains: the GLUE natural language understanding benchmark (Wang et al., 2019), the DART RDF-to-text generation benchmark (Nan et al., 2021), the Spider text-to-SQL generation benchmark (Yu et al., 2018), and CIFAR-10 for computer vision tasks (Krizhevsky et al., 2009). A more detailed introduction of the datasets considered in this paper is provided in Sec. B. Notably, prefix-tuning requires substantially more parameters than other PEFT methods, as it employs a multilayer perceptron at each layer to project a fixed sequence into soft tokens for training stability. For all other PEFT methods, we constrain the trainable parameters to fewer than 0.5% for language tasks and 1% for vision tasks, ensuring a fair comparison. The higher allowance for vision tasks accommodates the need for extensive fine-tuning for new modalities. Consequently, LoRA adapters are applied exclusively to linear projection matrices, leaving the SSM modules unchanged to comply with these parameter constraints.

Additional experiments on models like Jamba Lieber et al. (2024) and Mamba-II Dao & Gu (2024), and advanced PEFT methods yang Liu et al. (2024) are covered in Secs. C.5 and C.6.

**Results.** Table 1 presents our results. Parameter-based PEFT methods generally outperform prompt-based methods significantly, despite using the same number of trainable parameters—except for prefix-tuning, which underperforms despite using more parameters. LoRA consistently achieves the best performance across all tasks and metrics, occasionally surpassing full fine-tuning while tuning less than 1% of parameters. Detailed results for GLUE and Spider are available in Sec. C.2.

These findings above raise two critical questions: (i) Why do existing prompt-based PEFT methods lose effectiveness when applied to SSM-based models? (ii) Can LoRA achieve better performance when applying on both linear projection matrices and SSM modules? To address these questions, we conduct both theoretical analysis and further empirical studies on prompt-based PEFT methods and LoRA in the context of SSMs.

| Dataset | GLUE | DART | | SAMSum | | | Spider | CIFAR-10 |
|---|---|---|---|---|---|---|---|---|
| Metric (↑) | Avg. Score | METEOR | BLEU | R1 | R2 | RL | Acc. | Acc. |
| Prompt Tuning | 63.8 | 66.2 | 39.8 | 50.1 | 25.6 | 41.6 | 43.6 | 30.4 |
| Prefix-Tuning | 68.6 | 66.6 | 42.5 | 50.6 | 26.5 | 42.1 | 39.7 | 41.0 |
| BitFit | 76.8 | 67.0 | 43.7 | 50.3 | 25.7 | 41.9 | 48.4 | 44.4 |
| LoRA (Linear Projection Matrices) | **80.5** | **70.4** | **49.1** | **50.9** | **27.0** | **42.3** | **57.5** | **61.1** |
| Full Fine-Tuning | 80.5 | 71.0 | 51.8 | 51.2 | 27.3 | 42.9 | 66.2 | 60.0 |

Table 1: **Benchmarking popular Parameter-Efficient Fine-Tuning (PEFT) methods on five real-world datasets.** R1/R2/RL stand for ROUGE-1/2/L. For all PEFT methods except prefix-tuning, we report the best results for cases where fewer than 0.5% of parameters are tunable for language tasks and fewer than 1% for vision tasks (i.e., CIFAR-10) after comprehensive hyperparameter search. Prefix-tuning is an exception, as it requires training a multilayer perceptron at each layer to project a fixed sequence into soft tokens for training stability, consuming more trainable parameters than our threshold. **Bold** numbers indicate outperformance over all PEFT methods, while underlined numbers indicate outperformance over full fine-tuning.

## 4.1 LIMITATIONS OF APPLYING EXISTING PROMPT-BASED METHODS ON SSMS

This part addresses our first question arised in this section: Why do existing prompt-based PEFT methods lose effectiveness when applied to SSM-based models? We approach this by establishing an upper bound on the performance of existing prompt-based PEFT methods.

A key feature of SSMs is their next-token prediction mechanism, which relies solely on the current token and hidden states, without considering previous tokens directly. The hidden states encapsulate all information from preceding tokens. Consequently, prepending tokens to an SSM is functionally equivalent to tuning the initial state, as demonstrated by the following proposition. The formal version and proof of Proposition 1 are presented in Sec. C.3.

**Proposition 1** (Informal: Expressivity of Prefix-Tuning on SSMs). *The maximum expressiveness achievable via prefix-tuning on SSMs is equivalent to the expressiveness of solely tuning the initial hidden state $h_0$.*

To evaluate the performance of initial state tuning, we conducted experiments on the GLUE benchmark, comparing prompt-tuning, prefix-tuning, initial state tuning, and LoRA across seven GLUE tasks. Table 2 presents our findings. The results demonstrate that initial state tuning generally outperforms prefix-tuning, corroborating our analysis. However, LoRA significantly surpasses initial state tuning in performance. These observations lead us to conclude that the limitations of initial state tuning inherently constrain the potential of existing prompt-based methods, preventing them from outperforming LoRA in the context of SSM-based models. While the reason for the underperformance of initial state tuning is unclear, we identify explaining it as an interesting direction for future research. Nevertheless, we propose a plausible explanation. We hypothesize that SSM's exclusive reliance on hidden states, without direct access to previous tokens or states, severely restricts the impact of initial state tuning, particularly for long sequences. This aligns with the findings of Fu et al. (2022), which demonstrate SSM's limitations in recalling older tokens.

| Task | RTE | MRPC | CoLA | SST-2 | QNLI | QQP | MNLI | Avg. Score |
|---|---|---|---|---|---|---|---|---|
| Prompt Tuning | 56.0 | 71.6 | 12.0 | 89.4 | 76.8 | 79.6 | 61.5 | 63.8 |
| Prefix-Tuning | 69.5 | 75.7 | 43.4 | 91.5 | 83.4 | 83.1 | 35.6 | 68.6 |
| Initial State Tuning | 66.8 | 75.1 | 52.4 | 92.4 | 86.4 | 86.1 | 78.5 | 76.8 |
| LoRA (Linear Projection Matrices) | **70.4** | **82.8** | **60.6** | **92.4** | **88.4** | **87.7** | **81.5** | **80.5** |

Table 2: **Comparison of prompt-tuning, prefix-tuning, initial state tuning, and LoRA on seven tasks from the GLUE benchmark.** We report the Matthews correlation (↑) for CoLA, overall (matched and mismatched) accuracy (↑) for MNLI, and accuracy for other tasks. Initial State Tuning and LoRA are constrained to use less than 0.5% trainable parameters. **Bold** numbers indicate the best performance across all three methods, while underlined numbers show the highest score among prompt-based methods (prefix-tuning and initial state tuning). Initial state tuning outperforms prefix-tuning and prompt-tuning on five out of seven tasks, while LoRA consistently outperforms all prompt-based methods.

## 4.2 OPTIMAL APPLICATION OF LoRA IN SSM-BASED MODELS

In our previous experiments, we applied LoRA exclusively to linear projection matrices. However, SSM-based models typically comprise various modules, including S4 (convolution layer), S6, and multiple distinct linear projection matrices. To investigate the impact of applying LoRA to different components, we conduct a comprehensive study across five datasets.

| Model | | Mamba-130M | | | | Mamba-1.4B | | | |
|---|---|---|---|---|---|---|---|---|---|
| Dataset | | GLUE | DART | | CIFAR-10 | | SAMSum | | Spider |
| Metric (↑) | Params. (%) | Avg. Score | METEOR | BLEU | Acc. | Params. (%) | R1 R2 | RL | Acc. |
| SSM Modules | .92 | 79.3 | 69.9 | **50.8** | 44.0 | .46 | 50.5 26.4 | 42.2 | 56.3 |
| Linear Projection Matrices | 1.02 | **80.5** | **71.2** | 49.2 | **62.8** | .51 | **50.8** **26.9** | **42.8** | 54.7 |
| Both | 1.92 | 80.2 | 71.0 | 49.5 | 60.4 | .97 | **50.8** 26.6 | 42.7 | **56.4** |

Table 3: **For LoRA, targeting only the linear projection matrices yields better performance than applying it to all modules in Mamba.** Consistent rank is maintained across all three methods.

We examine LoRA's performance when applied to SSM modules and linear projection matrices separately, as well as in combination. For linear projections, we test LoRA on all possible matrices. For SSM modules, we apply LoRA to all weight matrices (e.g., weight matrices of input-dependent step size $\Delta$) in SSM modules. For the state transition matrices $A$, given their diagonal structure for each channel, we treat them as vectors, concatenate the channels into a matrix, and apply LoRA. The results are presented in Table 3. We observe that applying LoRA to linear projection matrices achieves superior performance on six out of eight metrics. Interestingly, additional tuning of SSM modules lead to decreased performance in some cases. This suggests that LoRA might not be well-suited for tuning SSM modules, while being highly effective for linear projection matrices. This conclusion also extends to other models, including Mamba-II, Jamba, and an advanced variant of LoRA known as DoRA (yang Liu et al., 2024), with corresponding results available in Sec. C.5 and Sec. C.6.

To further elucidate this concept, we present the following lemma, which examines a simplified model architecture consisting of S6 with a linear input projection matrix at each layer. We demonstrate that fine-tuning the projection matrix $W_{\text{in}}$ encompasses the expressivity of fine-tuning the parameters $W_B$, $W_C$, and $W_{\Delta,\uparrow}$.

**Lemma 2** (Expressivity of Fine-Tuning Projection Matrices). *Consider an S6 with an additional linear input projection matrix $W_{in}$. Denote the input-dependent SSM parameters $\{\{\overline{A}_n^{(d)}\}_{d=1}^{D}, \overline{B}_n, C_n\}_{n=1}^{N}$ as $\theta(\cdot; \{A^{(d)}\}_{d=1}^{D}, W_B, W_C, W_{\Delta,\uparrow}, W_{\Delta,\downarrow}, W_{in})$. For any given $\overline{W}_B$, $\overline{W}_C$, and $\overline{W}_{\Delta,\uparrow}$, there exists a $\widehat{W_{in}}$ such that for any input sequences $X \in \mathbb{R}^{D \times N}$,*

$$\theta(X; \{A^{(d)}\}_{d=1}^{D}, \overline{W}_B, \overline{W}_C, \overline{W}_{\Delta,\uparrow}, W_{\Delta,\downarrow}) = \theta(X; \{A^{(d)}\}_{d=1}^{D}, W_B, W_C, W_{\Delta,\uparrow}, W_{\Delta,\downarrow}, \widehat{W_{in}}).$$

We expand upon this section in Sec. C.4, where we provide more detailed statements of the above assertion and its corresponding proofs. Additionally, we empirically examine applying LoRA to different weight matrices of Mamba, which incorporates multiple linear projection matrices in each layer, including output projection matrices $W_{\text{out}}$ after the S6 module and input projection matrices $W_{\text{in}}$ before the gating and convolutional layer. Our experiment results, however, reveal that applying LoRA to different matrices achieves similar performance, as detailed in Sec. C.4.

## 5 DIMENSION SELECTION FOR TUNING STATE-SPACE MODELS

In Sec. 4.2, we demonstrate the efficacy of LoRA in fine-tuning linear projection matrices. Theoretically, fine-tuning all components should offer greater expressive power. However, Table 3 indicates that applying LoRA to SSM modules might paradoxically decrease performance. Therefore, we aim to develop an algorithm specifically tailored for tuning SSM modules. To achieve this, we first seek to understand the relative importance of different parameters within SSM modules.

### 5.1 UNDERSTANDING THE ROLES OF STATE MATRIX $A$, INPUT TRANSITION VECTOR $B$, AND OUTPUT MAPPING VECTOR $C$ FOR A SINGLE CHANNEL IN S4 MODULES

**Problem Setting.** Inspired by Zeng & Lee (2024)'s theoretical analysis of LoRA's expressive power, we adopt a similar framework to explore the expressive potential of various parameters in the S4 model. Specifically, we assume a target model that performs well on the intended task and a frozen model, which may be either pretrained or randomly initialized. Our goal is to identify a parameter-efficient method to update the frozen model so that it becomes functionally equivalent to the target model. In alignment with Zeng & Lee (2024), we assume that the frozen model's capacity is equal to or exceeds that of the target model. This assumption is based on two main considerations: (i) analytical tractability, which necessitates that the frozen model must have the potential to match the functionality of the target model, and (ii) a practical rationale, given that the models typically used in practice are often overparameterized. Assume that both the target model and the frozen model are S4, with the target model having a hidden state dimension $H_\star$ and the frozen model having a hidden state dimension $H \geq H_\star$. Meanwhile, suppose that all the hidden dimensions of both models are valid, meaning that none of the parameter elements are zero. The target model, frozen model, and the updated model after tuning the parameters on the frozen model can be formulated using discretized parameters $\overline{A}, \overline{B}, C$ as follows:

(Target model) $\quad f^\star(x)_n = \sum_{m=1}^{n} C_\star \overline{A}_\star^{m-n} \overline{B}_\star x_m$, where $\text{diag}(\overline{A}_\star), \overline{B}_\star, C_\star \in \mathbb{R}^{H_\star}$,

(Frozen model) $\quad f_0(x)_n = \sum_{m=1}^{n} C \overline{A}^{m-n} \overline{B} x_m$, where $\text{diag}(\overline{A}), \overline{B}, C \in \mathbb{R}^{H}$,

(Updated model) $\quad \hat{f}(x)_n = \sum_{m=1}^{n} \widehat{C} \widehat{\overline{A}}^{m-n} \widehat{\overline{B}} x_m$, where $\text{diag}(\widehat{\overline{A}}), \widehat{\overline{B}}, \widehat{C} \in \mathbb{R}^{H}$.

**Parameter Efficiency Analysis on S4.** Let $\mathcal{P}^H$ denote the set of all $H \times H$ permutation matrices. Given this formulation, we present our first analysis of parameter efficiency for the S4 model in the following lemma. This analysis is based on the parameters after necessary discretization $(\overline{A}, \overline{B}, C)$.

**Lemma 3** (Essential Discretized Parameter Set for S4). *Consider the parameters after discretization, i.e., $\overline{A}, \overline{B}, C$. To achieve functional equivalence between the updated model and the target model,*

*i.e., $\hat{f} \equiv f^\star$, it is sufficient to tune the following number of parameters:*

$$\min_{\boldsymbol{P} \in \mathcal{P}^H} \overbrace{\left\| \left[ \boldsymbol{P}^\top (\mathrm{diag}(\overline{\boldsymbol{A}}) \otimes \overline{\boldsymbol{B}} \otimes \boldsymbol{C}^\top) \right]_{(H_\star+1):H} \right\|_0}^{\text{eliminating redundant dimensions}} + \underbrace{\left\| \left[ \boldsymbol{P}^\top \overline{\boldsymbol{A}} \boldsymbol{P} \right]_{1:H_\star, 1:H_\star} - \overline{\boldsymbol{A}}_\star \right\|_0}_{\text{aligning the state matrix}} + \overbrace{\underbrace{\left\| \left[ \boldsymbol{P}^\top (\overline{\boldsymbol{B}} \otimes \boldsymbol{C}^\top) \right]_{1:H_\star} - \overline{\boldsymbol{B}}_\star \otimes \boldsymbol{C}_\star^\top \right\|_0}_{\text{aligning input-output interactions}}}^{\text{aligning used dimensions with target model}}.$$

This lemma highlights the significance of identifying essential hidden state dimensions. The term $\left\| \left[ \boldsymbol{P}^\top (\mathrm{diag}(\overline{\boldsymbol{A}}) \otimes \overline{\boldsymbol{B}} \otimes \boldsymbol{C}^\top) \right]_{(H_\star+1):H} \right\|_0$ underscores the importance of excluding redundant dimensions. This can be achieved by either directly removing these dimensions from the state matrix $\overline{\boldsymbol{A}}$, or by updating $\overline{\boldsymbol{B}}$ or $\boldsymbol{C}$ to ensure that only the selected hidden state dimensions are utilized during the input transition or output mapping phases. Once redundant dimensions are filtered out, tuning only the essential dimensions is sufficient to align the updated model with the target model. Proofs and further details are provided in Sec. D.1.

## 5.2 SSM DIMENSION SELECTION ALGORITHM

Inspired by Lemma 3, we introduce the *Dimension Selection* algorithm to construct adapters on SSMs for fine-tuning. This algorithm first selects unimportant dimensions and sets them to zero, filtering out irrelevant information based on Lemma 3. For enhanced parameter efficiency, we then update only the most important channels and state dimensions within these selected subsets. Regardless of other selections, we consistently tune the coefficients of residual connections and biases in linear projections, as these components contain a negligible number of parameters. However, we will later demonstrate that in practice, tuning residual connections and biases is unnecessary. The detailed pseudo-code is presented in Alg. 1. Given that tuning $\boldsymbol{C}$ alone is as effective as tuning both $\boldsymbol{B}$ and $\boldsymbol{C}$ for S4 (Gupta et al., 2022), subsequent discussions on S4 will focus solely on $\boldsymbol{C}$, excluding $\{\boldsymbol{B}^{(d)}\}_{d=1}^D$ for simplicity, without loss of generality.

---

**Algorithm 1:** Dimension Selection Algorithm for S4

**Input:** Dataset $\mathcal{D}$, warmup epochs $E_0$, train epochs $E$, number of layers $L$, total channels $D$, total states $H$, initial state sparsity $\beta_0$, initial channel sparsity $\alpha_0$, state update fraction $\beta$, channel update fraction $\alpha$

**Output:** Model adapter

```
/* Warmup Epochs                                                    */
```
1 Update SSM modules using $\mathcal{D}$ for $E_0$ epochs;
```
/* Setup Adapters                                                   */
```
2 **for** $l = 1$ **to** $L$ **do**
```
   /* Set dimensions as zero                                        */
```
3      Sort channels based on magnitude of $\overline{\boldsymbol{A}}^{(d)}$ at each channel;
4      Set final $(1 - \beta_0)D$ as zero by letting $\boldsymbol{C}^{(d)} = \boldsymbol{0}$, denote non-zero channels as set $\mathbb{D}$;
5      **for** $d \in \mathbb{D}$ **do**
6          Sort states based on magnitude of $\bar{A}_h^{(d)}$ at each state dimension;
7          Set final $(1 - \alpha_0)H$ as zero by letting corresponding $C_h^{(d)} = 0$, denote non-zero states as set $\mathbb{H}$;
```
   /* Unfreeze dimensions                                           */
```
8      Sort non-zero channels $\mathbb{D}$ based on magnitude of parameter changes at each channel;
9      Denote first $\beta|\mathbb{D}|$ as $\mathbb{D}'$;
10     **for** $d \in \mathbb{D}'$ **do**
11         Sort non-zero state dimensions based on magnitude of parameter changes;
12         Construct adapter to update first $\alpha|\mathbb{H}|$ states at $d$-th channel;
```
   /* Include Residual Connections and Bias                         */
```
13     Construct adapter for all residual connections and bias;

---

We refer to our method as SDLoRA. This approach extends beyond applying LoRA to linear projection matrices by *S*electively updating certain subset of channels and states *D*imensions, which are chosen by Alg. 1. In Sec. D.9, we analyze the overhead of SDLoRA and demonstrate that

the additional dimension selection algorithm introduces only a marginal increase in computational overhead. Overall, SDLoRA is not only faster but also more memory-efficient compared to LoRA.

Our analysis considers cases where each input token $x_t \in \mathcal{X}$, with $\mathcal{X} \in \mathbb{R}^D$ bounded, and the input sequence length is finite. The following theorem elucidates the expressive capacity of SDLoRA on deep S4 models. For proof and additional details, refer to Sec. D.2.

**Theorem 4** (Expressive Power of SDLoRA on Deep S4 Models). *Consider a $D$-dimensional input sequence. Assume that the linear layers in the model have linear activation functions. Using SDLoRA, any deep S4 model with $H$ hidden states per channel and $L$ layers can be updated to accurately present any target deep S4 model without residual connections, having a reduced hidden state dimension $H^\star < H$, and fewer layers $L^\star < L$. This can be achieved by selectively fine-tuning at most $\lceil DL^\star/L \rceil$ channels, $H^\star$ hidden states on SSM modules, applying rank-$\lceil \frac{L}{L^\star} \rceil$ updates on linear projection matrices and updating residual connections and biases at each layer, while additionally fully fine-tuning the linear projection matrix of the last layer only.*

This theorem demonstrates that a larger pretrained model requires selecting fewer channels and hidden states at each layer. Furthermore, if the target task is less complex — evidenced by a smaller target model with fewer layers $L^\star$ and hidden states $H^\star$ — the number of channels and hidden states needed is also reduced. This finding aligns with the theoretical analysis of LoRA presented in Zeng & Lee (2024), which shows that larger pretrained models require fewer learnable parameters (referred to as "lower rank" in their context) during fine-tuning, especially for simpler tasks. Although this theorem is constrained by the assumptions of linear activations and the absence of residual connections in the target model, while also requiring fully fine-tuning the linear project matrix of last layer, our findings have broader implications. Our following experimental results suggest that these findings generalize beyond these restrictions.

### 5.3 EMPIRICAL EVALUATION ON DEEP S4 MODELS

In this experiment, we seek to validate the theoretical guarantees for SDLoRA under more general conditions, including residual connections in the target model and ReLU activations in both frozen model and target model, without full fine-tuning the linear projection matrix of the last layer. Additionally, we assess SDLoRA's empirical performance on both synthetic and real datasets. More experiment setup details are provided in Sec. D.3.

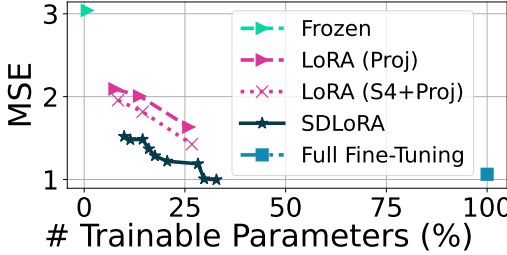

Figure 1: Approximation error of PEFT methods on deep S4 models for synthetic experiments.

| Method | # Params (%) | Accuracy |
|---|---|---|
| Frozen | 0.00 | 73.9 |
| LoRA (Proj) | 16.00 | 77.6 |
| LoRA (S4+Proj) | 15.52 | 77.6 |
| SDLoRA | 11.17 | **78.0** |
| Full Fine-Tuning | 100.00 | 77.6 |

Table 4: Accuracy comparison between SDLoRA and LoRA on deep S4 models for CIFAR-10 (Krizhevsky et al., 2009).

**Synthetic Dataset.** For the synthetic dataset, we employ a regression setting to validate our theoretical results. *(Experiment Setting)* We randomly initialize two models: a one-layer deep S4 model as the target and a four-layer deep S4 model as the frozen model. The task is to update the frozen model to match the functionality of the target model. We generate an input sequence $X$ of length 200 and dimension 64, with values uniformly drawn from integers between 0 and 9. This input is then processed through the target model to obtain the corresponding outputs. These input-output pairs are used to train the frozen model over 500 iterations using the Mean Squared Error (MSE) loss.

*(Results)* Figure 1 displays the MSE, averaged across all tokens, plotted against the trainable parameters of different methods. We observe that by using only $\approx 28\%$ of the total parameters of the frozen S4 model, SDLoRA closely approximates the performance of the target S4 model, achieving results comparable to full fine-tuning, thereby substantiating our theorem. Meanwhile, we observe that SDLoRA outperforms both the approach of applying LoRA solely to linear projection matrices and the approach of applying LoRA to both the S4 module and linear projection matrices. In this

latter approach, the diagonal vectors of state matrices $A^{(d)}$, input transition vectors $B^{(d)}$ and output mapping vectors $C^{(d)}$ are naively concatenated across $D$ channels into three $D \times H$ matrices before low-rank updates are applied. In Sec. D.3, we also evaluate an extension of SDLoRA that performs sparse tuning on the linear projection matrices by updating only the columns corresponding to the channels selected by Alg.1, instead of applying LoRA. This extension shows promising results.

**CIFAR-10.** Previous work (Dinh et al., 2022) demonstrates that large language models can be fine-tuned for image classification tasks. Here, we consider the this challenging task of adapting SSMs for computer vision. In this experiment, we conduct experiments on the CIFAR-10 dataset (Krizhevsky et al., 2009). We employ an eight-layer deep S4 model with a hidden state dimension of 16 and a model dimension of 64. Since pretrained deep S4 models are not available, we simulate a pretrained scenario by fully updating the model for 50 epochs first, then subsequently evaluating the PEFT methods over an additional 5 epochs. The results, as reported in Table 4, indicate that SDLoRA outperforms LoRA with fewer trainable parameters.

## 5.4 EMPIRICAL EVALUATION ON MAMBA

Lastly, we conduct experiments on pretrained Mamba models. We consider four datasets, using Mamba-130M for GLUE and DART, and Mamba-1.4B for SAMSum and Spider. We evaluate three configurations each for LoRA and SDLoRA, applying LoRA to distinct parameter subsets and varying SDLoRA's state freeze ratios while maintaining a 99% channel freeze ratio. In this experiment, we allow channels and states to learn directly from the datasets without manually setting any to zero. We then select a LoRA-rank such that all configurations have a similar number of trainable parameters for a fair comparison. Residual connections and biases are frozen in this experiment. All reported values represent averages across three simulations, with learning rates independently selected for each simulation. For more details, please see Sec. D.4. The experimental results are reported in Table 5. The results demonstrate that SDLoRA outperforms LoRA for fine-tuning the SSM even when 99% of the channels are frozen. This result underscores the efficacy of SDLoRA. The same conclusions are further supported by additional models, including Jamba and Mamba-II, as well as more datasets, such as CelebA (Liu et al., 2015), and other LoRA variants, including DoRA and LoRA+ (Hayou et al., 2024). The corresponding results are presented in Secs. D.5 to D.8, respectively.

| Model | | Mamba-130M | | | | | | | Mamba-1.4B | | | | | | | | |
|---|---|---|---|---|---|---|---|---|---|---|---|---|---|---|---|---|---|
| Method | Params (%) | GLUE | | DART | | | | Params (%) | SAMSum | | | | | | Spider | |
| | | Avg. Score (↑) | | BLEU (↑) | | METEOR (↑) | | | R1 (↑) | | R2 (↑) | | RL (↑) | | Acc. (↑) | |
| | | Val | Test | Val | Test | Val | Test | | Val | Test | Val | Test | Val | Test | Val | Test |
| LoRA | .3178 | 80.71 | 78.74 | 50.44 | 41.27 | 70.00 | 65.84 | .1594 | 51.59 | 50.56 | 27.66 | 26.49 | 42.87 | 42.22 | 82.08 | 61.19 |
| | .3600 | 80.79 | 79.39 | 51.03 | 42.02 | 70.16 | 66.18 | .1810 | 51.61 | **51.03** | **28.15** | 26.81 | 43.18 | 42.36 | 83.52 | **62.64** |
| | .3883 | 80.39 | 79.49 | 50.70 | 41.55 | 69.83 | 65.98 | .1947 | 51.48 | 50.90 | 27.90 | 26.63 | 43.26 | 42.41 | 82.98 | 59.25 |
| SDLoRA | .3492 | 80.93 | **79.75** | 51.45 | 42.37 | 70.45 | **66.60** | .1760 | 51.63 | 50.90 | 27.97 | **26.86** | 43.32 | **42.52** | 84.36 | 62.57 |
| | .3498 | **81.05** | 79.16 | 51.47 | **43.85** | **70.46** | 66.38 | .1761 | 51.61 | 50.76 | 28.02 | 26.65 | 43.38 | 42.29 | **84.48** | 59.96 |
| | .3509 | 80.67 | 78.73 | **51.54** | 42.56 | 70.45 | 66.45 | .1764 | **51.74** | 50.86 | 28.08 | 26.54 | 43.39 | 42.19 | 84.19 | 61.25 |

Table 5: **Performance comparison between SDLoRA and LoRA on pretrained Mamba models.** **Bold** numbers indicate the best performance for each task. Underlined numbers indicate that the model outperforms all models fine-tuned via the alternative method for the same task (e.g., SDLoRA outperforms all LoRA methods, or vice versa). On Mamba-130M, we compare the performance of SDLoRA and LoRA on GLUE (Wang et al., 2019) and DART (Nan et al., 2021) benchmarks. On Mamba-1.4B, we compare performance of SDLoRA and LoRA on SAMSum (Gliwa et al., 2019) and Spider (Yu et al., 2018) benchmarks. R1/R2/RL stand for ROUGE-1/2/L.

## 6 CONCLUSION

In this paper, we present the first study on the performance of PEFT methods applied to SSM-based models. Our evaluation of existing PEFT methods provides valuable insights and guidelines for future researchers to parameter-efficiently fine-tune SSM-based models to other domains. Moreover, we take a first step in establishing a theoretical framework for studying PEFT methods on SSM-based models. Furthermore, we introduce SDLoRA, the first PEFT method specifically tailored for SSM-based models, which outperforms existing methods. While our work offers numerous valuable insights, we discuss limitations and further works in Sec. E.

## REPRODUCIBILITY STATEMENT

We are committed to ensuring the reproducibility of our research. To this end, we provide our complete implementation at https://anonymous.4open.science/r/ssm-peft-8F6F/. This repository contains instructions needed to reproduce the results reported in our work. We also include detailed documentation and example commands for running the experiments, along with requirements for dependencies to facilitate a smooth setup.

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

# Appendix

## A    IN-DEPTH INTRODUCTION OF BASELINES

In this section, we provide a more detailed description of the baseline methods.

**LoRA (Hu et al., 2021).**    LoRA (Low-Rank Adaptation) focuses on fine-tuning large models by freezing most of the pretrained parameters and injecting trainable low-rank matrices into each layer of the Transformer's architecture. The intuition behind using low-rank matrices comes from linear algebra, where a large matrix can be closely approximated by the product of two smaller matrices. The number of trainable parameters can be controlled with the rank of the low-rank matrices. LoRA also uses a scaling parameter (LoRA alpha) for the weight matrices to control the balance of the original model weights and LoRA weights during training. After fine-tuning, LoRA weights can be merged with the original model weights, introducing no additional inference overhead.

**Prompt Tuning (Lester et al., 2021).**    Prompt tuning freezes all model weights and prepends a trainable soft prompt to the input prompt. The soft prompt consists of trainable virtual tokens, which are continuous. At inference time, prompt tuning introduces an inference overhead based on the number of virtual tokens used.

**Prefix-Tuning (Li & Liang, 2021).**    Prefix-tuning also prepends trainable tokens to the input like prompt tuning but injects separate prefixes in every layer. For each Transformer layer, prefix-tuning prepends trainable embeddings to the attention's $K$ and $V$ matrix. The authors have found that directly training these prefixes can lead to unstable training, so they propose to over-parameterize them with a large MLP to increase training stability. After training, the MLP can be dropped. Like prompt tuning, prefix-tuning introduces an inference overhead, scaling linearly with the number of trainable embeddings.

**BitFit (Zaken et al., 2022).**    BitFit is a simple but effective PEFT method that freezes all model weights except the bias terms, consequently greatly reducing the number of trainable parameters. As no additional parameters are added, no inference overhead occurs.

## B    DETAILS OF DATASETS

In this paper, we consider five datasets across three domains: (i) Natural Language Understanding (NLU), represented by GLUE (Wang et al., 2019); (ii) Natural Language Generation (NLG), including SAMSum (Gliwa et al., 2019), Spider (Yu et al., 2018) and DART (Nan et al., 2021); and (iii) Computer Vision (CV), represented by CIFAR-10 (Krizhevsky et al., 2009).

**GLUE (Wang et al., 2019).**    The GLUE (General Language Understanding Evaluation) benchmark is a collection of datasets used for training, evaluating, and analyzing natural language understanding models across a range of diverse tasks. The benchmark includes nine sentence- or sentence-pair language understanding tasks that require various features of understanding, such as sentiment analysis, linguistic acceptability, semantic textual similarity, and question answering. We use seven datasets from the GLUE benchmark (RTE, MRPC, CoLA, SST-2, QNLI, QQP, MNLI) where the model has to choose between two or three (for MNLI) different choices for the respective task. Except for CoLA, we evaluate all used datasets with the accuracy metric. For CoLA, Matthews correlation is employed.

**SAMSum (Gliwa et al., 2019).**    SAMSum is a dataset for dialogue summarization research, comprising approximately 16,000 synthetic text conversations with accompanying summaries. Created by English-fluent linguists, these exchanges simulate real-world digital communications across various topics and styles. The conversations range from informal to formal, incorporating elements like slang and emoticons to reflect authentic messaging patterns. Each dialogue is paired with a concise, third-person summary, capturing its essential content. This structure makes SAMSum particularly useful for developing and evaluating automated summarization systems capable of processing conversational text.

| Data | | Size (Train) | Size (Val) | Size (Test) | Max. seq. len. | #Epochs | Mamba Size | Metrics |
|---|---|---|---|---|---|---|---|---|
| GLUE | RTE | 1992 | 498 | 277 | 291 | 10 | 130M | Accuracy |
| | MRPC | 2934 | 734 | 408 | 105 | 10 | 130M | Accuracy |
| | CoLA | 6840 | 1711 | 1043 | 47 | 10 | 130M | Matthews corr. |
| | SST-2 | 53879 | 13470 | 872 | 68 | 10 | 130M | Accuracy |
| | QNLI | 83794 | 20949 | 5463 | 602 | 10 | 130M | Accuracy |
| | QQP | 291076 | 72770 | 40430 | 316 | 3 | 130M | Accuracy |
| | MNLI | 314161 | 78541 | 19647 | 425 | 3 | 130M | Accuracy |
| Spider | | 5543 | 1375 | 1034 | 1412 | 10 | 1.4B, 2.8B | Accuracy |
| SAMSum | | 14732 | 818 | 819 | 1174 | 10 | 1.4B | ROUGE |
| DART | | 62659 | 2768 | 5097 | 491 | 10 | 130M | METEOR, BLEU |
| CIFAR-10 | | 40000 | 10000 | 10000 | 1730 | 5 | 130M | Accuracy |

Table 6: **Datasets and models for our experiments.** For each dataset, we report the number of training, validation, and test samples, maximum sequence length, training epochs, model size, and evaluation metric used.

**Spider (Yu et al., 2018).** Spider is a large-scale, complex, and cross-domain semantic parsing and text-to-SQL dataset. It contains about 10,000 annotated SQL queries, distributed across 200+ databases, each with multiple tables. We follow Scholak et al. (2021) and use about 7,000 examples for training and about 1,000 examples for validation, where we ignore sequences longer than 1536 tokens. The dataset consists of English question and SQL query pairs, which cover a wide range of SQL operations including SELECT, WHERE, COUNT, GROUP BY, ORDER BY, JOIN, and more. Given an English question and an SQL database scheme, the task for the model is to translate the English question into an appropriate SQL statement. Evaluation is performed via accuracy where the output is considered as correct if the model's predicted SQL query and the included GT SQL query give the same result when executed on the database. The dataset additionally categorizes each query into easy (25%), medium (40%), hard (20%), and extra hard (15%) based on the complexity of the required SQL statement. For evaluation, we report the execution accuracy of all categories.

**DART (Nan et al., 2021).** The DART (DAta Record to Text) benchmark is a large-scale, structured dataset designed for RDF-to-text (Resource Description Framework-to-text) generation with 80,000+ instances. The DART benchmark is composed of a collection of structured data triples and corresponding text summaries which are organized into different categories. The task of the DART benchmark is to generate natural language summaries that correctly represent the given structured data inputs. DART is typically evaluated with METEOR and BLEU.

**CIFAR-10 (Krizhevsky et al., 2009).** The CIFAR-10 (Canadian Institute For Advanced Research) dataset is a collection of images that are commonly used to train machine learning and computer vision algorithms. It is one of the most widely used datasets for image classification. The CIFAR-10 dataset contains 60,000 (50,000 for training, 10,000 for validation) $32\times32$ color images in 10 different classes. The 10 different classes are: airplane, car, bird, cat, deer, dog, frog, horse, ship, and truck. There are 6,000 images of each class. For training, we center crop each image to $24\times24$ pixels and flatten each image to a string, with a total of $24\times24\times3$ words, where each word is a number between 0-255 representing the respective pixel value. Although CIFAR-10 is a dataset for computer vision, previous work (Dinh et al., 2022) showed that Transformers can be adapted to the vision domain from the language domain, and we tested this ability on the state-space model.

The dataset characteristics, including our train, validation and test set sizes, sequence lengths, and number of epochs, are summarized in Table 6.

## C  DETAILS OF SEC. 4: BENCHMARKING PEFT METHODS ON SSM-BASED MODELS

In this section, we provide a comprehensive experimental setup, proofs and further discussion of theoretical results, and more detailed experimental outcomes.

### C.1  EXPERIMENT SETUP

For each dataset, we choose the model size of Mamba depending on how challenging the dataset is and perform a small grid search for one epoch on a subset of the data (1k-2k instances) with learning rates $\{4 \times 10^{-1}, 2 \times 10^{-1}, 1 \times 10^{-1}, ..., 1 \times 10^{-5}\}$ to find the optimal learning rate of each PEFT method. Afterward, we train the best setting for each PEFT method on the full data for several epochs (Table 6) using an NVIDIA RTX 3090 GPU for the 130M model and an NVIDIA A100 for the larger 1.4B and 2.8B models in mixed precision (BF16). We only report the validation metric of the best epoch during training (early stopping) in our results. We fine-tune the Mamba models (Gu & Dao, 2024) pretrained from Pile (Gao et al., 2020) with AdamW with a linear learning rate decay schedule. For LoRA we set rank to 8, alpha to 8, and dropout to 0.1 for all experiments. For evaluating NLG tasks, we employ beam search with five beams and a maximum beam length of 1024.

### C.2  EXTENDED RESULTS ON BENCHMARKING EXISTING PEFT METHODS

We present comprehensive fine-tuning results for the GLUE benchmark (Wang et al., 2019), DART dataset (Nan et al., 2021), SAMSum dataset (Gliwa et al., 2019) and Spider dataset (Yu et al., 2018) in Table 7, Table 8, Table 9 and Table 10, respectively. These experimental results encompass various LoRA implementations (on different weight matrices and modules) and provide more fine-grained results across all subtasks.

| Layer | | Method | # Params (%) | RTE | MRPC | CoLA | SST-2 | QNLI | QQP | MNLI | Avg. |
|---|---|---|---|---|---|---|---|---|---|---|---|
| Pretrained | | | 0.00 | 46.9 | 67.9 | 0.0 | 52.4 | 50.5 | 36.8 | 32.3 | 41.0 |
| All | All | Full | 100.00 | 71.1 | 80.6 | 63.2 | 92.2 | 87.4 | 87.9 | 80.8 | 80.5 |
| | | LoRA | 1.92 | 69.9 | 80.9 | 61.4 | 91.9 | 88.4 | 87.6 | 81.1 | 80.2 |
| Prompt | Prompt Tuning | 16 tokens | 0.01 | 56.0 | 71.6 | 12.0 | 89.4 | 76.8 | 79.6 | 61.5 | 63.8 |
| | Prefix-Tuning | 1 token (no MLP) | 0.03 | 67.5 | 75.7 | 43.4 | 91.5 | 83.4 | 83.1 | 35.6 | 68.6 |
| Bias | $\beta_\Delta$, Conv1d | BitFit | 0.06 | 69.5 | 80.4 | 54.7 | 92.0 | 86.2 | 85.3 | 77.2 | 77.9 |
| Linear Projection Matrices | All | LoRA | 1.02 | 70.0 | 82.4 | 57.7 | 93.3 | 88.7 | 88.7 | 82.5 | 80.5 |
| | $W_{\text{in},x}$ | LoRA | 0.34 | 70.4 | 82.1 | 57.4 | 91.7 | 88.3 | 87.7 | 81.2 | 79.8 |
| | $W_{\text{in},z}$ | LoRA | 0.34 | 70.0 | 82.4 | 58.1 | 92.4 | 87.3 | 87.3 | 80.4 | 79.7 |
| | $W_{\text{in},x}, W_{\text{in},z}$ | LoRA | 0.68 | 70.4 | 84.3 | 62.4 | 92.5 | 88.6 | 88.3 | 81.7 | 81.2 |
| | $W_{\text{out}}$ | LoRA | 0.34 | 70.4 | 82.8 | 60.6 | 92.4 | 88.4 | 87.7 | 81.5 | 80.5 |
| S6 | All | Full | 4.31 | 69.7 | 78.9 | 59.1 | 91.5 | 88.1 | 87.5 | 80.5 | 79.3 |
| | | LoRA | 0.92 | 66.1 | 78.7 | 57.8 | 90.8 | 87.8 | 86.9 | 79.8 | 78.3 |
| | $A$ | Full | 0.46 | 68.2 | 82.1 | 54.2 | 90.9 | 86.4 | 87.9 | 79.4 | 78.4 |
| | $W_B, W_C, W_{\Delta,\downarrow}$ | Full | 2.28 | 69.7 | 77.0 | 55.8 | 91.4 | 85.4 | 85.0 | 76.8 | 77.3 |
| | | LoRA | 0.69 | 67.9 | 78.9 | 48.8 | 91.4 | 86.9 | 85.8 | 78.6 | 76.9 |
| | $W_{\Delta,\uparrow}$ | Full | 1.40 | 66.1 | 75.2 | 56.7 | 91.1 | 86.2 | 87.1 | 78.5 | 77.3 |
| | | LoRA | 0.23 | 67.1 | 79.9 | 55.1 | 90.9 | 52.7 | 86.6 | 78.7 | 73.0 |
| | Conv1d | Full | 0.14 | 68.2 | 78.4 | 57.9 | 91.1 | 86.0 | 86.0 | 78.0 | 77.9 |
| Others | $D$, LayerNorm | Full | 0.04 | 65.3 | 79.2 | 40.3 | 91.1 | 83.9 | 86.0 | 67.0 | 73.3 |

Table 7: **Full experimental results on the GLUE (Wang et al., 2019) benchmark.** We report accuracy (↑) for RTE, MRPC, SST-2, QNLI, QQP, and MNLI tasks. CoLA performance is measured using Matthews Correlation Coefficient (↑). Mamba-130M is employed in this experiment. In each Mamba block, $W_{\text{in},x}$ and $W_{\text{in},z}$ are input projections that preprocess the input for SSM modules and the gating branch, respectively. $W_{\text{out}}$ denotes the output projection after the gating mechanism. $W_B$ and $W_C$ are weight matrices for computing input-dependent $B_n$ and $C_n$. $W_{\Delta,\downarrow}$ and $W_{\Delta,\uparrow}$ represent down and up projections of low-rank weight matrices in the linear layer computing input-dependent step size $\Delta_n$. $\beta_\Delta$ represents the bias in this linear layer. $D$ denotes the weight of residual connections.

| Layer | | Method | # Params (%) | METEOR | BLEU |
|---|---|---|---|---|---|
| All | All | Full | 100.00 | 71.0 | 51.8 |
| | | LoRA | 1.92 | 71.0 | 49.5 |
| Prompt | Prompt Tuning | 64 tokens | 0.04 | 66.2 | 39.8 |
| | Prefix-Tuning | 64 tokens | 22.69 | 66.6 | 42.5 |
| Bias | $\beta_{\Delta}$, `Conv1d` | BitFit | 0.06 | 67.0 | 43.7 |
| Linear Projection Matrices | All | LoRA | 1.02 | 71.2 | 49.2 |
| | $W_{\mathrm{in},x}$ | LoRA | 0.34 | 70.3 | 48.9 |
| | $W_{\mathrm{in},z}$ | LoRA | 0.34 | 70.4 | 49.1 |
| | $W_{\mathrm{in},x}, W_{\mathrm{in},z}$ | LoRA | 0.68 | 70.9 | 49.5 |
| | $W_{\mathrm{out}}$ | LoRA | 0.34 | 70.7 | 47.0 |
| S6 | All | Full | 4.31 | 70.3 | 48.7 |
| | | LoRA | 0.92 | 69.9 | 50.8 |
| | $A$ | Full | 0.46 | 69.3 | 48.1 |
| | $W_B, W_C, W_{\Delta,\downarrow}$ | Full | 2.28 | 70.1 | 50.0 |
| | | LoRA | 0.69 | 68.8 | 48.0 |
| | $W_{\Delta,\uparrow}$ | Full | 1.40 | 69.6 | 47.2 |
| | | LoRA | 0.23 | 68.9 | 47.0 |
| | `Conv1d` | Full | 0.14 | 68.6 | 47.9 |
| Others | $D$, `LayerNorm` | Full | 0.04 | 67.0 | 44.2 |

Table 8: **Full experimental results on the DART (Nan et al., 2021) benchmark.** We report METEOR (↑) and BLEU (↑) scores. Mamba-130M is utilized in this experiment. In each Mamba block, $W_{\mathrm{in},x}$ and $W_{\mathrm{in},z}$ are input projections that preprocess the input for SSM modules and the gating branch, respectively. $W_{\mathrm{out}}$ denotes the output projection after the gating mechanism. $W_B$ and $W_C$ are weight matrices for computing input-dependent $B_n$ and $C_n$. $W_{\Delta,\downarrow}$ and $W_{\Delta,\uparrow}$ represent down and up projections of low-rank weight matrices in the linear layer computing input-dependent step size $\Delta_n$. $\beta_{\Delta}$ represents the bias in this linear layer. $D$ denotes the weight of residual connections.

| Layer | | Method | # Params (%) | R1 | R2 | RL |
|---|---|---|---|---|---|---|
| All | All | Full | 100.00 | 51.2 | 27.3 | 42.9 |
| | | LoRA | 0.97 | 50.8 | 26.6 | 42.7 |
| Prompt | Prompt Tuning | 64 tokens | 0.01 | 50.1 | 25.6 | 41.6 |
| | Prefix-Tuning | 64 tokens | 12.81 | 50.6 | 26.5 | 42.1 |
| Bias | $\beta_{\Delta}$, `Conv1d` | BitFit | 0.03 | 50.3 | 25.7 | 41.9 |
| Linear Projection Matrices | All | LoRA | 0.51 | 50.8 | 26.9 | 42.8 |
| | $W_{\mathrm{in},x}$ | LoRA | 0.17 | 49.8 | 25.4 | 41.2 |
| | $W_{\mathrm{in},z}$ | LoRA | 0.17 | 50.0 | 26.1 | 41.7 |
| | $W_{\mathrm{in},x}, W_{\mathrm{in},z}$ | LoRA | 0.34 | 50.9 | 27.0 | 42.3 |
| | $W_{\mathrm{out}}$ | LoRA | 0.17 | 49.9 | 25.4 | 41.5 |
| S6 | All | Full | 4.46 | 51.1 | 26.9 | 42.2 |
| | | LoRA | 0.46 | 50.5 | 26.4 | 42.2 |
| | $A$ | Full | 0.23 | 50.1 | 25.9 | 41.7 |
| | $W_B, W_C, W_{\Delta,\downarrow}$ | Full | 2.29 | 50.5 | 26.0 | 41.8 |
| | | LoRA | 0.35 | 50.4 | 26.0 | 41.8 |
| | $W_{\Delta,\uparrow}$ | Full | 1.85 | 50.3 | 25.7 | 41.6 |
| | | LoRA | 0.12 | 50.2 | 25.4 | 41.3 |
| | `Conv1d` | Full | 0.07 | 50.1 | 25.7 | 41.9 |
| Others | $D$, `LayerNorm` | Full | 0.02 | 49.6 | 24.8 | 41.1 |

Table 9: **Full experimental results on the SAMSum (Gliwa et al., 2019) benchmark.** R1, R2, and RL represent ROUGE-1 (↑), ROUGE-2 (↑), and ROUGE-L (↑), respectively. Mamba-1.4B is utilized in this experiment. In each Mamba block, $W_{\mathrm{in},x}$ and $W_{\mathrm{in},z}$ are input projections that preprocess the input for SSM modules and the gating branch, respectively. $W_{\mathrm{out}}$ denotes the output projection after the gating mechanism. $W_B$ and $W_C$ are weight matrices for computing input-dependent $B_n$ and $C_n$. $W_{\Delta,\downarrow}$ and $W_{\Delta,\uparrow}$ represent down and up projections of low-rank weight matrices in the linear layer computing input-dependent step size $\Delta_n$. $\beta_{\Delta}$ represents the bias in this linear layer. $D$ denotes the weight of residual connections.

| Layer | | Method | # Params (%) | All | Easy | Medium | Hard | Extra |
|---|---|---|---|---|---|---|---|---|
| All | All | Full | 100.00 | 66.2 | 84.3 | 69.5 | 53.4 | 43.4 |
| | | LoRA | 0.97 | 56.4 | 76.2 | 57.0 | 47.7 | 34.3 |
| Prompt | Prompt Tuning | 64 tokens | 0.01 | 43.6 | 65.3 | 42.4 | 33.3 | 25.3 |
| | Prefix-Tuning | 64 tokens | 12.81 | 39.7 | 65.7 | 38.6 | 31.0 | 15.1 |
| Bias | $\boldsymbol{\beta_\Delta}$,Conv1d | BitFit | 0.03 | 51.3 | 74.2 | 50.9 | 43.1 | 26.5 |
| Linear Projection Matrices | All | LoRA | 0.51 | 54.7 | 75.0 | 55.6 | 46.0 | 31.3 |
| | $\boldsymbol{W}_{\mathrm{in},x}$ | LoRA | 0.17 | 60.8 | 76.6 | 63.5 | 52.9 | 38.6 |
| | $\boldsymbol{W}_{\mathrm{in},z}$ | LoRA | 0.17 | 46.3 | 68.5 | 45.7 | 36.8 | 24.7 |
| | $\boldsymbol{W}_{\mathrm{in},x}, \boldsymbol{W}_{\mathrm{in},z}$ | LoRA | 0.34 | 57.5 | 77.4 | 58.7 | 45.4 | 37.3 |
| | $\boldsymbol{W}_{\mathrm{out}}$ | LoRA | 0.17 | 61.8 | 81.9 | 65.2 | 45.4 | 39.8 |
| S6 | All | Full | 4.46 | 56.7 | 76.6 | 57.8 | 46.0 | 34.9 |
| | | LoRA | 0.46 | 56.3 | 75.0 | 56.5 | 50.6 | 33.7 |
| | $\boldsymbol{A}$ | Full | 0.23 | 51.1 | 71.4 | 52.5 | 42.5 | 25.9 |
| | $\boldsymbol{W}_B, \boldsymbol{W}_C, \boldsymbol{W}_{\Delta,\downarrow}$ | Full | 2.29 | 47.2 | 72.2 | 46.9 | 35.6 | 22.9 |
| | | LoRA | 0.35 | 55.0 | 73.8 | 56.7 | 44.3 | 33.7 |
| | $\boldsymbol{W}_{\Delta,\uparrow}$ | Full | 1.85 | 56.8 | 77.0 | 59.4 | 43.7 | 33.1 |
| | | LoRA | 0.12 | 58.0 | 78.6 | 59.4 | 48.9 | 33.1 |
| | Conv1d | Full | 0.07 | 53.2 | 74.6 | 52.9 | 43.7 | 31.9 |
| Others | $\boldsymbol{D}$,LayerNorm | Full | 0.02 | 49.6 | 70.6 | 50.4 | 40.2 | 25.9 |

(a) Comprehensive experimental results on Spider using Mamba-1.4B.

| Layer | | Method | # Params (%) | All | Easy | Medium | Hard | Extra |
|---|---|---|---|---|---|---|---|---|
| All | All | Full | 100.00 | 71.8 | 87.5 | 73.5 | 63.8 | 51.8 |
| | | LoRA | 0.80 | 70.9 | 90.7 | 74.0 | 58.6 | 45.8 |
| Prompt | Prompt Tuning | 64 tokens | 0.01 | 50.7 | 75.4 | 53.8 | 37.4 | 19.3 |
| | Prefix-Tuning | 1 token | 10.82 | 45.1 | 75.0 | 45.1 | 32.2 | 13.9 |
| Bias | $\boldsymbol{\beta_\Delta}$,Conv1d | BitFit | 0.02 | 59.9 | 82.3 | 60.8 | 52.9 | 31.3 |
| Linear Projection Matrices | All | LoRA | 0.42 | 58.2 | 74.6 | 58.3 | 51.7 | 40.4 |
| | $\boldsymbol{W}_{\mathrm{in},x}$ | LoRA | 0.14 | 66.7 | 87.9 | 67.7 | 56.9 | 42.8 |
| | $\boldsymbol{W}_{\mathrm{in},z}$ | LoRA | 0.14 | 65.4 | 86.7 | 68.8 | 54.6 | 35.5 |
| | $\boldsymbol{W}_{\mathrm{in},x}, \boldsymbol{W}_{\mathrm{in},z}$ | LoRA | 0.28 | 65.2 | 89.1 | 67.3 | 51.7 | 38.0 |
| | $\boldsymbol{W}_{\mathrm{out}}$ | LoRA | 0.14 | 67.0 | 87.1 | 69.1 | 52.9 | 46.4 |
| S6 | All | Full | 4.44 | 65.7 | 81.9 | 68.8 | 58.0 | 41.0 |
| | | LoRA | 0.38 | 63.9 | 86.3 | 68.2 | 49.4 | 34.3 |
| | $\boldsymbol{A}$ | Full | 0.19 | 56.6 | 77.0 | 58.1 | 46.0 | 33.1 |
| | $\boldsymbol{W}_B, \boldsymbol{W}_C, \boldsymbol{W}_{\Delta,\downarrow}$ | Full | 2.27 | 58.8 | 79.0 | 61.0 | 50.6 | 31.3 |
| | | LoRA | 0.29 | 60.3 | 82.7 | 63.0 | 46.6 | 33.7 |
| | $\boldsymbol{W}_{\Delta,\uparrow}$ | Full | 1.91 | 62.2 | 82.3 | 65.7 | 51.7 | 33.7 |
| | | LoRA | 0.10 | 62.2 | 80.2 | 66.6 | 49.4 | 36.7 |
| | Conv1d | Full | 0.06 | 62.5 | 81.9 | 66.1 | 51.1 | 35.5 |
| Others | $\boldsymbol{D}$,LayerNorm | Full | 0.02 | 51.0 | 71.0 | 51.1 | 42.5 | 29.5 |

(b) Comprehensive experimental results on Spider using Mamba-2.8B.

Table 10: **Full experimental results on Spider (Yu et al., 2018) dataset.** We report the accuracy (↑) for Spider and its subsets. We consider two models in our experiments: Mamba-1.4B and Mamba-2.8B. In each Mamba block, $\boldsymbol{W}_{\mathrm{in},x}$ and $\boldsymbol{W}_{\mathrm{in},z}$ are input projections that preprocess the input for SSM modules and the gating branch, respectively. $\boldsymbol{W}_{\mathrm{out}}$ denotes the output projection after the gating mechanism. $\boldsymbol{W}_B$ and $\boldsymbol{W}_C$ are weight matrices for computing input-dependent $\boldsymbol{B}_n$ and $\boldsymbol{C}_n$. $\boldsymbol{W}_{\Delta,\downarrow}$ and $\boldsymbol{W}_{\Delta,\uparrow}$ represent down and up projections of low-rank weight matrices in the linear layer computing input-dependent step size $\boldsymbol{\Delta}_n$. $\boldsymbol{\beta_\Delta}$ represents the bias in this linear layer. $\boldsymbol{D}$ denotes the weight of residual connections.

| Layer | | Method | # Params (%) | Accuracy |
|---|---|---|---|---|
| Pretrained | | | 0.00 | 0.08 |
| All | All | Full | 100.00 | 59.96 |
| | | LoRA | 1.92 | 60.35 |
| Bias | $\boldsymbol{\beta_\Delta}$, `Conv1d` | BitFit | 0.06 | 44.40 |
| Linear Projection Matrices | All | LoRA | 1.02 | 62.79 |
| | $\boldsymbol{W}_{\text{in},x}$ | LoRA | 0.34 | 53.49 |
| | $\boldsymbol{W}_{\text{in},z}$ | LoRA | 0.34 | 58.15 |
| | $\boldsymbol{W}_{\text{in},x}, \boldsymbol{W}_{\text{in},z}$ | LoRA | 0.68 | 61.04 |
| | $\boldsymbol{W}_{\text{out}}$ | LoRA | 0.34 | 52.04 |
| S6 | All | Full | 4.31 | 55.51 |
| | | LoRA | 0.92 | 43.96 |
| | $\boldsymbol{A}$ | Full | 0.46 | 61.21 |
| | $\boldsymbol{W_B}, \boldsymbol{W_C}, \boldsymbol{W_{\Delta,\downarrow}}$ | Full | 2.28 | 49.51 |
| | | LoRA | 0.69 | 52.27 |
| | $\boldsymbol{W_{\Delta,\uparrow}}$ | Full | 1.40 | 34.54 |
| | | LoRA | 0.23 | 56.49 |
| | `Conv1d` | Full | 0.14 | 55.65 |
| Others | $\boldsymbol{D}$, `LayerNorm` | Full | 0.04 | 58.09 |

Table 11: **Full experimenal results on the CIFAR-10 (Krizhevsky et al., 2009) dataset.** We report accuracy (↑). Mama-130M is utilized in this experiment. In each Mamba block, $\boldsymbol{W}_{\text{in},x}$ and $\boldsymbol{W}_{\text{in},z}$ are input projections that preprocess the input for SSM modules and the gating branch, respectively. $\boldsymbol{W}_{\text{out}}$ denotes the output projection after the gating mechanism. $\boldsymbol{W_B}$ and $\boldsymbol{W_C}$ are weight matrices for computing input-dependent $\boldsymbol{B}_n$ and $\boldsymbol{C}_n$. $\boldsymbol{W_{\Delta,\downarrow}}$ and $\boldsymbol{W_{\Delta,\uparrow}}$ represent down and up projections of low-rank weight matrices in the linear layer computing input-dependent step size $\boldsymbol{\Delta}_n$. $\boldsymbol{\beta_\Delta}$ represents the bias in this linear layer. $\boldsymbol{D}$ denotes the weight of residual connections.

## C.3 LIMITATIONS OF APPLYING PROMPT-BASED METHODS ON SSMS

We provide the formal version of Proposition 1 and its corresponding proof here. We start by introducing the necessary notations. Denote the space of S4 mechanisms with $D$ channels as $\mathcal{F}_{\text{S4},D}$. Let $\boldsymbol{H}_0 = (\boldsymbol{h}_0^{(1)}, \boldsymbol{h}_0^{(2)}, \ldots, \boldsymbol{h}_0^{(D)}) \in \mathbb{R}^{H \times D}$ represent the initial hidden state, and $\boldsymbol{X} = (\boldsymbol{x}_1, \boldsymbol{x}_2, \ldots, \boldsymbol{x}_N) \in \mathbb{R}^{D \times N}$ denote the input sequence. The output of the S4 mechanism is represented as $f(\boldsymbol{X}; \boldsymbol{H}_0)$. Furthermore, for $d$-th channel, let state transition matrix $\overline{\boldsymbol{A}}^{(d)} = \text{diag}\,(a_1^{(d)}, \cdots, a_H^{(d)})$ and input transition vector $\overline{\boldsymbol{B}}^{(d)} = (b_1, \cdots, b_H)^\top$, where $d = 1, \ldots, D$. For any vector $\boldsymbol{v} \in \mathbb{R}^n$, we use $\boldsymbol{v}_{i:j} \in \mathbb{R}^{j-i}$ to denote the subvector of $\boldsymbol{v}$ containing elements from $i \in \mathbb{N}^+$ to $j \in \mathbb{N}^+$, where $i < j$. Similarly, for any matrix $\boldsymbol{M} \in \mathbb{R}^{m \times n}$, we use $\boldsymbol{M}_{i_1:j_1, i_2:j_2}$ to denote the submatrix containing rows $i_1 \in \mathbb{N}^+$ to $j_1 \in \mathbb{N}^+$ and columns $i_2 \in \mathbb{N}^+$ to $j_2 \in \mathbb{N}^+$, where $i_1 < j_1, i_2 < j_2$.

**Proposition 5** (Formal Version of Proposition 1). *Let $f \in \mathcal{F}_{\text{S4},D}$ be an S4 mechanism. Consider prefix-tuning that prepends a sequence $\boldsymbol{P} = (\boldsymbol{p}_1, \ldots, \boldsymbol{p}_M) \in \mathbb{R}^{D \times M}$ to the input sequence $\boldsymbol{X} = (\boldsymbol{x}_1, \boldsymbol{x}_2, \ldots, \boldsymbol{x}_N) \in \mathbb{R}^{D \times N}$. For any prefix $\boldsymbol{P} \in \mathbb{R}^{D \times M}$, there exists an initial hidden state $\boldsymbol{H}_0^\star \in \mathbb{R}^{H \times D}$ such that the output of S4 after prefix-tuning and that after initial state tuning are identical, i.e., $f(\boldsymbol{X}; \boldsymbol{H}_0^\star) \equiv f([\boldsymbol{P}, \boldsymbol{X}]; \boldsymbol{H}_0)_{1:D,M+1:M+N}$ for all $\boldsymbol{X} \in \mathbb{R}^{D \times N}$.*

*Furthermore, assume that $\prod_{0 \le i < j \le H}(a_j^{(d)} - a_i^{(d)}) \ne 0$ and $\prod_{k=1}^H b_k^{(d)} \ne 0$ for all channels $d = 1, \ldots, D$. Then the converse (i.e., for any $\boldsymbol{H}_0 \in \mathbb{R}^{H \times D}$, there exists a $\boldsymbol{P}^\star \in \mathbb{R}^{D \times M}$ such that $f([\boldsymbol{P}^\star, \boldsymbol{X}]; \boldsymbol{H}_0)_{1:D,M+1:M+N} \equiv f(\boldsymbol{X}; \boldsymbol{H}_0^\star)$ for all $\boldsymbol{X} \in \mathbb{R}^{D \times N}$) holds if and only if $M \ge H$.*

*Proof of Proposition 5.* Given that operations in S4 are independent across all channels, we can, without loss of generality, consider the case where the number of channels $D = 1$. Consequently,

we can simplify our notation: the initial hidden states $\boldsymbol{H}_0 \in \mathbb{R}^{H \times D}$ become $\boldsymbol{h}_0 \in \mathbb{R}^H$, the input sequence $\boldsymbol{X} \in \mathbb{R}^{D \times N}$ becomes $\boldsymbol{x} \in \mathbb{R}^N$, and the prefix $\boldsymbol{P} \in \mathbb{R}^{D \times M}$ becomes $\boldsymbol{p} \in \mathbb{R}^M$. We omit the superscript $(d)$ denoting the channel index. To differentiate between the hidden states and output of prefix-tuned S4 (i.e., $f([\boldsymbol{P}, \boldsymbol{X}]; \boldsymbol{H}_0)_{1:D, M+1:M+N}$) and initial state tuned S4 (i.e., $f(\boldsymbol{X}; \boldsymbol{H}_0^\star)$), we introduce superscripts "PT" and "IST" respectively. The "PT" superscript denotes hidden states and output of S4 after prefix-tuning, while "IST" indicates those after initial state tuning.

We divide the proposition into two statements:

1. For any prefix $\boldsymbol{p} \in \mathbb{R}^M$, there exists an initial hidden state $\boldsymbol{h}_0^\star \in \mathbb{R}^H$ such that the output of S4 after prefix-tuning and that after initial state tuning are identical, i.e., $f(\boldsymbol{x}; \boldsymbol{h}_0^\star) \equiv f([\boldsymbol{p}, \boldsymbol{x}]; \boldsymbol{h}_0)_{M+1:N+M}$ for all $\boldsymbol{x} \in \mathbb{R}^N$.

2. Furthermore, assume that $\prod_{0 \leq i < j \leq H}(a_j - a_i) \neq 0$ and $\prod_{k=1}^H b_k \neq 0$. Then the converse (i.e., for any $\boldsymbol{h}_0 \in \mathbb{R}^H$, there exists a $\boldsymbol{p}^\star \in \mathbb{R}^M$ such that $f([\boldsymbol{p}^\star, \boldsymbol{x}]; \boldsymbol{h}_0)_{M+1:N+M} \equiv f(\boldsymbol{x}; \boldsymbol{h}_0^\star)$ for all $\boldsymbol{x} \in \mathbb{R}^N$) holds if and only if $M \geq H$.

We will first prove the first statement and then proceed to prove the second statement.

**Statement 1.** The recurrent computation formulation of S4 in (2) implies that for each position $i$, the output $y_i$ depends solely on the previous hidden state $h_{i-1}$ and the current input $x_i$. Thus, to demonstrate that $f(\boldsymbol{x}; \boldsymbol{h}_0^\star) \equiv f([\boldsymbol{p}, \boldsymbol{x}]; \boldsymbol{h}_0)_{M+1:N+M}$ for all $\boldsymbol{x} \in \mathbb{R}^N$, it suffices to show that the hidden state for predicting output $y_1^{\text{IST}}$ equals that for predicting output $y_{M+1}^{\text{PT}}$, where $y_1^{\text{IST}}$ and $y_{M+1}^{\text{PT}}$ are outputs corresponding to the input $x_1$ for initial state tuning and prefix-tuning, respectively. In other words, it is sufficient to show that the initial state of initial-state-tuned model $\boldsymbol{h}_0^{\text{IST}} = \boldsymbol{h}_0^\star$ is equal to the $(M+1)$-th hidden state of prefix-tuned model $\boldsymbol{h}_{M+1}^{\text{PT}} = \sum_{m=1}^M \overline{\boldsymbol{A}}^{M-m}\overline{\boldsymbol{B}}p_m$. When this equality holds, the subsequent hidden states and outputs for both versions of S4 will be identical, as the input sequence from that point onward is the same. Therefore, We prove the first statement by letting

$$\boldsymbol{h}_0^\star = \sum_{m=1}^M \overline{\boldsymbol{A}}^{M-m}\overline{\boldsymbol{B}}p_m.$$

**Statement 2.** We aim to investigate the conditions under which there exists a $\boldsymbol{h}_0^\star \in \mathbb{R}^H$ such that for any $\boldsymbol{p} \in \mathbb{R}^M$, $f([\boldsymbol{p}^\star, \boldsymbol{x}]; \boldsymbol{h}_0)_{M+1:N+M} \neq f(\boldsymbol{x}; \boldsymbol{h}_0^\star)$. This is equivalent to demonstrating the existence of $\boldsymbol{h}_0^\star \in \mathbb{R}^H$ such that

$$\boldsymbol{h}_0^\star \neq \sum_{m=1}^M \overline{\boldsymbol{A}}^{M-m}\overline{\boldsymbol{B}}p_m, \quad \text{for all } \boldsymbol{p} \in \mathbb{R}^M.$$

This condition can be further reformulated as

$$\mathbb{R}^H \setminus \text{span}(\overline{\boldsymbol{A}}^M\overline{\boldsymbol{B}}, \overline{\boldsymbol{A}}^{M-1}\overline{\boldsymbol{B}}, \ldots, \overline{\boldsymbol{B}}) \neq \emptyset,$$

which is equivalent to

$$\text{span}(\overline{\boldsymbol{A}}^M\overline{\boldsymbol{B}}, \overline{\boldsymbol{A}}^{M-1}\overline{\boldsymbol{B}}, \ldots, \overline{\boldsymbol{B}}) \subsetneq \mathbb{R}^H. \tag{5}$$

To determine when this condition holds, we analyze three distinct cases: (i) $M < H$, (ii) $M = H$, and (iii) $M > H$.

*(Case 1: When $M < H$).* In this scenario, it is obvious that (5) holds. The existence of such a $\boldsymbol{h}_0^\star$ is guaranteed because the dimension of the span is at most $M$, which is strictly less than $H$. This choice of $\boldsymbol{h}_0^\star$ ensures that it cannot be represented as a linear combination of the vectors in the span, thereby establishing the inequality.

*(Case 2: When $M = H$).* In this scenario, $\text{span}(\overline{\boldsymbol{A}}^M\overline{\boldsymbol{B}}, \overline{\boldsymbol{A}}^{M-1}\overline{\boldsymbol{B}}, \ldots, \overline{\boldsymbol{B}}) = \mathbb{R}^H$ if and only if $(\overline{\boldsymbol{A}}^M\overline{\boldsymbol{B}}, \overline{\boldsymbol{A}}^{M-1}\overline{\boldsymbol{B}}, \ldots, \overline{\boldsymbol{B}})$ are linearly independent. Note that

$$\det(\overline{\boldsymbol{A}}^M\overline{\boldsymbol{B}}, \overline{\boldsymbol{A}}^{M-1}\overline{\boldsymbol{B}}, \ldots, \overline{\boldsymbol{B}}) = \det(\overline{\boldsymbol{A}}^M, \overline{\boldsymbol{A}}^{M-1}, \ldots, \boldsymbol{1}) \prod_{k=1}^H b_k, \tag{6}$$

where

$$\det(\overline{\boldsymbol{A}}^M, \overline{\boldsymbol{A}}^{M-1}, \ldots, \boldsymbol{1}) = \det \begin{bmatrix} a_1^{H-1} & \cdots & a_1^2 & a_1 & 1 \\ a_2^{H-1} & \cdots & a_2^2 & a_2 & 1 \\ \vdots & \ddots & \vdots & \vdots & \vdots \\ a_H^{H-1} & \cdots & a_H^2 & a_H & 1 \end{bmatrix} \qquad \text{(Expand)}$$

$$= (-1)^{\frac{H(H-1)}{2}} \prod_{0 \le i < j \le H}^{H} (a_j - a_i). \qquad \text{(Vandermonde matrix)} \quad (7)$$

Combining (6) and (7) yields

$$\det(\overline{\boldsymbol{A}}^M \overline{\boldsymbol{B}}, \overline{\boldsymbol{A}}^{M-1} \overline{\boldsymbol{B}}, \ldots, \overline{\boldsymbol{B}}) = (-1)^{\frac{H(H-1)}{2}} \prod_{0 \le i < j \le H}^{H} (a_j - a_i) \prod_{k=1}^{H} b_k.$$

Therefore, if and only if $\prod_{1 \le i < j \le H}(a_j - a_i) \ne 0$ and $\prod_{k=1}^H b_k \ne 0$, we have

$$\det(\overline{\boldsymbol{A}}^M \overline{\boldsymbol{B}}, \overline{\boldsymbol{A}}^{M-1} \overline{\boldsymbol{B}}, \ldots, \overline{\boldsymbol{B}}) \ne 0,$$

which is both necessary and sufficient for the linear independence of $(\overline{\boldsymbol{A}}^M \overline{\boldsymbol{B}}, \overline{\boldsymbol{A}}^{M-1} \overline{\boldsymbol{B}}, \ldots, \overline{\boldsymbol{B}})$, and consequently, for the condition in (5) to be satisfied.

*(Case 3: When $M > H$).* The analysis presented in case 2 extends naturally to this scenario.

The combination of the three cases above completes the proof of statement 2. $\qquad\square$

## C.4 Optimal Application of LoRA in SSM-based Models

Several studies (Hu et al., 2023; He et al., 2021) present findings on Transformers, indicating that applying LoRA to linear projection matrices yields performance comparable to or marginally superior to that of attention layers. In contrast, our experimental results on SSMs reveal that applying LoRA to linear projection matrices is more effective than applying it to S6 (see Table 3). To elucidate this phenomenon, we examine the influence of updating linear projection matrices on the model's output.

**Notations.** For the feasibility of the analysis, we consider a simplified SSM-based architecture which only consists of the input projection matrix $\boldsymbol{W}_{\text{in}} \in \mathbb{R}^{D \times D}$ and the S6 module parameterized by diagonal state transition matrices $\{\boldsymbol{A}^{(d)}\}_{d=1}^D$ with $\boldsymbol{A}^{(d)} \in \mathbb{R}^{H \times H}$, the weight matrices $\boldsymbol{W}_B, \boldsymbol{W}_C \in \mathbb{R}^{H \times D}$ for computing input-dependent input transition vectors $\boldsymbol{B}_n \in \mathbb{R}^H$ and output mapping vectors $\boldsymbol{C}_n \in \mathbb{R}^H$, the down and up projection matrices $\boldsymbol{W}_{\boldsymbol{\Delta},\downarrow} \in \mathbb{R}^{D \times R}, \boldsymbol{W}_{\boldsymbol{\Delta},\uparrow} \in \mathbb{R}^{R \times D}$ (where $R$ is the rank) for low-rank weight matrices for computing the input-depdenent step size $\boldsymbol{\Delta}_n = (\Delta_n^{(1)}, \ldots, \Delta_n^{(D)}) \in \mathbb{R}_D$, for $n = 1, \ldots, N$. Define $\boldsymbol{W}_{\text{S6}} = [\boldsymbol{W}_B^\top, \boldsymbol{W}_C^\top, \boldsymbol{W}_{\boldsymbol{\Delta},\uparrow}^\top]^\top \in \mathbb{R}^{(2H+R) \times D}$. In the Mamba implementation, $\boldsymbol{W}_{\text{S6}}$ is implemented as the weight matrix of a single linear layer, referred to as x_proj in the codebase. Therefore, the parameters of the S6 can be formulated as

$$\boldsymbol{\theta}(\cdot; \{\boldsymbol{A}\}_{d=1}^D, \boldsymbol{W}_{\text{S6}}, \boldsymbol{W}_{\boldsymbol{\Delta},\downarrow}, \boldsymbol{W}_{\text{in}}) = \{\overline{\boldsymbol{A}}_n, \overline{\boldsymbol{B}}_n, \boldsymbol{C}_n\}_{n=1}^N.$$

Consider input sequence $\boldsymbol{X} = (\boldsymbol{x}_1, \ldots, \boldsymbol{x}_N) \in \mathbb{R}^{D \times N}$. Let $\boldsymbol{Z} = (\boldsymbol{z}_1, \ldots, \boldsymbol{z}_N) \in \mathbb{R}^{D \times N}$ denote the intermediate output after the input projection. The intermediate output at position $n \in \{1, \ldots, N\}$ is

$$\boldsymbol{z}_n = \boldsymbol{W}_{\text{in}} \boldsymbol{x}_n. \qquad (8)$$

Note that

$$\boldsymbol{B}_n = \boldsymbol{W}_B \boldsymbol{z}_n, \quad \boldsymbol{C}_n = \boldsymbol{W}_C \boldsymbol{z}_n, \quad \boldsymbol{\Delta}_n = \text{softplus}(\boldsymbol{W}_{\boldsymbol{\Delta},\uparrow} \boldsymbol{W}_{\boldsymbol{\Delta},\downarrow} \boldsymbol{z}_n + \boldsymbol{\beta}_{\boldsymbol{\Delta}}), \qquad (9)$$

and after discretization, we have

$$\overline{\boldsymbol{A}}_n^{(d)} = \exp(\Delta_n^{(d)} \boldsymbol{A}^{(d)}), \quad \overline{\boldsymbol{B}}_n = \Delta_n^{(d)} \boldsymbol{B}_n = \Delta_n^{(d)} \boldsymbol{W}_B \boldsymbol{z}_n. \qquad (10)$$

Combining (8), (9) and (10) yields

$$\boldsymbol{\theta}(\boldsymbol{X}; \{\boldsymbol{A}\}_{d=1}^D, \boldsymbol{W}_{\text{S6}}, \boldsymbol{W}_{\boldsymbol{\Delta},\downarrow}, \boldsymbol{W}_{\text{in}}) = \{\overline{\boldsymbol{A}}_n, \overline{\boldsymbol{B}}_n, \boldsymbol{C}_n\}_{n=1}^N, \text{ where} \qquad (11)$$

$$\overline{\boldsymbol{A}}_n^{(d)} = \exp(\Delta_n^{(d)} \boldsymbol{A}^{(d)}), \quad \overline{\boldsymbol{B}}_n = \Delta_n^{(d)} \boldsymbol{W}_B \boldsymbol{W}_{\text{in}} \boldsymbol{x}_n, \quad \boldsymbol{C}_n = \boldsymbol{W}_C \boldsymbol{W}_{\text{in}} \boldsymbol{x}_n,$$
$$\boldsymbol{\Delta}_n = \text{softplus}(\boldsymbol{W}_{\boldsymbol{\Delta},\downarrow} \boldsymbol{W}_{\boldsymbol{\Delta},\uparrow} \boldsymbol{W}_{\text{in}} \boldsymbol{x}_n + \boldsymbol{\beta}_{\boldsymbol{\Delta}}).$$

**Theoretical Analysis.** In the following theorem, we demonstrate that applying LoRA exclusively to $\boldsymbol{W}_{\text{in}}$ is equivalent to applying it to $\boldsymbol{W}_{\text{S6}}$.

**Lemma 6** (Detailed Version of Lemma 2). *Consider a model consists of an S6 module augmented with a linear input projection $\boldsymbol{W}_{in} \in \mathbb{R}^{D \times D}$. For any fine-tuned model where only $\boldsymbol{W}_{S6}$ is updated to $\overline{\boldsymbol{W}}_{S6}$, there exists $\widehat{\boldsymbol{W}}_{in}$ such that updating only $\boldsymbol{W}_{in}$ to $\widehat{\boldsymbol{W}}_{in}$ yields:*

$$\boldsymbol{\theta}(\boldsymbol{X}; \{\boldsymbol{A}^{(d)}\}_{d=1}^{D}, \overline{\boldsymbol{W}}_{S6}, \boldsymbol{W}_{\boldsymbol{\Delta}, \downarrow}, \boldsymbol{W}_{in}) = \boldsymbol{\theta}(\boldsymbol{X}; \{\boldsymbol{A}^{(d)}\}_{d=1}^{D}, \boldsymbol{W}_{S6}, \boldsymbol{W}_{\boldsymbol{\Delta}, \downarrow}, \widehat{\boldsymbol{W}}_{in}) \tag{12}$$

*Proof of Lemma 6.* In this proof, we use $\bar{\cdot}$ to denote the corresponding notations for the model with only $\boldsymbol{W}_{S6}$ updated, and use $\widehat{\cdot}$ to denote the corresponding notations for the model with only $\boldsymbol{W}_{\text{in}}$ updated.

To demonstrate (12), it is sufficient, according to (11), to find $\widehat{\boldsymbol{W}}_{\text{in}}$ that satisfies the following equations:

$$\overline{\boldsymbol{W}}_{\boldsymbol{C}} \boldsymbol{W}_{\text{in}} = \boldsymbol{W}_{\boldsymbol{C}} \widehat{\boldsymbol{W}}_{\text{in}} \tag{13}$$
$$\overline{\boldsymbol{W}}_{\boldsymbol{\Delta}, \uparrow} \boldsymbol{W}_{\text{in}} = \boldsymbol{W}_{\boldsymbol{\Delta}, \uparrow} \widehat{\boldsymbol{W}}_{\text{in}}$$
$$\overline{\boldsymbol{W}}_{\boldsymbol{B}} \boldsymbol{W}_{\text{in}} = \boldsymbol{W}_{\boldsymbol{B}} \widehat{\boldsymbol{W}}_{\text{in}}.$$

Since $\boldsymbol{W}_{\text{S6}} = \begin{bmatrix} \boldsymbol{W}_{\boldsymbol{B}} \\ \boldsymbol{W}_{\boldsymbol{C}} \\ \boldsymbol{W}_{\boldsymbol{\Delta}, \uparrow} \end{bmatrix}$, the three conditions (13) can be written as

$$\overline{\boldsymbol{W}}_{\text{S6}} \boldsymbol{W}_{\text{in}} = \boldsymbol{W}_{\text{S6}} \widehat{\boldsymbol{W}}_{\text{in}}. \tag{14}$$

By applying Singular Value Decomposition (SVD) to $\boldsymbol{W}_{\text{S6}}$ and $(\boldsymbol{W}_{\text{S6}} - \overline{\boldsymbol{W}}_{\text{S6}}) \boldsymbol{W}_{\text{in}}$, we obtain:

$$\boldsymbol{W}_{\text{S6}} = \boldsymbol{U} \begin{bmatrix} \boldsymbol{\Sigma} & \boldsymbol{O}_{(2H+R) \times (D-2H-R)} \end{bmatrix} \boldsymbol{V}^{\top}, \tag{15}$$
$$(\boldsymbol{W}_{\text{S6}} - \overline{\boldsymbol{W}}_{\text{S6}}) \boldsymbol{W}_{\text{in}} = \boldsymbol{U}' \begin{bmatrix} \boldsymbol{\Sigma}' & \boldsymbol{O}_{(2H+R) \times (D-2H-R)} \end{bmatrix} \boldsymbol{V}'^{\top},$$

where $\boldsymbol{U}, \boldsymbol{U}' \in \mathbb{R}^{(2H+R) \times (2H+R)}$, $\boldsymbol{\Sigma}, \boldsymbol{\Sigma}' \in \mathbb{R}^{(2H+R) \times (2H+R)}$, and $\boldsymbol{V}, \boldsymbol{V}' \in \mathbb{R}^{D \times D}$. The diagonal elements of $\boldsymbol{\Sigma}$ and $\boldsymbol{\Sigma}'$ are in decreasing order.

We let

$$\widehat{\boldsymbol{W}}_{\text{in}} = \boldsymbol{V} \begin{bmatrix} \boldsymbol{\Sigma}^{-1} \boldsymbol{U}^{\top} \overline{\boldsymbol{W}}_{\text{S6}} \boldsymbol{W}_{\text{in}} \\ \boldsymbol{Q} \end{bmatrix}, \tag{16}$$

where $\boldsymbol{Q} \in \mathbb{R}^{(D-2H-R) \times D}$ is an arbitrary matrix to be determined later. Plugging (15) and (16) back to $\boldsymbol{W}_{\text{S6}} \widehat{\boldsymbol{W}}_{\text{in}}$ and simplifying results in

$$\boldsymbol{W}_{\text{S6}} \widehat{\boldsymbol{W}}_{\text{in}}$$
$$= \boldsymbol{U} \begin{bmatrix} \boldsymbol{\Sigma} & \boldsymbol{O}_{(2H+R) \times (D-2H-R)} \end{bmatrix} \boldsymbol{V}^{\top} \boldsymbol{V} \begin{bmatrix} \boldsymbol{\Sigma}^{-1} \boldsymbol{U}^{\top} \overline{\boldsymbol{W}}_{\text{S6}} \boldsymbol{W}_{\text{in}} \\ \boldsymbol{Q} \end{bmatrix} \qquad ((15) \ \& \ (16))$$
$$= \overline{\boldsymbol{W}}_{\text{S6}} \boldsymbol{W}_{\text{in}}, \qquad\qquad\qquad (\text{Simplifying})$$

which demonstrates that (14) is satisfied and completes the proof. $\square$

## C.5 BENCHMARKING LoRA ON JAMBA AND MAMBA-II

In this section, we expand our analysis beyond the deep S4 model and Mamba. Specifically, we incorporate the Transformer-SSM hybrid model Jamba (Lieber et al., 2024) (Jamba-Tiny-319M) and Mamba-II (Dao & Gu, 2024) (Mamba-II-130M and Mamba-II-1.3B).

**Benchmarking LoRA Across Different Layers of Jamba.** Table 12 presents the benchmark results of LoRA and full fine-tuning across different layers of Jamba. Our findings demonstrate that, on Jamba, LoRA is more effective on linear projection layers than on SSM modules, which aligns with our conclusion on Mamba.

| Layer | | Method | # Params (%) | METEOR | BLEU |
|---|---|---|---|---|---|
| All | All | Full | 100.00 | 70.8 | 45.0 |
| Attention | All | LoRA | 0.02 | 63.5 | 19.7 |
| MLP | All | LoRA | 1.37 | 70.9 | 46.2 |
| Linear Projection Matrices + S6 | All | LoRA | 0.31 | 70.2 | 40.0 |
| Linear Projection Matrices | $W_{in}$ | LoRA | 0.11 | 68.9 | 37.8 |
| | $W_{out}$ | LoRA | 0.05 | 67.7 | 31.9 |
| S6 | All | Full | 0.54 | 69.2 | 35.5 |
| | $W_B, W_C, W_{\Delta,\downarrow}$ | LoRA | 0.15 | 66.6 | 24.2 |

Table 12: Benchmark results of LoRA on DART (Nan et al., 2021) dataset using Jamba-Tiny-319M (Lieber et al., 2024).

**Benchmarking LoRA Across Different Layers of Mamba-II.** Tables 13 to 15 present the benchmark results of LoRA and full fine-tuning across different layers of Mamba-II. We follow the same experimental setup used for Mamba-I and demonstrate that, on Mamba-II, our conclusion holds: LoRA is more effective on linear projection layers than on SSM modules.

| | Model | | Mamba-II-130M | | Mamba-II-1.3B |
|---|---|---|---|---|---|
| Target Layers | Dataset Metric (↑) | Params (%) | DART METEOR | BLEU | Spider Acc. |
| SSM Modules | LoRA | < 1.0 | 64.2 | 40.1 | 54.1 |
| Linear Layers | LoRA | | 67.1 | 43.0 | 57.9 |
| Both | LoRA | < 3.0 | 66.9 | 45.4 | 64.5 |

Table 13: Summary of benchmark results of LoRA on Mamba-II.

| Layer | | Method | # Params (%) | METEOR | BLEU |
|---|---|---|---|---|---|
| All | All | Full | 100.00 | 66.6 | 34.9 |
| | | LoRA | 1.39 | 66.9 | 45.4 |
| Linear Projection Matrices | $W_{in}, W_{out}$ | LoRA | 1.02 | 67.1 | 44.7 |
| | $W_{in}$ | LoRA | 0.68 | 67.1 | 43.0 |
| | $W_{out}$ | LoRA | 0.34 | 66.8 | 42.3 |
| S6 | All | Full | 4.17 | 65.7 | 39.7 |
| | | LoRA | 0.38 | 64.2 | 40.1 |
| | $W_B, W_C, W_{\Delta}$ | Full | 4.00 | 66.0 | 36.2 |
| | | LoRA | 0.38 | 64.8 | 39.5 |

Table 14: Full benchmark results of LoRA on DART (Nan et al., 2021) dataset using Mamba-II 130M (Dao & Gu, 2024).

## C.6 BENCHMARKING DoRA ON MAMBA

To provide a more comprehensive analysis, we included evaluations of DoRA (yang Liu et al., 2024), an advanced variant of LoRA. We evaluate the performance of DoRA on the DART dataset using Mamba-130M and on the Spider dataset using Mamba-1.4B. The results are summarized in Tables 16 to 18. Our findings are consistent with observations seen in LoRA: applying DoRA to linear projection matrices proves more effective than its application to SSM modules. Interestingly, applying DoRA to SSM modules not only offers limited benefits but, in some cases, even degrades performance. This is particularly evident on the Spider dataset, when comparing the configurations of applying DoRA to both linear projection matrices and SSM modules versus solely targeting linear

| Layer | | Method | # Params (%) | All | Easy | Medium | Hard | Extra |
|---|---|---|---|---|---|---|---|---|
| All | All | Full | 100.00 | 64.8 | 85.9 | 65.7 | 54.0 | 42.2 |
| | | LoRA | 0.71 | 64.5 | 81.0 | 66.4 | 56.9 | 42.8 |
| Linear Projection Matrices | $W_\text{in}, W_\text{out}$ | LoRA | 0.52 | 50.4 | 68.5 | 52.0 | 44.8 | 24.7 |
| | $W_\text{in}$ | LoRA | 0.35 | 57.5 | 76.2 | 59.4 | 48.9 | 33.7 |
| | $W_\text{out}$ | LoRA | 0.18 | 57.9 | 81.0 | 56.7 | 51.7 | 33.1 |
| S6 | All | Full | 2.42 | 55.1 | 76.2 | 56.1 | 42.5 | 34.3 |
| | | LoRA | 0.18 | 54.1 | 74.2 | 58.1 | 46.0 | 21.7 |
| | $A_\text{log}$ | Full | 0.00 | 21.5 | 46.0 | 18.8 | 11.5 | 2.4 |
| | $W_B, W_C, W_\Delta$ | Full | 2.34 | 50.3 | 73.0 | 52.2 | 39.7 | 22.3 |
| | | LoRA | 0.18 | 55.5 | 77.4 | 55.2 | 46.6 | 33.1 |

Table 15: Full benchmark results of LoRA on Spider (Yu et al., 2018) dataset using Mamba-II 1.3B (Dao & Gu, 2024).

projection matrices. Furthermore, we observe slightly better results on the smaller Mamba-130M with DoRA, while for Mamba-1.4B, LoRA performs better.

| | Model | | Mamba-130M | | Mamba-1.4B |
|---|---|---|---|---|---|
| Target Layers | Dataset Metric (↑) | Params (%) | DART | | Spider |
| | | | METEOR | BLEU | Acc. |
| SSM Modules | LoRA | < 0.4 | 68.86 | 47.05 | 58.03 |
| | DoRA | | 68.79 | 47.07 | 55.32 |
| Linear Layers | LoRA | < 0.4 | 70.25 | 48.86 | 61.80 |
| | DoRA | | 70.81 | 49.93 | 61.32 |
| Both | LoRA | < 3.0 | 70.97 | 49.52 | 56.38 |
| | DoRA | | 70.94 | 51.36 | 55.71 |

Table 16: Summary of benchmark results of DoRA on Mamba.

| Layer | | Method | # Params (%) | All | Easy | Medium | Hard | Extra |
|---|---|---|---|---|---|---|---|---|
| All | All | DoRA | 1.02 | 55.7 | 77.0 | 57.0 | 47.1 | 29.5 |
| Linear Projection Matrices | All | DoRA | 0.55 | 57.2 | 79.4 | 58.7 | 46.0 | 31.3 |
| | $W_{\text{in},x}$ | DoRA | 0.19 | 58.4 | 80.2 | 60.1 | 49.4 | 30.7 |
| | $W_{\text{in},z}$ | DoRA | 0.19 | 59.8 | 83.9 | 60.1 | 50.6 | 32.5 |
| | $W_\text{in}$ | DoRA | 0.37 | 60.7 | 78.6 | 62.1 | 52.9 | 38.6 |
| | $W_\text{out}$ | DoRA | 0.18 | 61.3 | 79.4 | 63.9 | 50.0 | 39.2 |
| S6 | $W_B, W_C, W_{\Delta,\downarrow}$ | DoRA | 0.48 | 58.9 | 77.4 | 62.1 | 47.1 | 34.9 |
| | $A$ | DoRA | 0.13 | 50.5 | 72.6 | 51.1 | 44.3 | 22.3 |
| | $W_B, W_C$ | DoRA | 0.35 | 55.3 | 78.2 | 57.8 | 41.4 | 28.9 |
| | $W_{\Delta,\uparrow}$ | DoRA | 0.13 | 55.3 | 76.2 | 59.2 | 42.5 | 27.1 |

Table 17: Full benchmark results of DoRA on DART (Nan et al., 2021) dataset using Mamba-130M.

# D    DETAILS OF SEC. 5: SDLoRA

## D.1    UNDERSTANDING THE ROLES OF STATE MATRIX $A$, INPUT TRANSITION VECTOR $B$, AND OUTPUT MAPPING VECTOR $C$ FOR A SINGLE CHANNEL IN S4 MODULES

**Problem Setting.** Inspired by Zeng & Lee (2024)'s theoretical analysis of LoRA's expressive power, we adopt a similar framework to explore the expressive potential of various parameters in the S4 model. Specifically, we assume a target model that performs well on the intended task and a frozen model, which may be either pretrained or randomly initialized. Our goal is to identify a

| Layer | | Method | # Params (%) | METEOR | BLEU |
|---|---|---|---|---|---|
| All | All | DoRA | 2.02 | 70.9 | 51.4 |
| Linear Projection Matrices | All | DoRA | 1.09 | 71.2 | 50.8 |
| | $\boldsymbol{W}_{\text{in},x}$ | DoRA | 0.37 | 70.8 | 49.9 |
| | $\boldsymbol{W}_{\text{in},z}$ | DoRA | 0.37 | 70.2 | 48.3 |
| | $\boldsymbol{W}_{\text{in}}$ | DoRA | 0.74 | 70.7 | 51.6 |
| | $\boldsymbol{W}_{\text{out}}$ | DoRA | 0.36 | 70.7 | 46.0 |
| S6 | $\boldsymbol{W}_B, \boldsymbol{W}_C, \boldsymbol{W}_{\Delta,\downarrow}$ | DoRA | 0.95 | 70.2 | 50.0 |
| | $\boldsymbol{A}$ | DoRA | 0.26 | 68.8 | 47.1 |
| | $\boldsymbol{W}_B, \boldsymbol{W}_C$ | DoRA | 0.69 | 68.3 | 47.3 |
| | $\boldsymbol{W}_{\Delta,\uparrow}$ | DoRA | 0.26 | 68.4 | 46.3 |

Table 18: Full benchmark results of DoRA on Spider (Yu et al., 2018) dataset using Mamba-1.4B.

parameter-efficient method to update the frozen model so that it becomes functionally equivalent to the target model. In alignment with Zeng & Lee (2024), we assume that the frozen model's capacity is equal to or exceeds that of the target model. This assumption is based on two main considerations: (i) analytical tractability, which necessitates that the frozen model must have the potential to match the functionality of the target model, and (ii) a practical rationale, given that the models typically used in practice are often overparameterized. Assume that both the target model and the frozen model are S4, with the target model having a hidden state dimension $H_\star$ and the frozen model having a hidden state dimension $H \geq H_\star$. Meanwhile, suppose that all the hidden dimensions of both models are valid, meaning that none of the parameter elements are zero. The target model, frozen model, and the updated model after tuning the parameters on the frozen model can be formulated using discretized parameters $\overline{\boldsymbol{A}}, \overline{\boldsymbol{B}}, \boldsymbol{C}$ as follows:

$$\text{(Target model)} \quad f^\star(\boldsymbol{x})_n = \sum_{m=1}^{n} \boldsymbol{C}_\star \overline{\boldsymbol{A}}_\star^{m-n} \overline{\boldsymbol{B}}_\star x_m, \text{ where } \operatorname{diag}(\overline{\boldsymbol{A}}_\star), \overline{\boldsymbol{B}}_\star, \boldsymbol{C}_\star \in \mathbb{R}^{H_\star},$$

$$\text{(Frozen model)} \quad f_0(\boldsymbol{x})_n = \sum_{m=1}^{n} \boldsymbol{C} \overline{\boldsymbol{A}}^{m-n} \overline{\boldsymbol{B}} x_m, \text{ where } \operatorname{diag}(\overline{\boldsymbol{A}}), \overline{\boldsymbol{B}}, \boldsymbol{C} \in \mathbb{R}^{H},$$

$$\text{(Updated model)} \quad \hat{f}(\boldsymbol{x})_n = \sum_{m=1}^{n} \widehat{\boldsymbol{C}} \widehat{\overline{\boldsymbol{A}}}^{m-n} \widehat{\overline{\boldsymbol{B}}} x_m, \text{ where } \operatorname{diag}(\widehat{\overline{\boldsymbol{A}}}), \widehat{\overline{\boldsymbol{B}}}, \widehat{\boldsymbol{C}} \in \mathbb{R}^{H}.$$

**Parameter Efficiency Analysis on S4.** Let $\mathcal{P}^H$ denote the set of all $H \times H$ permutation matrices. Given this formulation, we present our first analysis of parameter efficiency for the S4 model in the following lemma. This analysis is based on the parameters after necessary discretization $(\overline{\boldsymbol{A}}, \overline{\boldsymbol{B}}, \boldsymbol{C})$.

**Lemma 3** (Essential Discretized Parameter Set for S4). *Consider the parameters after discretization, i.e., $\overline{\boldsymbol{A}}, \overline{\boldsymbol{B}}, \boldsymbol{C}$. To achieve functional equivalence between the updated model and the target model, i.e., $\hat{f} \equiv f^\star$, it is sufficient to tune the following number of parameters:*

$$\min_{\boldsymbol{P} \in \mathcal{P}^H} \underbrace{\left\| \left[ \boldsymbol{P}^\top (\operatorname{diag}(\overline{\boldsymbol{A}}) \otimes \overline{\boldsymbol{B}} \otimes \boldsymbol{C}^\top) \right]_{(H_\star+1):H} \right\|_0}_{\text{eliminating redundant dimensions}} + \underbrace{\left\| \left[ \boldsymbol{P}^\top \overline{\boldsymbol{A}} \boldsymbol{P} \right]_{1:H_\star, 1:H_\star} - \overline{\boldsymbol{A}}_\star \right\|_0}_{\text{aligning the state matrix}} + \underbrace{\left\| \left[ \boldsymbol{P}^\top (\overline{\boldsymbol{B}} \otimes \boldsymbol{C}^\top) \right]_{1:H_\star} - \overline{\boldsymbol{B}}_\star \otimes \boldsymbol{C}_\star^\top \right\|_0}_{\text{aligning input-output interactions}}.$$

$$\overbrace{\phantom{xxxxxxxxxxxxxxxxxxxxxxxxxxxxxxxxxxx}}^{\text{aligning used dimensions with target model}}$$

*Proof of Lemma 3.* The key idea of this proof is straightforward. To facilitate the analysis and update the frozen model to be equivalent to the target model, we first equalize the number of hidden state dimensions between the two models. This is achieved by expanding the target model's $\boldsymbol{A}_\star$, $\boldsymbol{B}_\star$, and $\boldsymbol{C}_\star$ to match the $H$ hidden state dimensions of the frozen model, padding the additional $H - H^\star$ dimensions with zeros.

Define $\otimes$ as the element-wise product. We can express the target model as:

$$f^\star(\boldsymbol{x})_n = \sum_{m=1}^{n} \begin{bmatrix} \boldsymbol{C}_\star & \boldsymbol{0}^\top \end{bmatrix} \begin{bmatrix} \overline{\boldsymbol{A}}_\star & \boldsymbol{O} \\ \boldsymbol{O} & \boldsymbol{O} \end{bmatrix}^{n-m} \begin{bmatrix} \overline{\boldsymbol{B}}_\star \\ \boldsymbol{0} \end{bmatrix} x_m$$

$$= \sum_{m=1}^{n} \text{diag}\left(\begin{bmatrix} \overline{\boldsymbol{A}}_\star & \boldsymbol{O} \\ \boldsymbol{O} & \boldsymbol{O} \end{bmatrix}\right)^{n-m} \left(\begin{bmatrix} \boldsymbol{C}_\star^\top \\ \boldsymbol{0} \end{bmatrix} \otimes \begin{bmatrix} \overline{\boldsymbol{B}}_\star \\ \boldsymbol{0} \end{bmatrix}\right) x_m$$

Consider any permutation matrix $\boldsymbol{P} \in \mathcal{P}^H$. Applying $\boldsymbol{P}$ to permute the frozen model leaves the model functionally unchanged:

$$f_0(\boldsymbol{x})_n = \sum_{m=1}^{n} \boldsymbol{C} \overline{\boldsymbol{A}}^{n-m} \overline{\boldsymbol{B}} x_m = \sum_{m=1}^{n} \boldsymbol{C}\boldsymbol{P} \left(\boldsymbol{P}^\top \overline{\boldsymbol{A}} \boldsymbol{P}\right)^{n-m} \boldsymbol{P}^\top \overline{\boldsymbol{B}} x_m$$

$$= \sum_{m=1}^{n} \text{diag}\left(\boldsymbol{P}^\top \overline{\boldsymbol{A}} \boldsymbol{P}\right)^{n-m} \left((\boldsymbol{P}^\top \boldsymbol{C}^\top) \otimes (\boldsymbol{P}^\top \overline{\boldsymbol{B}})\right) x_m$$

Therefore, to make the updated model equivalent to the target model, we need to update $\boldsymbol{P}^\top \overline{\boldsymbol{A}} \boldsymbol{P}$ to align with $\begin{bmatrix} \overline{\boldsymbol{A}}_\star & \boldsymbol{O} \\ \boldsymbol{O} & \boldsymbol{O} \end{bmatrix}$, and $(\boldsymbol{P}^\top \boldsymbol{C}^\top) \otimes (\boldsymbol{P}^\top \overline{\boldsymbol{B}})$ to align with $\begin{bmatrix} \boldsymbol{C}_\star^\top \\ \boldsymbol{0} \end{bmatrix} \otimes \begin{bmatrix} \overline{\boldsymbol{B}}_\star \\ \boldsymbol{0} \end{bmatrix}$. If they are already matching or partially matched for certain entries, no updates are required for those entries; only the unmatched entries need to be updated. Then, the required trainable parameters for this permutation matrix $\boldsymbol{P}$ are:

$$\left\| \left[\boldsymbol{P}^\top (\text{diag}(\overline{\boldsymbol{A}}) \otimes \overline{\boldsymbol{B}} \otimes \boldsymbol{C}^\top)\right]_{(H_\star+1):H} \right\|_0 + \left\| \left[\boldsymbol{P}^\top \overline{\boldsymbol{A}} \boldsymbol{P}\right]_{1:H_\star,1:H_\star} - \overline{\boldsymbol{A}}_\star \right\|_0 + \left\| \left[\boldsymbol{P}^\top (\overline{\boldsymbol{B}} \otimes \boldsymbol{C}^\top)\right]_{1:H_\star} - \overline{\boldsymbol{B}}_\star \otimes \boldsymbol{C}_\star^\top \right\|_0.$$

Optimizing the permutation matrix $\boldsymbol{P} \in \mathcal{P}^H$ yields the desired results. $\qquad\square$

This lemma highlights the significance of identifying essential hidden state dimensions. The term $\left\| \left[\boldsymbol{P}^\top (\text{diag}(\overline{\boldsymbol{A}}) \otimes \overline{\boldsymbol{B}} \otimes \boldsymbol{C}^\top)\right]_{(H_\star+1):H} \right\|_0$ underscores the importance of excluding redundant dimensions. This can be achieved by either directly removing these dimensions from the state matrix $\overline{\boldsymbol{A}}$, or by updating $\overline{\boldsymbol{B}}$ or $\boldsymbol{C}$ to ensure that only the selected hidden state dimensions are utilized during the input transition or output mapping phases. Once redundant dimensions are filtered out, tuning only the essential dimensions is sufficient to align the updated model with the target model.

Furthermore, based on the lemma, the roles of the input transition vector $\overline{\boldsymbol{B}}$ and $\boldsymbol{C}^\top$ are nearly identical, as they consistently appear together as the combined term $\overline{\boldsymbol{B}} \otimes \boldsymbol{C}^\top$, which is also discussed in Gupta et al. (2022). Consequently, one could opt to tune either $\overline{\boldsymbol{B}}$ or $\boldsymbol{C}$ exclusively or alternatively, split the indices into two groups, tuning $\overline{\boldsymbol{B}}$ for the first group and $\boldsymbol{C}$ for the second. Both vectors indicate how information from different hidden state dimensions is integrated, whereas $\overline{\boldsymbol{A}}$ plays a distinct role, determining how the hidden states are stored.

In practice, instead of directly using the discretized parameters $\overline{\boldsymbol{A}}, \overline{\boldsymbol{B}}, \boldsymbol{C}$, S4 is implemented using the continuous parameters $\boldsymbol{A}, \boldsymbol{B}, \boldsymbol{C}$ with step size $\Delta$. To provide further practical guidance on parameter tuning, the following two lemmas analyze the parameter efficiency of continuous parameters under different discretization methods: Two exemplary methods of discretization are bilinear and zero-order hold (ZOH):

$$\text{(Bilinear)} \begin{cases} \overline{\boldsymbol{A}} = (\boldsymbol{I} - \Delta/2\boldsymbol{A})^{-1}(\boldsymbol{I} + \Delta/2\boldsymbol{A}) \\ \overline{\boldsymbol{B}} = (\boldsymbol{I} - \Delta/2\boldsymbol{A})^{-1} \cdot \Delta\boldsymbol{B}, \end{cases} \quad \text{(ZOH)} \begin{cases} \overline{\boldsymbol{A}} = \exp(\Delta\boldsymbol{A}) \\ \overline{\boldsymbol{B}} = (\Delta\boldsymbol{A})^{-1}(\exp(\Delta\boldsymbol{A}) - \boldsymbol{I}) \cdot \Delta\boldsymbol{B}. \end{cases} \tag{17}$$

**Lemma 7** (Essential Continuous Parameter Set for S4 with Bilinear Discritization). *Consider the parameters before discretization, i.e., $\boldsymbol{A}, \boldsymbol{B}, \boldsymbol{C}$, and they are discretized via bilinear discretization. To achieve functional equivalence between the updated model and the target model, i.e., $\hat{f} \equiv f^\star$, it is sufficient to tune the following number of parameters:*

$$\min_{\boldsymbol{P} \in \mathcal{P}^H} \underbrace{\left\| \left[\Delta\boldsymbol{P}^\top (\text{diag}(\boldsymbol{I} + \Delta/2\boldsymbol{A}) \otimes \boldsymbol{B} \otimes \boldsymbol{C}^\top)\right]_{(H_\star+1):H} \right\|_0}_{\text{eliminating redundant dimensions}} + \underbrace{\left\| \left[\boldsymbol{P}^\top \boldsymbol{A} \boldsymbol{P}\right]_{1:H_\star,1:H_\star} - \boldsymbol{A}_\star \right\|_0}_{\text{aligning the state matrix}} + \underbrace{\left\| \left[\boldsymbol{P}^\top (\boldsymbol{B} \otimes \boldsymbol{C}^\top)\right]_{1:H_\star} - \boldsymbol{B}_\star \otimes \boldsymbol{C}_\star^\top \right\|_0}_{\text{aligning input-output interactions}}.$$

*Proof of Lemma 7.* Combining Lemma 3 and the Bilinear discretization method in (17) yields the desired results. $\qquad\square$

**Lemma 8** (Essential Continuous Parameter Set for S4 with ZOH Discritization). *Consider the parameters before discretization, i.e., $\boldsymbol{A}, \boldsymbol{B}, \boldsymbol{C}$, and they are discretized via ZOH discretization. To achieve functional equivalence between the updated model and the target model, i.e., $\hat{f} \equiv f^{\star}$, it is sufficient to tune the following number of parameters:*

$$\min_{\boldsymbol{P} \in \mathcal{P}^H} \underbrace{\left\| \left[ \Delta \boldsymbol{P}^{\top} (\operatorname{diag}(\exp(\Delta \boldsymbol{A}) - \boldsymbol{I}) \otimes \boldsymbol{B} \otimes \boldsymbol{C}^{\top}) \right]_{(H_{\star}+1):H} \right\|_0}_{\text{eliminating redundant dimensions}} + \underbrace{\left\| \left[ \boldsymbol{P}^{\top} \boldsymbol{A} \boldsymbol{P} \right]_{1:H_{\star}, 1:H_{\star}} - \boldsymbol{A}_{\star} \right\|_0}_{\text{aligning the state matrix}} + \underbrace{\left\| \left[ \boldsymbol{P}^{\top} (\boldsymbol{B} \otimes \boldsymbol{C}^{\top}) \right]_{1:H_{\star}} - \boldsymbol{B}_{\star} \otimes \boldsymbol{C}_{\star}^{\top} \right\|_0}_{\text{aligning input-output interactions}}.$$

$$\overbrace{\phantom{aligning used dimensions with target model}}^{\text{aligning used dimensions with target model}}$$

*Proof of Lemma 8.* Combining Lemma 3 and the ZOH discretization method in (17) yields the desired results. $\qquad\square$

The insights provided by Lemma 7 and Lemma 8 are the same as those provided by Lemma 3. The analysis here supports the second step of SDLoRA presented in Sec. 5.

### D.2 EXTENSION TO DEEP S4 MODELS

Our previous analysis focused on single-channel S4 models. We now expand our investigation to more complex scenarios involving deep S4 models for both target and frozen architectures, incorporating $D$ channels and varying layer depths. In this section, we consider two PEFT methods: (i) *Selective Dimension Tuning* (SDT) and (ii) SDLoRA. The key distinction between SDT and SDLoRA lies in their treatment of linear projection matrices. SDT exclusively updates the columns of weight matrices corresponding to the updatable channels identified through Alg. 1. In contrast, SDLoRA employs LoRA to modify these matrices. It is worth noting that the linear projection matrix updates in SDT are inherently low-rank, making it a specialized case of SDLoRA. Our analysis starts with SDT, and it automatically applies to SDLoRA.

In this analysis, we assume that each input token $x_t$ belongs to $\mathcal{X}$, a bounded subset of $\mathbb{R}^D$, and that the length of the input sequence is finite. Let the frozen model have $L$ layers, and the target model have $L^{\star}$ layers, where $L \geq L^{\star}$. Similar to the technique used in Zeng & Lee (2024) and Giannou et al. (2023). The basic idea of updating the frozen model to match the functionality of the target model is to utilize every $\lceil L/L^{\star} \rceil$ layers of the frozen model to approximate every layer of the target model. We start introducing this proof idea from the simplest case where $L^{\star} = 1, L = D$. In this scenario, we can simply choose one different channel to tune and maintain all other channels at zero at every layer. The outputs from the various channels of the deep S4 layers are then combined through a residual connection. This proof idea inspires us to perform channel selection and make use of the residual connections, which is the first and third step of SDLoRA presented in Sec. 5. Building on this idea, we present the following results for when the target model has only $L^{\star} = 1$ layer, and $L = D = 2$.

**Lemma 9.** *Consider a $D$-dimensional input sequence. Assume that the linear layers in the model have linear activation functions. Using SDT, any deep S4 model with $H$ hidden states per channel and $L$ layers can be updated to accurately present any target one-layer deep S4 model without residual connections, having a reduced hidden state dimension $H^{\star} < H$. Then this can be achieved by selectively fine-tuning at most $\lceil D/L \rceil$ channels, $H^{\star}$ hidden states, and residual connections at each layer, while additionally fully fine-tuning the linear projection matrix of the last layer only.*

*Proof of Lemma 9.* In this proof, we start by considering the case where $L = D$. In this case, we update a single distinct channel for each layer while setting the other channels to zero. Essentially, we modify the frozen model so that each layer corresponds to and functions as an individual channel in the target model. To be more specific, we fully update the first channel in the first layer to match the first channel of the target model, second channel in the second layer to match the second channel of the target model, so on and so forth.

For the $l$-th layer of the frozen model, we append subscript $l$ to all parameters of the deep S4 layer as introduced in (4). For the $d$-th channel, corresponding notations are denoted with a superscript $(d)$. We define the $t$-th intermediate output token of the $l$-th deep S4 layer as $\boldsymbol{z}_{l,t} \in \mathbb{R}^D$. Additionally, the

updated S4 module in layer $l$ is denoted as $\widehat{\text{S4}}_l$, with $\widehat{\text{S4}}_{l,t}$ referring specifically to the sub-function that outputs the $t$-th token. Therefore, for the $t$-th intermediate output token of the $l$-th deep S4 layer of the updated model can be written as

$$\boldsymbol{z}_{l,t} = \widehat{\boldsymbol{W}}_l \cdot \widehat{\text{S4}}_{l,t}(\boldsymbol{z}_{l-1,1}, \ldots, \boldsymbol{z}_{l-1,t}) + \widehat{\boldsymbol{\beta}}_l + \widehat{\boldsymbol{u}}_l \otimes \boldsymbol{z}_{l-1,t}$$

$$= \widehat{\boldsymbol{W}}_l \cdot \begin{bmatrix} \widehat{\text{S4}}_{l,t}^{(1)}(z_{l-1,1}^{(1)}, \ldots, z_{l-1,t}^{(1)}) \\ \vdots \\ \widehat{\text{S4}}_{l,t}^{(D)}(z_{l-1,1}^{(D)}, \ldots, z_{l-1,t}^{(D)}) \end{bmatrix} + \widehat{\boldsymbol{\beta}}_l + \widehat{\boldsymbol{u}}_l \otimes \boldsymbol{z}_{l-1,t},$$

where $\widehat{\boldsymbol{W}}_l \in \mathbb{R}^{D \times D}, \widehat{\boldsymbol{\beta}}_l \in \mathbb{R}^D$ are the updated weight and biases of the $l$-th layer of the frozen model, and $\widehat{\boldsymbol{u}}_l \in \mathbb{R}^D$ is the updated residual connection weight of the frozen model.

**For layers $l < L = D$.** We follow the steps provided in Sec. 5 to update the $l$-th layer of the frozen model such that it functionally equivalent to the $l$-th channel of the target model. For the reader's convince, we restate our strategies here:

- **(Channel Selection)** Select $D' \leq D$ ($D' = 1$ here) important channels for making predictions. Any channel $d$ that is not utilized will have their corresponding $\boldsymbol{C}^{(d)}$ set to zero, eliminating the need to update parameters for $\boldsymbol{A}^{(d)}$ and the $d$-th column of $\boldsymbol{W}$. To be more specific, we let $\boldsymbol{C}^{(d)} = \boldsymbol{0}$ for all $d \neq l$ in this scenario.

- **(Hidden State Selection)** Within the selected channels, select $H' \leq H$ important hidden states. For any hidden state that is not used within a selected channel $d$, the corresponding element in $\boldsymbol{C}^{(d)}$ will be set to zero, thus eliminating the need to tune the corresponding element in $\boldsymbol{A}^{(d)}$. To be more specific, we can achieve $\widehat{\text{S4}}_{l,t}^{(l)}(\cdot) = \text{S4}_{\star,t}^{(l)}(\cdot)$ by Lemma 3.

- **(Residual and Bias Tuning)** Regardless of other selections, SDLoRA consistently tunes the coefficients of residual connections and biases in linear projections, as these components contain a negligible number of parameters. In this scenario, we let $\widehat{\boldsymbol{\beta}}_l = \boldsymbol{0}$, $\widehat{\boldsymbol{u}}_l = [\underbrace{1 \ \cdots \ 1}_{l-1 \text{ elements}} \ 0 \ \underbrace{1 \ \cdots \ 1}_{D-l \text{ elements}}]^\top$.

This construction yields

$$\boldsymbol{z}_{l,t} = \begin{bmatrix} z_{l-1,t}^{(1)} & \cdots & z_{l-1,t}^{(l-1)} & \text{S4}_{\star,t}^{(l)}(z_{l,1}^{(l)}, \ldots, z_{l,t}^{(l)}) & z_{l-1,t}^{(l+1)} & \cdots & z_{l-1,t}^{(D)} \end{bmatrix}^\top.$$

Consequently, only the $l$-th channel is active in the $l$-th layer, while all other layers function as identity mappings, propagating the output of the preceding layer without modification.

**For layer $l = L = D$.** Based on the setup of the first $L - 1$ layers, we have

$$\boldsymbol{z}_{L-1,t} = \begin{bmatrix} \text{S4}_{\star,t}^{(1)}(x^{(1)}) & \cdots & \text{S4}_{\star,t}^{(L-1)}(x^{(L-1)}) & x^{(L)} \end{bmatrix}^\top.$$

For the last layer, we let

$$\widehat{\boldsymbol{W}}_L = \boldsymbol{W}_\star, \quad \widehat{\boldsymbol{\beta}}_L = \boldsymbol{\beta}_\star, \quad \widehat{\boldsymbol{u}}_L = \boldsymbol{0},$$

$$\widehat{\text{S4}}_{L,t}^{(L)}(\cdot) = \text{S4}_{\star,t}^{(L)}(\cdot), \text{ which can be achieved by Lemma 3.}$$

It is easy to verify that the output of the updated frozen model is identical to the output of the target model, i.e.,

$$\boldsymbol{y}_t = \boldsymbol{z}_{L,t} = \boldsymbol{W}_\star \begin{bmatrix} \text{S4}_{\star,t}^{(1)}(x^{(1)}) & \cdots & \text{S4}_{\star,t}^{(L-1)}(x^{(L-1)}) & \text{S4}_{\star,t}^{(L)}(x^{(L)}) \end{bmatrix}^\top + \boldsymbol{\beta}_\star.$$

Thus far, we have demonstrated that the statement holds when $L = D$. This analysis can be readily extended to cases where $L \neq D$ by tuning $\lceil D/L \rceil$ channels at each layer. For example, when $L = D/2$, we can tune two channels per layer using a construction similar to the one described above. This generalization completes the proof. $\qquad\square$

**Theorem 10** (Expressive Power of SDLoRA on Deep S4 Models). *Consider a $D$-dimensional input sequence. Assume that the linear layers in the model have linear activation functions. Using SDT, any deep S4 model with $H$ hidden states per channel and $L$ layers can be updated to accurately present any target deep S4 model without residual connections, having a reduced hidden state dimension $H^\star < H$, and fewer layers $L^\star < L$. This can be achieved by selectively fine-tuning at most $\lceil DL^\star/L \rceil$ channels, $H^\star$ hidden states, and residual connections at each layer.*

*Proof of Theorem 10.* We update every $\lceil D/L \rceil$ layers of the frozen model to approximate each layer of the target model. By applying Lemma 9 iteratively to each set of $\lceil D/L \rceil$ layers, we obtain the desired result. $\qquad\square$

For reader's convience, we restate the following statement presented in the main body again here.

**Theorem 4** (Expressive Power of SDLoRA on Deep S4 Models). *Consider a $D$-dimensional input sequence. Assume that the linear layers in the model have linear activation functions. Using SDLoRA, any deep S4 model with $H$ hidden states per channel and $L$ layers can be updated to accurately present any target deep S4 model without residual connections, having a reduced hidden state dimension $H^\star < H$, and fewer layers $L^\star < L$. This can be achieved by selectively fine-tuning at most $\lceil DL^\star/L \rceil$ channels, $H^\star$ hidden states on SSM modules, applying rank-$\lceil \frac{L}{L^\star} \rceil$ updates on linear projection matrices and updating residual connections and biases at each layer, while additionally fully fine-tuning the linear projection matrix of the last layer only.*

*Proof of Theorem 4.* Since SDT is a special case of SDLoRA, Theorem 10 directly implies the desired statement. $\qquad\square$

**SDLoRA for Mamba.** In the Mamba model, the output mapping vector $C$ is input-dependent, making it unsuitable for direction modification. Therefore, we focus our channel and hidden state selection solely on $A$. For any channels or hidden states that are not selected, we set the corresponding elements of $A$ to minimal values, effectively setting the associated entries in $\overline{A}$ to zero. For channels and states that are updatable, we update the corresponding entries for $A$. However, since $B^{(d)}$ and $C^{(d)}$ cannot be directly updated, we modify the corresponding weight matrices that compute these vectors. Specifically, for updatable channels, we update the corresponding columns in $W_B$ and $W_C$; for updatable states, we adjust the corresponding rows in these weight matrices.

### D.3 Experiments on Deep S4 Models

**Synthetic.** For selecting channels and hidden states, we initiate with a warmup learning rate between $1e-2$ and $1e-3$ and conduct 20 warmup iterations. Learning rates are adjusted between $5e-2$, $1e-2$, $5e-3$, and $1e-3$. We apply LoRA with ranks of 2 and 4 to the SSM and with ranks of 4, 8, and 16 to the linear projection matrices. Non-zero states are selected from the sets $\{4, 8\}$, and non-zero channels from $\{8, 16\}$.

We additionally consider SDT (Selective Dimension Tuning), which is introduced in Sec. D.2, and the results are visualized in Fig. 2. We observe that SDT even outperforms SDLoRA in this synthetic experiments, demonstrating highly promising performance. Unfortunetly, we fail to make it work on pretraned Mamba, and identify it as one of the promising future directions.

**CIFAR-10 (Krizhevsky et al., 2009).** We adhere to the preprocessing steps for CIFAR-10 as outlined by Gu et al. (2022a). The LoRA ranks for linear projection matrices are tuned among $\{1, 2, 4, 8, 16\}$, and for the S4 component, ranks are set from $\{1, 2, 4\}$. Non-zero states are chosen from $\{8, 12, 16\}$, and non-zero channels from $\{48, 64\}$. A warmup phase includes 1 epoch with a learning rate of $1e-2$. For linear projection matrices, LoRA ranks are explored at $\{2, 4, 8, 16\}$, and for the SSM, ranks at $\{2, 4, 8\}$. All state dimensions are updated, and channel dimensions considered for updates are $\{4, 8, 16, 32\}$.

### D.4 Experiments on Pretrained Mamba

Here, we provide more experiment details. Unless otherwise stated, our experiment setting is identical to Sec. C.1. For LoRA, we consider three different LoRA configurations at each layer, involving

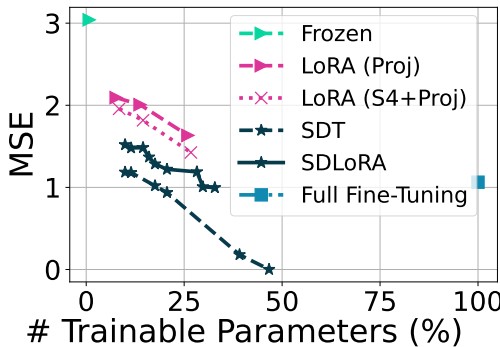

Figure 2: **Performance comparison between various methods.** SDT (Selective Dimension Tuning) is compared to SDLoRA. Unlike SDLoRA, which applies LoRA to linear projection matrices, SDT performs sparse tuning on linear projection matrices by updating only the columns corresponding to channels selected via Alg. 1. Notably, SDT achieves superior performance, matching full fine-tuning results while using only 25% of the parameters, and even surpassing full fine-tuning with more parameters. Extending SDT to real datasets is considered a promising future direction for SDLoRA.

the following matrices which comprise most of the parameters: $W_{\text{out}}$ (output linear projection), $W_B, W_C$ (weight matrices for computing input-dependent $B_n, C_n$), and $W_{\Delta,\downarrow}, W_{\Delta,\uparrow}$ (down and up projection matrices of LoRA adapters for computing $\Delta$). The three LoRA application methods are: (i) $W_{\text{out}}, W_B, W_C$, and $W_{\Delta,\downarrow}, W_{\Delta,\uparrow}$; (ii) $W_{\text{out}}, W_B, W_C$ and $W_{\Delta,\downarrow}$; and (iii) $W_{\text{out}}$ and $W_{\Delta,\uparrow}$. For SDLoRA, we set the channel freeze ratio at 99% across all scenarios. We select the state freeze ratio $\alpha$ from the set $75\%, 90\%, 95\%$ and apply LoRA exclusively to $W_{\text{out}}$ to maintain a comparable number of trainable parameters. Residual connections and bias are frozen in this experiment. For the warmup, we employ 500 data batches to fully train the SSM modules prior to dimension selection, except for the RTE task in GLUE, where we use 250 batches due to its limited dataset size. Note that the parameters are reverted back after the warmup stage.

## D.5  SDLoRA RESULTS ON ADDITIONAL DATASET

CelebA (Liu et al., 2015) comprises 202,599 face images ($178 \times 218$ pixels), which is significantly larger than CIFAR-10, and contains 40 classification tasks (e.g., predicting attributes like gender, hair color, and glasses). We report four metrics: (i) average accuracy and overall accuracy for (ii) easy, (iii) medium, and (iv) hard tasks. Here, overall accuracy refers to the accuracy of correctly predicting all target labels within a specific subset of tasks. Tasks are categorized as easy (13 tasks), medium (14 tasks), or hard (13 tasks) by clustering based on average performance. To ensure computational feasibility, we reduced the resolution by cropping images to retain only the face and then resizing them to $32 \times 32$ pixels. This preprocessing helps maintain a manageable sequence length for efficient runtime.

**Results**  We conducted experiments on Mamba-130M, and the results are summarized in Table 19. The table demonstrates that SDLoRA consistently outperforms LoRA across tasks of varying difficulty levels.

## D.6  EXPERIMENTS ON JAMBA AND MAMBA-II

In this section, we expand our analysis beyond the deep S4 model and Mamba. Specifically, we incorporate the Transformer-SSM hybrid model Jamba (Lieber et al., 2024) (Jamba-Tiny-319M and Jamba-Mini-52B) and Mamba-II (Dao & Gu, 2024) (Mamba-II-130M and Mamba-II-1.3B).

**Experiment Results on Jamba**  We froze the Transformer layers, tuning only the Mamba layers, while adhering to the same experimental settings used for Mamba. To accommodate the Jamba-Tiny

| Model | Mamba-130M | | | | |
|---|---|---|---|---|---|
| Dataset | Params (%) | CelebA | | | |
| Metric (↑) | | Acc. (All) | Acc. (Easy) | Acc. (Medium) | Acc. (Hard) |
| LoRA | 0.3178 | 87.79 | 58.53 | 24.19 | 4.18 |
| | 0.3600 | 88.58 | 60.10 | 26.21 | 5.19 |
| | 0.3883 | 87.67 | 58.32 | 24.01 | 4.08 |
| SDLoRA | 0.3492 | **88.61** | 60.50 | 26.27 | **5.40** |
| | 0.3498 | 88.40 | 59.75 | 25.69 | 5.01 |
| | 0.3509 | 88.50 | **60.52** | **26.30** | 4.96 |

Table 19: Performance comparison between SDLoRA and LoRA on CelebA (Liu et al., 2015) using Mamba-130M. **Bold** numbers indicate the best performance for each task.

52B model on a single 80GB GPU, we quantized all non-Mamba layers to 4-bit precision, following an approach similar to QLoRA, and reduced the batch size.

The performance comparison between LoRA and SDLoRA is shown in Table 20. SDLoRA outperforms LoRA on nine out of eleven tasks, demonstrating that SDLoRA's strong performance on Mamba effectively transfers to hybrid models as well.

| Model | Jamba-Tiny-319M | | | | | | | Jamba-Mini-52B | | | | | |
|---|---|---|---|---|---|---|---|---|---|---|---|---|---|
| Dataset | Params (%) | DART | | SAMSum | | | Spider | Params (%) | DART | | SAMSum | | |
| Metric (↑) | | METEOR | BLEU | R1 | R2 | RL | Acc. | | METEOR | BLEU | R1 | R2 | RL |
| LoRA | 0.05030 | 65.03 | 27.17 | 37.13 | 16.43 | 30.90 | 35.49 | 0.004951 | 73.00 | 52.86 | 55.31 | 31.71 | 46.47 |
| | 0.05690 | **67.90** | **39.02** | 40.80 | 18.54 | 33.87 | 44.07 | 0.005629 | 72.81 | 52.65 | 55.12 | 31.63 | 46.64 |
| | 0.06153 | 65.05 | 23.18 | 39.15 | 17.70 | 32.79 | 37.67 | 0.006051 | 72.94 | 52.63 | 56.36 | 33.48 | 47.91 |
| SDLoRA | 0.05536 | 67.18 | 31.49 | 41.11 | 18.48 | 33.84 | 48.58 | 0.005484 | 72.87 | 52.46 | 56.08 | 32.79 | 47.61 |
| | 0.05540 | 67.86 | 31.43 | 41.69 | 19.17 | 34.47 | **50.40** | 0.005488 | **73.07** | 52.79 | **56.53** | **33.50** | **47.96** |
| | 0.05549 | 67.80 | 33.03 | **42.18** | **19.19** | **34.95** | 49.60 | 0.005497 | 72.95 | **53.11** | 56.14 | 33.08 | 47.56 |

Table 20: Performance comparison between SDLoRA and LoRA on Jamba-Tiny-319M and Jamba-Mini-52B. **Bold** numbers indicate the best performance for each task.

**Experiment Results on Mamba-II** For Mamba-II, however, applying SDLoRA is not straightforward because Mamba-II further constrains $A$ such that all (non-zero) entries must have the same value. Therefore, our original dimension selection approach cannot be directly applied here. We consider a naive extension of SDLoRA by selecting dimensions in the projection matrices for input mapping vector $B$ and the projection matrices for output mapping vector $C$ using their respective magnitude, and fine-tune the selected dimensions and all elements of state transition matrix $A$.

Tables 21 and 22 compare the performance of LoRA and SDLoRA on Mamba-II. The results demonstrate that SDLoRA consistently outperforms LoRA on Mamba-II models.

| Model | Mamba-II-130M | | | | Mamba-II-1.3B | | | | |
|---|---|---|---|---|---|---|---|---|---|
| Dataset | Params (%) | DART | | Params (%) | SAMSum | | | Spider | |
| Metric (↑) | | METEOR | BLEU | | R1 | R2 | RL | Acc. | |
| LoRA | 0.3354 | 68.71 | 48.09 | 0.1614 | 49.73 | 26.14 | 41.53 | 72.36 | |
| SDLoRA | 0.3393 | **70.60** | **48.93** | 0.1767 | **50.72** | **27.21** | **42.54** | **84.15** | |

Table 21: Performance comparison between SDLoRA and LoRA on Mamba-II-130M and Mamba-II-1.3B. **Bold** numbers indicate the best performance for each task.

### D.7 DORA AND SDDORA RESULTS

We have included evaluations of DoRA (an advanced LoRA variant) alongside SDDoRA to provide a more comprehensive analysis. We extended our investigation to include SDDoRA and evaluated its performance against DoRA alone using the DART benchmark on the Mamba-130M model. The results, presented in Table 23, show that integrating selective dimension tuning with DoRA enhances its effectiveness and achieves superior performance compared to using DoRA alone.

| Model | Mamba-II-130M | | | | | | | |
|---|---|---|---|---|---|---|---|---|
| Dataset Accuracy (↑) | Params (%) | | | | GLUE | | | |
| | | RTE | MRPC | COLA | SST2 | QNLI | QQP | MNLI |
| LoRA | 0.3354 | 63.4 | 80.9 | - | 89.1 | 85.3 | 87.1 | 78.6 |
| SDLoRA | 0.3393 | **64.3** | **82.3** | - | **94.1** | **87.0** | **88.3** | **81.1** |

Table 22: Performance comparison between SDLoRA and LoRA on GLUE (Wang et al., 2019) dataset using Mamba-II-130M. **Bold** numbers indicate the best performance for each task (- indicates experiments still under investigation due to identified issues).

| Model | Mamba-130M | | | Mamba-1.4B | | | |
|---|---|---|---|---|---|---|---|
| Dataset Metric (↑) | Params (%) | DART | | Params (%) | SAMSum | | |
| | | METEOR | BLEU | | R1 | R2 | RL |
| DoRA | 0.3618 | 70.01 | 49.86 | 0.1813 | 51.42 | 27.78 | 42.89 |
| | 0.4025 | 70.40 | 51.22 | 0.2024 | 51.78 | 27.70 | 43.23 |
| | 0.4040 | 69.94 | 50.53 | 0.2024 | 51.75 | 28.04 | 43.44 |
| SDDoRA | 0.3630 | 70.33 | 51.32 | 0.1831 | **52.11** | **28.28** | **43.65** |
| | 0.3633 | **70.80** | **51.55** | 0.1832 | 51.86 | 28.28 | 43.48 |
| | 0.3639 | 70.50 | 50.80 | 0.1835 | 51.70 | 28.02 | 43.39 |

Table 23: Performance comparison between SDDoRA and DoRA on Mamba-130M and Mamba-1.4B. **Bold** numbers indicate the best performance for each task.

## D.8 LoRA+ AND SDLoRA+ RESULTS

We have included evaluations of LoRA+ (Hayou et al., 2024) (an advanced LoRA variant) alongside SDLoRA+ to provide a more comprehensive analysis. We extended our investigation to include SDLoRA+ and evaluated its performance against LoRA+ across various datasets on both Mamba-I and Mamba-II. The results, presented in Table 24, show that integrating selective dimension tuning with LoRA+ enhances its effectiveness and achieves superior performance compared to using LoRA+ alone.

| Model | Mamba-I-130M | | Mamba-II-130M | | Mamba-II-1.3B | | | |
|---|---|---|---|---|---|---|---|---|
| Dataset Metric (↑) | DART | | DART | | SAMSum | | | Spider |
| | METEOR | BLEU | METEOR | BLEU | R1 | R2 | RL | Acc. |
| LoRA+ | 70.06 | 50.91 | 69.78 | 49.14 | 49.83 | 26.09 | 41.66 | 73.75 |
| SDLoRA+ | **70.58** | **51.93** | **70.48** | **49.99** | **50.81** | **27.19** | **42.4** | **84.22** |

Table 24: Performance comparison between SDLoRA+ and LoRA+ on Mamba-I and Mamba-II. **Bold** numbers indicate the best performance for each task. We test all experiments under various parameter settings (<0.4%) for both LoRA+ and SDLoRA+, and report the best values.

## D.9 MEMORY USAGE AND RUNTIME ANALYSIS OF SDLoRA

To assess the memory usage and runtime of SDLoRA and LoRA, we conducted experiments on four different models, including both SSM and hybrid architectures. Unless specified otherwise, for each model and method, dataset were generated with 2,500 batches of data samples, each batch comprising a random sequence of 1,500 tokens. The simulation was repeated four times, including dataset generation. All experiments were carried out on a single H100 GPU, and the reported metrics represent averages across the four simulations. Consistent with our previous experiments, we used the original hyperparameter settings, ensuring that SDLoRA included more trainable parameters than LoRA.

**Memory Usage Analysis**  The memory usage of LoRA and SDLoRA is presented in Table 25. Our observations indicate that SDLoRA requires less memory than LoRA. This difference can be

attributed to the design of the LoRA adapters, which involve matrix multiplication of two low-rank matrices. In contrast, tuning SSM with the same number of parameters does not require any matrix multiplication, resulting in lower memory usage.

| Memory Usage (GB) | Mamba-130M | Mamba-1.4B | Jamba-Tiny-319M | Jamba-Mini-52B |
|---|---|---|---|---|
| LoRA | 7.753 | 37.167 | 7.207 | 71.986 |
| SDLoRA | **5.738** | **26.491** | **6.605** | **67.193** |

Table 25: Memory usage comparison between SDLoRA and LoRA on various models. **Bold** numbers indicate the lowest memory usage for each model.

**Runtime Analysis**  Fine-tuning with SDLoRA consists of two stages: (1) dimension selection and (2) standard training. In this study, we first compare the runtime of SDLoRA and LoRA during stage 2 (training) and then evaluate the additional runtime introduced by SDLoRA during stage 1 (dimension selection). Our results show that the dimension selection stage adds only marginal runtime overhead, and SDLoRA is more efficient than LoRA in standard training.

*Training:* When the channels and states have been selected, the training of SDLoRA is faster than LoRA when the same number of trainable parameters are considered.

The runtimes are reported in Table 26. We observe that, despite having more trainable parameters, SDLoRA is faster than LoRA. We attribute this to the fact that LoRA introduces additional FLOPs due to the extra matrix multiplication operations required for each update (specifically, the multiplication of two low-rank matrices).

| Avg. Runtime (Seconds) | Mamba-130M | Mamba-1.4B | Jamba-Tiny-319M | Jamba-Mini-52B |
|---|---|---|---|---|
| LoRA | $410.0 \pm 80.0$ | $2060.0 \pm 135.0$ | $352.5 \pm 107.5$ | $3427.5 \pm 185.0$ |
| SDLoRA | $\mathbf{330.0} \pm 77.5$ | $\mathbf{1697.5} \pm 87.5$ | $\mathbf{257.5} \pm 72.5$ | $\mathbf{3065.0} \pm 232.5$ |

Table 26: Runtime comparison of SDLoRA and LoRA during stage 2 (training).

*Dimension Selection:* For dimension selection, our method first performs an Initial Subset Training, and then selects the dimensions based on the magnitude of parameter changes across different dimensions.

1. *Initial Subset Training:* We update the model by going through only a subset of the dataset (e.g., 3% of batches in DART experiments), which is sufficient in practice.

2. *Magnitude-Based Dimension Selection:* After the subset training, we select dimensions based on the magnitude of parameter changes observed.

In this experiment, we simulate a real scenario using datasets with 2,500 batches, considering a small subset containing 125 batches (5% of the full dataset). We repeat the experiments 80 times, and the reported numbers are averaged across these simulations. The following table presents the runtime analysis of the dimension selection stage in SDLoRA.

Table 27 demonstrates that the dimension selection stage adds only negligible runtime.

| Avg. Runtime (Seconds) | Mamba-130M | Mamba-1.4B | Jamba-Tiny-319M | Jamba-Mini-52B |
|---|---|---|---|---|
| Initial Subset Training | $16.250 \pm 3.880$ | $85.250 \pm 5.130$ | $15.750 \pm 1.000$ | $163.630 \pm 10.120$ |
| Magnitude-Based Dimension Selection | $0.280 \pm 0.000$ | $0.520 \pm 0.120$ | $0.090 \pm 0.000$ | $0.240 \pm 0.040$ |
| Total Time | $16.530 \pm 3.880$ | $85.770 \pm 5.250$ | $15.840 \pm 1.000$ | $163.870 \pm 10.160$ |
| Proportion of Training 1 Epoch | $0.050\times$ | $0.051\times$ | $0.062\times$ | $0.053\times$ |
| Proportion of Training 5 Epoch | $\mathbf{0.010}\times$ | $\mathbf{0.010}\times$ | $\mathbf{0.012}\times$ | $\mathbf{0.011}\times$ |

Table 27: Runtime comparison of SDLoRA and LoRA during stage 1 (dimension selection).

# E    LIMITATIONS & FUTURE WORKS

While our work offers numerous valuable insights, some limitations exist. Theoretically, our guarantees for SDLoRA are limited to linear activations and require full fine-tuning of the last layer. However, our experiments demonstrate that SDLoRA does not suffer from these limitations in practice. Removing such restrictions for SDLoRA in theory or developing new PEFT methods under more general theoretical cases is an interesting future direction. Additionally, our theory only demonstrates that updating a subset of channels and states is sufficient, without providing guidance on optimal selection. Our channel and state selection, based on a warmup stage and parameter magnitude, may not be optimal. Further investigation into the impact of channel/state selection and development of improved dimension selection algorithms presents an interesting avenue for future work. Lastly, our work primarily focuses on SSM-based models. Studying PEFT methods on SSM-Transformer hybrid models (Lieber et al., 2024; Park et al., 2024), is an interesting future direction.

