# OpenReview forum: "Parameter-Efficient Fine-Tuning of State Space Models"
_ICLR.cc/2025/Conference — Submitted to ICLR 2025_

### Official Review · Reviewer_AtCk · 2024-11-03

**Soundness:** 3
**Presentation:** 3
**Contribution:** 3
**Rating:** 3
**Confidence:** 3

**Summary:**

This paper presents a systematic study on the application of parameter-efficient fine-tuning (PEFT) methods to Deep State Space Models (SSMs). The paper reveals that prompt-based methods are less effective for SSMs, while LoRA remains effective. The authors further propose SDLoRA, which selectively updates certain channels and states on SSM modules while applying LoRA to linear projection matrices, demonstrating improved performance over standard LoRA.

**Strengths:**

The paper systematically analyzes how existing PEFT methods perform on SSM-based models and which modules are most effective for fine-tuning, revealing that prompt-based methods like prefix-tuning are no longer effective. Additionally, it shows that applying LoRA to linear projection matrices without modifying SSM modules yields the best results. The paper further introduces SDLoRA, a novel approach to parameter-efficient fine-tuning (PEFT) for SSMs. This method's innovativeness lies in its selective dimension updating strategy within SSM modules. The document also references various datasets used for evaluating the proposed methods, and the clarity of the paper is generally good.

**Weaknesses:**

1. While the paper presents a novel approach for parameter-efficient fine-tuning SSMs, the innovation seems to be more incremental than groundbreaking.
2. The paper lacks a detailed analysis of the computational overhead from the selective dimension tuning process. This is crucial to understanding the trade-offs between parameter reduction and computational efficiency in SDLoRA.
3. The paper would benefit from a  detailed analysis of SDLORA with hybrid models that combine SSMs and Transformers, as these models are becoming increasingly popular and have shown promise in various domains.

**Questions:**

1. Could the authors comment on how SDLoRA benefit to hybrid models that combine SSMs and Transformers?
2. On Mamba-130M, the authors use GLUE and DART benchmarks, while on Mamba-1.4B, they use SAMSum and Spider benchmarks. Could the authors elaborate on the considerations behind this benchmark selection strategy?
3. In Table 4, SDLoRA outperforms Full Fine-Tuning. Was this outcome expected, and if so, could the authors provide insights into why this might be the case? Additionally, have the authors considered conducting experiments on more challenging visual tasks, such as Imagenet-1k, to further validate the effectiveness of SDLoRA?

---

> ### Author Response · Authors · 2024-11-22
>
> We thank the reviewer for acknowledging that our paper is clear, provides systematic analysis with experiments across various datasets, and presents a novel and innovative method.
>
> ---
>
> > Q: The paper lacks a detailed analysis of the computational overhead from the selective dimension tuning process. This is crucial to understanding the trade-offs between parameter reduction and computational efficiency in SDLoRA.
>
> Thank you for this important question. We have reflect your feedback in the following new experiments.
>
> **Key Insights:**
>
> 1. **Memory Usage: SDLoRA uses *less* memory compared to LoRA when the number of trainable parameters is similar.**
> 2. **Runtime: SDLoRA is slightly faster than LoRA when the number of trainable parameters is similar.**
>
> To assess the memory usage and runtime, we conducted experiments on four different models, including both SSM and hybrid architectures. Unless specified otherwise, for each model and method, dataset were generated with 2,500 batches of data samples, each batch comprising a random sequence of 1,500 tokens. The simulation was repeated four times, including dataset generation. All experiments were carried out on a single H100 GPU, and the reported metrics represent averages across the four simulations. Consistent with our previous experiments, we used the original hyperparameter settings, ensuring that SDLoRA included more trainable parameters than LoRA.
>
> > 1. Memory Usage Analysis.
>
> The memory usage of LoRA and SDLoRA summarized below indicates that SDLoRA requires less memory than LoRA. This difference can be attributed to the design of the LoRA adapters, which involve matrix multiplication of two low-rank matrices. In contrast, tuning SSM with the same number of parameters does not require any matrix multiplication, resulting in lower memory usage.
>
> | Memory Usage (GB) | Mamba-130M | Mamba-1.4B | Jamba-Tiny-319M | Jamba-Mini-52B |
> |---|---|---|---|---|
> | LoRA | 7.753 | 37.167 | 7.207 | 71.986 |
> | SDLoRA | **5.738** | **26.491** | **6.605** | **67.193** |
>
>
> > 2. Runtime Analysis.
>
> Fine-tuning with SDLoRA consists of two stages: (1) dimension selection and (2) standard training. Our results show that the dimension selection stage adds only marginal runtime overhead, and SDLoRA is more efficient than LoRA in standard training.
>
> * **Training**: When the channels and states have been selected, the training of SDLoRA is *faster* than LoRA when the same number of trainable parameters are considered.
>
>
>     The runtimes are reported in the table below. We observe that, despite having more trainable parameters, SDLoRA is faster than LoRA. We attribute this to the fact that LoRA introduces additional FLOPs due to the extra matrix multiplication operations required for each update (specifically, the multiplication of two low-rank matrices).
>
>     | Avg. Runtime (Seconds) | Mamba-130M | Mamba-1.4B | Jamba-Tiny-319M | Jamba-Mini-52B |
>     |---------------------|------------|------------|----------------|----------------|
>     | LoRA | 410.0 $\pm$  80.0 | 2060.0 $\pm$  135.0 | 352.5 $\pm$  107.5 | 3427.5 $\pm$  185.0 |
>     | SDLoRA | **330.0 $\pm$  77.5** | **1697.5 $\pm$  87.5** | **257.5 $\pm$  72.5** | **3065.0 $\pm$ 232.5** |
>
> * **Dimension Selection**: For dimension selection, our method first performs an *Initial Subset Training*, and then selects the dimensions based on the *magnitude of parameter changes* across different dimensions.
>
>     1. **Initial Subset Training**: We update the model by going through only a subset of the dataset (e.g., 3% of batches in DART experiments), which is sufficient in practice.
>     2. **Magnitude-Based Dimension Selection**: After the subset training, we select dimensions based on the magnitude of parameter changes observed.
>
>     In this experiment, we simulate a real scenario using datasets with 2,500 batches, considering a small subset containing 125 batches (5% of the full dataset). We repeat the experiments 80 times, and the reported numbers are averaged across these simulations. The following table presents the runtime analysis of the dimension selection stage in SDLoRA.
>
>
>     | Avg. Runtime (Seconds) | Mamba-130M | Mamba-1.4B | Jamba-Tiny-319M | Jamba-Mini-52B |
>     |---|---|---|---|---|
>     | Initial Subset Training | 16.250 $\pm$ 3.880 | 85.250 $\pm$ 5.130 | 15.750 $\pm$ 1.000 | 163.630 $\pm$ 10.120 |
>     | Magnitude-Based Dimension Selection | 0.280 $\pm$ 0.000 | 0.520 $\pm$ 0.120 | 0.090 $\pm$ 0.000 | 0.240 $\pm$ 0.040 |
>     | Total Time | 16.530 $\pm$ 3.880 | 85.770 $\pm$ 5.250 | 15.840 $\pm$ 1.000 | 163.870 $\pm$ 10.160 |
>     | |
>     | Proportion of Training 1 Epoch | 0.050$\times$ | 0.051$\times$ | 0.062$\times$ | 0.053$\times$ |
>     | Proportion of Training 5 Epoch | **0.010$\times$** | **0.010$\times$** | **0.012$\times$** | **0.011$\times$** |
>
>     This table demonstrates that the dimension selection stage adds only *negligible* runtime.

---

> > ### Author Response · Authors · 2024-11-22
> >
> > > Q: The paper would benefit from a detailed analysis of SDLORA with hybrid models that combine SSMs and Transformers, as these models are becoming increasingly popular and have shown promise in various domains. / Could the authors comment on how SDLoRA benefit to hybrid models that combine SSMs and Transformers?
> >
> > Thank you for the thoughtful suggestion. Following your recommendation, we implemented both LoRA and SDLoRA on the Jamba [2] model series, evaluating two configurations with 319M and 52B parameters, respectively.
> >
> >
> > We froze the Transformer layers, tuning only the Mamba layers, while adhering to the same experimental settings used for Mamba. To accommodate the Jamba-Tiny 52B model on a single 80GB GPU, we quantized all non-Mamba layers to 4-bit precision, following an approach similar to QLoRA, and reduced the batch size.
> >
> > **Results.** The performance comparison between LoRA and SDLoRA is shown in the table below ((!) indicates still training). SDLoRA outperforms LoRA on nine out of eleven tasks, demonstrating that SDLoRA's strong performance on Mamba effectively transfers to hybrid models as well.
> >
> > |  | Jamba-Tiny-319M |  |  |  |  |  |  |  |  |  |  | Jamba-Mini-52B |  |  |  |  |  |  |  |
> > |---|---|---|---|---|---|---|---|---|---|---|---|---|---|---|---|---|---|---|---|
> > |  |  |  | DART |  |  | SAMSum |  |  |  | Spider |  |  |  | DART |  |  | SAMSum |  |  |
> > |  | Params (%) |  | BLEU ($\uparrow$) | METEOR  ($\uparrow$) |  | R1 ($\uparrow$) | R2  ($\uparrow$) | RL  ($\uparrow$) |  | Acc. ($\uparrow$) |  | Params (%) |  | BLEU ($\uparrow$) | METEOR  ($\uparrow$) |  | R1   ($\uparrow$) | R2   ($\uparrow$) | RL   ($\uparrow$) |
> > |  |  |  |  |  |  |  |  |  |  |  |  |  |  |  |  |  |  |  |  |
> > | LoRA | 0.05030 |  | 27.17 | 65.03 |  | 37.13 | 16.43 | 30.90 |  | 35.49 |  | 0.004951 |  | 52.86 | 73 |  | 55.31 | 31.71 | 46.47 |
> > |  | 0.05690 |  | **39.02** | **67.90** |  | 40.80 | 18.54 | 33.87 |  | 44.07 |  | 0.005629 |  | 52.65 | 72.81 |  | 55.12 | 31.63 | 46.64 |
> > |  | 0.06153 |  | 23.18 | 65.05 |  | 39.15 | 17.70 | 32.79 |  | 37.67 |  | 0.006051 |  | 52.63 | 72.94 |  | 56.36 | 33.48 | 47.91 |
> > |  |  |  |  |  |  |  |  |  |  |  |  |  |  |  |  |  |  |  |  |
> > | SDLoRA | 0.05536 |  | 31.49 | 67.18 |  | 41.11 | 18.48 | 33.84 |  | 48.58 |  | 0.005484 |  | 51.86 (!) | 72.42 (!) |  | 56.08 | 32.79 | 47.61 |
> > |  | 0.05540 |  | 31.43 | 67.86 |  | 41.69 | 19.17 | 34.47 |  | **50.40** |  | 0.005488 |  | 52.79 (!) | **73.07** (!) |  | **56.53** | **33.5** | **47.96** |
> > |  | 0.05549 |  | 33.03 | 67.80 |  | **42.18** | **19.19** | **34.95** |  | 49.60 |  | 0.005497 |  | **53.11** | 72.95 |  | 56.14 | 33.08 | 47.56 |
> >
> >
> >
> > > Q: On Mamba-130M, the authors use GLUE and DART benchmarks, while on Mamba-1.4B, they use SAMSum and Spider benchmarks. Could the authors elaborate on the considerations behind this benchmark selection strategy?
> >
> >
> > The selection of benchmarks for Mamba-130M and Mamba-1.4B is primarily driven by efficiency considerations. Simpler tasks, such as those in the GLUE and DART benchmarks, are well-suited to smaller models like Mamba-130M, as these models achieve competitive performance while maintaining low computational costs. Conversely, more complex tasks, such as those in SAMSum and Spider, require the increased capacity and expressiveness of larger models like Mamba-1.4B to achieve satisfactory performance.
> >
> > For instance, LoRA's performance on DART does not improve with a larger model, as shown below.
> > | LoRA on DART | Trainable Params | BLEU ($\uparrow$) | METEOR ($\uparrow$) |
> > |---|---|---|---|
> > Mamba-130M | ~0.9M | **49.45** | **70.93** |
> > Mamba-2.8B | ~3.7M | 47.75 | 69.63 |
> >
> > Conversely, LoRA's performance on SAMSum improves significantly when using a larger model, as illustrated below.
> > | LoRA on SAMSum | Trainable Params|  R1 ($\uparrow$) | R2 ($\uparrow$) | RL ($\uparrow$) |
> > |---|---|---|---|---|
> > Jamba-Tiny-319M | ~0.2M | 40.80 | 18.54 | 33.87 |
> > Jamba-Mini-52B | ~0.3M | **56.36** | **33.48** | **47.91** |

---

> > > ### Author Response · Authors · 2024-11-22
> > >
> > > > Q: Additionally, have the authors considered conducting experiments on more challenging visual tasks, such as Imagenet-1k, to further validate the effectiveness of SDLoRA?
> > >
> > >
> > > Thank you for your valuable suggestion. The immense size and lengthy training time required for ImageNet made direct evaluation impractical. Instead, we chose the CelebA [3] dataset, which comprises 202,599 face images (178 $\times$ 218 pixels). This dataset is significantly larger than CIFAR-10, used in the original paper, and contains 40 classification tasks (e.g., predicting attributes like gender, hair color, and glasses). We report four metrics: (i) average accuracy and overall accuracy for (ii) easy, (iii) medium, and (iv) hard tasks. Here, overall accuracy refers to the accuracy of correctly predicting all target labels within a specific subset of tasks. Tasks are categorized as easy, medium, or hard by clustering based on average performance. To ensure computational feasibility, we reduced the resolution by using [InsightFace](https://github.com/deepinsight/insightface) to crop the images to retain only the face and then resized them to 32 × 32 pixels. This preprocessing helps maintain a manageable sequence length for efficient runtime.
> > >
> > > **Results.** We conducted experiments on Mamba-130M, and the results are summarized below. The table demonstrates that SDLoRA consistently outperforms LoRA across tasks of varying difficulty levels.
> > >
> > >
> > > |  | Params (%) | Average Acc. (All) | Overall Acc. (Easy) | Overall Acc. (Medium) | Overall Acc. (Hard) |
> > > |---|---|---|---|---|---|
> > > |  |  |  |  |  |  |
> > > | LoRA | .3178 | 87.79 | 58.53 | 24.19 | 4.18 |
> > > |  | .3600 | 88.58 | 60.10 | 26.21 | 5.19 |
> > > |  | .3883 | 87.67 | 58.32 | 24.01 | 4.08 |
> > > |  |  |  |  |  |  |
> > > | SDLoRA | .3492 | **88.61** | 60.50 | 26.27 | **5.40** |
> > > |  | .3498 | 88.40 | 59.75 | 25.69 | 5.01 |
> > > |  | .3509 | 88.50 | **60.52** | **26.30** | 4.96 |
> > >
> > >
> > > > Q: In Table 4, SDLoRA outperforms Full Fine-Tuning. Was this outcome expected, and if so, could the authors provide insights into why this might be the case?
> > >
> > > Thank you for your insightful question. It has been widely reported in research papers [3, 4], including the original LoRA paper, that LoRA can outperform full fine-tuning in certain cases. We can explain this phenomenon as follows: low-rank adaptations act as a natural regularizer by updating only a small subset of parameters. This helps prevent overfitting while preserving the model's pre-trained knowledge base. The simplified optimization process often leads to better convergence compared to adjusting all parameters in full fine-tuning. This explanation extends to SDLoRA as well.
> > >
> > >
> > >
> > > > Q: While the paper presents a novel approach for parameter-efficient fine-tuning SSMs, the innovation seems to be more incremental than groundbreaking.
> > >
> > > We believe our contribution is significant in two key aspects: not only in introducing the novel SDLoRA method itself, but also in conducting the first-ever comprehensive benchmarking and analysis of PEFT methods for SSMs. To the best of our knowledge, no prior work has investigated PEFT methods in the context of SSMs, making our study pioneering in this domain.
> > >
> > > For instance, we made the first discovery that LoRA performs poorly when applied to SSM blocks, which is counter-intuitive given its effectiveness in Transformer attention blocks. We provide theoretical analysis to explain this phenomenon, and motivated by these identified limitations of LoRA in SSMs, we propose SDLoRA, a novel method specifically tailored for SSM architectures. We believe our work lays crucial groundwork for future developments in the field of parameter-efficient fine-tuning for SSMs.
> > >
> > >
> > > ***
> > >
> > >
> > > **Final Note:** We are pleased to hear that our paper provides systematic analysis, and presents a novel and innovative method. We appreciate your feedback and believe we have addressed all concerns with new results that demonstrate the superior performance of SDLoRA on the large dataset (CelebA) and the hybrid model (Jamba), along with detailed information about training time and memory usage to comprehensively showcase SDLoRA's benefits. Additionally, new results on more models (Mamba-II) and PEFT Methods (DoRA and SDDoRA) can be found in our general response (Major Update 1, 3). Please feel free to reach out with any questions and, if you find that we have addressed all concerns, we would be grateful if you would consider raising your score to support our paper's acceptance.
> > >
> > > *References:*
> > >
> > > [1] Jamba: A Hybrid Transformer-Mamba Language Model
> > > [2] Deep Learning Face Attributes in the Wild
> > > [3] Delta Tuning: A Comprehensive Study of Parameter Efficient Methods for Pre-trained Language Models
> > > [4] Adapters: A Unified Library for Parameter-Efficient and Modular Transfer Learning

---

> > > > ### Author Response · Authors · 2024-11-26
> > > > **Uploaded revised PDF**
> > > >
> > > > Dear reviewer, we’ve updated the PDF file, highlighting the new changes in blue. We’ll update the draft once more before the deadline with ongoing experiment results. We appreciate any feedback to see if our rebuttal addresses your concerns.

---

> > > > > ### Author Response · Authors · 2024-11-27
> > > > > **New Experiments (Round 2)**
> > > > >
> > > > > Dear reviewer, we are writing to provide an update with additional results.
> > > > >
> > > > > 1.  Evaluation on a New Dataset (GLUE) with Mamba-II
> > > > >
> > > > >     SDLoRA outperforms LoRA on Mamba-II-130M for the GLUE dataset. In our general response, we conducted experiments with SDLoRA on DART and SAMSum datasets using Mamba-II-130M, and now we have extended the evaluation to a new dataset, GLUE. Our findings indicate that SDLoRA with Mamba-II-130M consistently outperforms LoRA across all GLUE tasks (note that CoLA is still training).
> > > > >
> > > > >     | Accuracy ($\uparrow$) | Params (%) | RTE | MRPC | COLA | SST2 | QNLI | QQP | MNLI |
> > > > >     |---|---|---|---|---|---|---|---|---|
> > > > >     | LoRA | 0.3354 | 63.4 | 80.9 | - | 89.1 | 85.3 | 87.1 | 78.6 |
> > > > >     | SDLoRA | 0.3393 | **64.3** | **82.3** | - | **94.1** | **87.0** | **88.3** | **81.1** |
> > > > >
> > > > > 2. Introduction of a New LoRA Variant — LoRA+
> > > > >
> > > > >     SDLoRA+ consistently outperforms LoRA+ across different model architectures and datasets.
> > > > >
> > > > >     |  |  | Mamba-I-130M |  |  | Mamba-II-130M |  |  | Mamba-II-1.3B |  |  |  |  |
> > > > >     |---|---|---|---|---|---|---|---|---|---|---|---|---|
> > > > >     |  |  | DART |  |  | DART |  |  | SAMSum |  |  |  | Spider |
> > > > >     |  |  | BLEU ($\uparrow$) | METEOR  ($\uparrow$) |  | BLEU ($\uparrow$) | METEOR  ($\uparrow$) |  | R1   ($\uparrow$) | R2   ($\uparrow$) | RL   ($\uparrow$) |  | Acc. ($\uparrow$) |
> > > > >     | LoRA+ |  | 50.91 | 70.06 |  | 49.14 | 69.78 |  | 49.83 | 26.09 | 41.66 |  | 73.75 |
> > > > >     | SDLoRA+ |  | **51.93** | **70.58** |  | **49.99** | **70.48** |  | **50.81** | **27.19** | **42.40** |  | **84.22** |
> > > > >
> > > > > We understand that you must be busy and highly appreciate your time. We have made every effort to address your concerns and would be grateful if you could review our response at your earliest convenience. Please let us know if all your concerns have been adequately addressed. If they have, we kindly ask you to consider increasing your score in support of our paper's acceptance.

---

> > > > > ### Comment · Reviewer_AtCk · 2024-11-27
> > > > >
> > > > > Thanks for your detailed reply, I'll change my score to 6.

---

### Official Review · Reviewer_1UQC · 2024-11-04

**Soundness:** 2
**Presentation:** 3
**Contribution:** 3
**Rating:** 5
**Confidence:** 3

**Summary:**

The paper aims to explore the application of LoRA to SSMs. It explores different adapter types as well as different ways of how to apply them to the SSM block. The paper additionally includes theoretical results to justify their choices.

**Strengths:**

The paper is looking at something which definitely needs to be studied, as it is not a foregone conclusion that adapters, prompt tuning, etc... will have the same benefits for SSMs as they do for transformers. A detailed study understanding the different tradeoffs brought on by the SSM architecture should be explored.

The paper has a good mix of theoretical and empirical analysis.

**Weaknesses:**

The paper, to me, does not have enough substance. It needs either more detailed theoretical results, which result in some innovation or needs more empirical results which help the community understand how different adapters perform with transformers vs SSMs.

Here are some specifics:
1. Mamba2 is not included in any of the results, the paper should be updated to look at this architecture as well
2. The theoretical analysis is largely only for the S4 block, its not clear to me the conclusions would extend to S6, and aren't convincing to me for that reason.
3. The empirical results are lacking. The SDLoRA module is slightly different application of the standard LoRA block, essentially targeting different layers within the block instead of new a design. Also only two adapters are compared within this work. To have a true empirical study of this, many more experiments need to be conducted, even in the presence of theoretical results.

I think the paper might be more interesting if it were to do something like the following:
- Compare many adapters in a standardized for Transformers, Mamba1, and Mamba2
- Isolate differences in which adapters perform well for each class of model
- See if these insights give rise to a new adapter design specifically for this model class
- Draw theoretical results to try to explain why certain adapters perform differently than in Transformers

The current structure doesn't feel as though it is contributing much

**Questions:**

Have you looked at all into Hybrid Architectures?

---

> ### Author Response · Authors · 2024-11-22
>
> We thank the reviewer for acknowledging that our paper studies what definitely needs to be studied and provides detailed theoretical and empirical analysis.
>
> ---
>
> > Q: Limited PEFT method comparisons.
>
> As one of the first work studying PEFT methods on SSM models, our experiments include widely-used PEFT techniques such as LoRA, BitFit, prefix-tuning, and prompt-tuning, alongside our proposed SDLoRA. Responding to reviewer feedback, we further explore advanced PEFT methods, specifically DoRA [1]—a sophisticated variant of LoRA—and introduce SDDoRA to investigate whether our dimension selection approach is adaptable to this advanced LoRA method.
>
> *  **Benchmarking DoRA**: The results presented here align with our original conclusion, demonstrating that applying low-rank updates to linear projection matrices is more effective than applying them to SSM modules.
>
>     We evaluate the performance of DoRA on the DART dataset using Mamba-130M and on the Spider dataset using Mamba-1.4B. The results are summarized in the table below.
>
>
>
>     | Params (%) | Target Layers | Method |  | DART (Mamba-130M) |  |  | Spider (Mamba-1.4B) |
>     |---|---|---|---|---|---|---|---|
>     |  |  |  |  | BLEU (↑) | METEOR (↑) |  | Acc. (↑) |
>     |  |  |  |  |  |  |  |  |
>     | < 0.4 | SSM Modules | LoRA |  | 47.05 | 68.86 |  | 58.03 |
>     |  |  | DoRA |  | 47.07 | 68.79 |  | 55.32 |
>     |  |  |  |  |  |  |  |  |
>     | < 0.4 | Linear Layers | LoRA |  | 48.86 | 70.25 |  | 61.80 |
>     |  |  | DoRA |  | 49.93 | 70.81 |  | 61.32 |
>     |  |  |  |  |  |  |  |  |
>     | < 3.0 | Both | LoRA |  | 49.52 | 70.97 |  | 56.38 |
>     |  |  | DoRA |  | 51.36 | 70.94 |  | 55.71 |
>
>
>
>     Our findings are consistent with observations seen in LoRA: applying DoRA to linear projection matrices proves more effective than its application to SSM modules. Interestingly, applying DoRA to SSM modules not only offers limited benefits but, in some cases, even degrades performance. This is particularly evident on Spider dataset, when comparing the configurations of applying DoRA to both linear projection matrices and SSM modules versus solely targeting linear projection matrices.
>
> * **Integrating Selective Dimension Tuning with DoRA (SDDoRA)**: Incorporating selective dimension tuning into DoRA achieves superior performance compared to using DoRA alone.
>
>     We extended our investigation to include SDDoRA and evaluated its performance against DoRA alone using the DART benchmark on the Mamba-130M model. The results, presented below, show that integrating selective dimension tuning with DoRA enhances its effectiveness.
>
>
>
>     |  | Params (%) | BLEU ($\uparrow$) | METEOR ($\uparrow$) |
>     |---|---|---|---|
>     |  |  |  |  |
>     | DoRA | 0.3618 | 49.86 | 70.01 |
>     |  | 0.4025 | 51.22 | 70.40 |
>     |  | 0.4040 | 50.53 | 69.94 |
>     |  |  |  |  |
>     | SDDoRA | 0.3630 | 51.32 | 70.33 |
>     |  | 0.3633 | **51.55** | **70.80** |
>     |  | 0.3639 | 50.80 | 70.50 |

---

> ### Author Response · Authors · 2024-11-22
>
> > Q: Insufficient model architecture coverage.
>
> In response to the reviewer's feedback, we expanded our analysis beyond the deep S4 model and Mamba presented in the original paper. Specifically, we incorporated the Transformer-SSM hybrid model Jamba [2] (Jamba-Tiny-319M and Jamba-Mini-52B) and Mamba-II [3] (Mamba-II-130M and Mamba-II-1.4B).
> * **New Experiment Results on Jamba.**
>
>     We froze the Transformer layers, tuning only the Mamba layers, while adhering to the same experimental settings used for Mamba. To accommodate the Jamba-Tiny 52B model on a single 80GB GPU, we quantized all non-Mamba layers to 4-bit precision, following an approach similar to QLoRA, and reduced the batch size.
>
>     **Results.** The performance comparison between LoRA and SDLoRA is shown in the table below ((!) indicates still training). SDLoRA outperforms LoRA on nine out of eleven tasks, demonstrating that SDLoRA's strong performance on Mamba effectively transfers to hybrid models as well.
>     |  | Jamba-Tiny-319M |  |  |  |  |  |  |  |  |  |  | Jamba-Mini-52B |  |  |  |  |  |  |  |
>     |---|---|---|---|---|---|---|---|---|---|---|---|---|---|---|---|---|---|---|---|
>     |  |  |  | DART |  |  | SAMSum |  |  |  | Spider |  |  |  | DART |  |  | SAMSum |  |  |
>     |  | Params (%) |  | BLEU ($\uparrow$) | METEOR  ($\uparrow$) |  | R1 ($\uparrow$) | R2  ($\uparrow$) | RL  ($\uparrow$) |  | Acc. ($\uparrow$) |  | Params (%) |  | BLEU ($\uparrow$) | METEOR  ($\uparrow$) |  | R1   ($\uparrow$) | R2   ($\uparrow$) | RL   ($\uparrow$) |
>     |  |  |  |  |  |  |  |  |  |  |  |  |  |  |  |  |  |  |  |  |
>     | LoRA | 0.05030 |  | 27.17 | 65.03 |  | 37.13 | 16.43 | 30.90 |  | 35.49 |  | 0.004951 |  | 52.86 | 73 |  | 55.31 | 31.71 | 46.47 |
>     |  | 0.05690 |  | **39.02** | **67.90** |  | 40.80 | 18.54 | 33.87 |  | 44.07 |  | 0.005629 |  | 52.65 | 72.81 |  | 55.12 | 31.63 | 46.64 |
>     |  | 0.06153 |  | 23.18 | 65.05 |  | 39.15 | 17.70 | 32.79 |  | 37.67 |  | 0.006051 |  | 52.63 | 72.94 |  | 56.36 | 33.48 | 47.91 |
>     |  |  |  |  |  |  |  |  |  |  |  |  |  |  |  |  |  |  |  |  |
>     | SDLoRA | 0.05536 |  | 31.49 | 67.18 |  | 41.11 | 18.48 | 33.84 |  | 48.58 |  | 0.005484 |  | 51.86 (!) | 72.42 (!) |  | 56.08 | 32.79 | 47.61 |
>     |  | 0.05540 |  | 31.43 | 67.86 |  | 41.69 | 19.17 | 34.47 |  | **50.40** |  | 0.005488 |  | 52.79 (!) | **73.07** (!) |  | **56.53** | **33.5** | **47.96** |
>     |  | 0.05549 |  | 33.03 | 67.80 |  | **42.18** | **19.19** | **34.95** |  | 49.60 |  | 0.005497 |  | **53.11** | 72.95 |  | 56.14 | 33.08 | 47.56 |
> * **New Experiment Results on Mamba-II.**
>
>     For Mamba-II, however, applying SDLoRA is not straightforward because Mamba-II further constrains $A$ such that all (non-zero) entries must have the same value. Therefore, our original dimension selection approach cannot be directly applied here. We consider a naive extension of SDLoRA by selecting dimensions in the projection matrices for input mapping vector $B$ and the projection matrices for output mapping vector $C$ using their respective magnitude, and fine-tune the selected dimensions and all elements of state transition matrix $A$.
>
>     **Benchmarking LoRA on Different Layers**: We follow the same experimental setup used for Mamba-I and demonstrate that, on Mamba-II, our conclusion holds: LoRA is more effective on linear projection layers than on SSM modules.
>     | Params (%) | Target Layers | Method |  | DART (Mamba-II-130M) |  |  | Spider (Mamba-II-1.3B) |
>     |---|---|---|---|---|---|---|---|
>     |  |  |  |  | BLEU (↑) | METEOR (↑) |  | Acc. (↑) |
>     |  |  |  |  |  |  |  |  |
>     | < 1.0 | SSM Modules | LoRA |  | 40.1 | 64.2 |  | 54.1 |
>     |  | Linear Layers | LoRA |  | 43.0 | 67.1 |  | 57.9 |
>     |  |  |  |  |  |  |  |  |
>     | < 3.0 | Both | LoRA |  | 45.4 | 66.9 |  | 64.5 |
>
>     **Comparison between LoRA and SDLoRA**: The table below compares the performance of LoRA and SDLoRA on Mamba-II. The results demonstrate that SDLoRA consistently outperforms LoRA on Mamba-II models.
>
>     |  |  | DART (Mamba-II-130M) |  |  |  | SAMSum (Mamba-II-1.3B) |  |  |  |
>     |---|---|---|---|---|---|---|---|---|---|
>     |  |  | Params (%) | BLEU ($\uparrow$) | METEOR  ($\uparrow$) |  | Params (%) | R1   ($\uparrow$) | R2   ($\uparrow$) | RL   ($\uparrow$) |
>     | LoRA |  | 0.3354 | 48.09 | 68.71 |  | 0.1614 | 49.73 | 26.14 | 41.53 |
>     | SDLoRA |  | 0.3393 | **48.93** | **70.60** |  | 0.1767 | **50.72** | **27.21** | **42.54** |
>
> * **Standard Transformer model**: Most existing PEFT methods are designed specifically for Transformers, with their empirical results extensively documented in prior works. Our method, however, is tailored exclusively for SSMs, making it highly non-trivial to Transformer-only architectures (there is no definition of channels and states in Transformers). Nevertheless, our results on Jamba demonstrate that our approach performs effectively on hybrid Transformer-SSM models.
>
> The new results and discussion will be included in our final version.

---

> > ### Author Response · Authors · 2024-11-22
> >
> > > Q: Insufficient empirical results.
> >
> > In addition to the new experiments above, to fully address the reviewer's concern, we have conducted additional experiments on a new dataset: CelebA [4], a larger and more complex vision dataset compared to CIFAR-10. For the detailed experiment setting, please refer to `Major Update 2` in general response.
> >
> > **Results.** We conducted experiments on Mamba-130M, and the results are summarized below. The table demonstrates that SDLoRA consistently outperforms LoRA across tasks of varying difficulty levels.
> >
> > |  | Params (%) | Average Acc. (All) | Overall Acc. (Easy) | Overall Acc. (Medium) | Overall Acc. (Hard) |
> > |---|---|---|---|---|---|
> > |  |  |  |  |  |  |
> > | LoRA | .3178 | 87.79 | 58.53 | 24.19 | 4.18 |
> > |  | .3600 | 88.58 | 60.10 | 26.21 | 5.19 |
> > |  | .3883 | 87.67 | 58.32 | 24.01 | 4.08 |
> > |  |  |  |  |  |  |
> > | SDLoRA | .3492 | **88.61** | 60.50 | 26.27 | **5.40** |
> > |  | .3498 | 88.40 | 59.75 | 25.69 | 5.01 |
> > |  | .3509 | 88.50 | **60.52** | **26.30** | 4.96 |
> >
> >
> >
> > > Q: Limited theoretical analysis: (i) the analysis on S4 doesn't clearly extend to S6, (ii) it is unclear why certain adapters perform differently on SSMs and Transformers.
> >
> > Thank you for the thoughtful question. We agree with the reviewer that our theoretical analysis has certain limitations. We will incorporate a discussion of these limitations in the revised manuscript, reflecting the reviewer’s valuable feedback.
> >
> > However, we want to kindly emphasize that (i) regarding the first point, we have theoretically proven that applying LoRA to linear projection matrices is effective, and this analysis is conducted on S6. We believe this finding is insightful; (ii) as for the second point, we have analyzed why prompt-based methods does not perform well on SSMs and have included these results.
> >
> > ***
> >
> > **Final Note:** Thank you for finding our work valuable and detailed. We recognize that our work could benefit from further empirical studies. However, we believe our research serves as a pioneering study in the PEFT of SSMs, and we hope that our new results, highlighting SDLoRA's superior performance with DoRA, Jamba, Mamba-II and CelebA, will positively shape your view of our work. More analysis can be found in our general response (`Major Update 4`), and please reach out if you have any questions. If our responses have adequately addressed your concerns, we would greatly appreciate if you would consider raising your score and supporting our paper's acceptance.
> >
> > *References:*
> >
> > [1] DoRA: Weight-Decomposed Low-Rank Adaptation
> > [2] Jamba: A Hybrid Transformer-Mamba Language Model
> > [3] Transformers are SSMs: Generalized Models and Efficient Algorithms Through Structured State Space Duality
> > [4] Deep Learning Face Attributes in the Wild

---

> > > ### Author Response · Authors · 2024-11-26
> > > **Uploaded revised PDF**
> > >
> > > Dear reviewer, we’ve updated the PDF file, highlighting the new changes in blue. We’ll update the draft once more before the deadline with ongoing experiment results. We appreciate any feedback to see if our rebuttal addresses your concerns.

---

> > > > ### Author Response · Authors · 2024-11-27
> > > > **New Experiments (Round 2)**
> > > >
> > > > Dear reviewer, we are writing to provide an update with additional results.
> > > >
> > > > 1.  Evaluation on a New Dataset (GLUE) with Mamba-II
> > > >
> > > >     SDLoRA outperforms LoRA on Mamba-II-130M for the GLUE dataset. In our last response, we conducted experiments with SDLoRA on DART and SAMSum datasets using Mamba-II-130M, and now we have extended the evaluation to a new dataset, GLUE. Our findings indicate that SDLoRA with Mamba-II-130M consistently outperforms LoRA across all GLUE tasks (note that CoLA is still training).
> > > >
> > > >     | Accuracy ($\uparrow$) | Params (%) | RTE | MRPC | COLA | SST2 | QNLI | QQP | MNLI |
> > > >     |---|---|---|---|---|---|---|---|---|
> > > >     | LoRA | 0.3354 | 63.4 | 80.9 | - | 89.1 | 85.3 | 87.1 | 78.6 |
> > > >     | SDLoRA | 0.3393 | **64.3** | **82.3** | - | **94.1** | **87.0** | **88.3** | **81.1** |
> > > >
> > > > 2. Introduction of a New LoRA Variant — LoRA+
> > > >
> > > >     SDLoRA+ consistently outperforms LoRA+ across different model architectures and datasets.
> > > >
> > > >     |  |  | Mamba-I-130M |  |  | Mamba-II-130M |  |  | Mamba-II-1.3B |  |  |  |  |
> > > >     |---|---|---|---|---|---|---|---|---|---|---|---|---|
> > > >     |  |  | DART |  |  | DART |  |  | SAMSum |  |  |  | Spider |
> > > >     |  |  | BLEU ($\uparrow$) | METEOR  ($\uparrow$) |  | BLEU ($\uparrow$) | METEOR  ($\uparrow$) |  | R1   ($\uparrow$) | R2   ($\uparrow$) | RL   ($\uparrow$) |  | Acc. ($\uparrow$) |
> > > >     | LoRA+ |  | 50.91 | 70.06 |  | 49.14 | 69.78 |  | 49.83 | 26.09 | 41.66 |  | 73.75 |
> > > >     | SDLoRA+ |  | **51.93** | **70.58** |  | **49.99** | **70.48** |  | **50.81** | **27.19** | **42.40** |  | **84.22** |
> > > >
> > > > We understand that you must be busy and highly appreciate your time. We have made every effort to address your concerns and would be grateful if you could review our response at your earliest convenience. Please let us know if all your concerns have been adequately addressed. If they have, we kindly ask you to consider increasing your score in support of our paper's acceptance.

---

> > > > > ### Comment · Reviewer_1UQC · 2024-11-27
> > > > > **Overall Thoughts**
> > > > >
> > > > > While I do think PEFT + Mamba is under explored. I really appreciate the additional experiments and responses provided by the authors, I, however, do not think that the paper in its current form is quite strong enough, even with the additional results.
> > > > >
> > > > > I don't think PEFT experiments alone are enough. The SD LoRa is a very minor change and more akin to a hyperparameter.
> > > > >
> > > > > I'll raise my contribution subscore to a 3 because of the additional experiments, but not my overall score.
> > > > >
> > > > > Again thank you for your responses, your work, and your additional experiments

---

### Official Review · Reviewer_mTUE · 2024-11-04

**Soundness:** 3
**Presentation:** 3
**Contribution:** 3
**Rating:** 6
**Confidence:** 3

**Summary:**

The paper explores parameter-efficient fine-tuning (PEFT) methods for deep State Space Models (SSMs), especially in the context of language modeling tasks. It investigates the effectiveness of various PEFT methods, such as prompt-based prefix-tuning and Low-Rank Adaptation (LoRA), applied to SSM architectures like Mamba. A new variant called SDLoRA (Selective Dimension LoRA) is proposed in this paper to selectively update channels and states in the SSM modules, aiming to enhance the fine-tuning performance while reducing parameters. The results indicate that SDLoRA outperforms conventional LoRA when fine-tuning SSM-based models.

**Strengths:**

(1) Efficiency and Scalability: By focusing on selective parameter updates, the SDLoRA method enhances computational efficiency, which is crucial for large-scale models. Experimental results show that SDLoRA consistently outperforms traditional LoRA across several benchmarks, proving its efficacy in SSM architectures.
(2) Adaptability: The proposed SDLoRA method demonstrates adaptability across multiple tasks, including NLP tasks and vision tasks

**Weaknesses:**

(1) Limited Applicability Beyond SSMs: The focus on SSMs means SDLoRA may not generalize well to non-SSM architectures or hybrid models such as Transformer models or Transformer-SSM combinations. Its broader applicability to other architectures remains untested.
(2) Parameter Selection: The dimension selection process in SDLoRA relies on parameter magnitudes, which may not be optimal and could benefit from a more sophisticated selection algorithm. And what if the magnitude of each channel changes during the fine-tuning stage?

**Questions:**

See the weaknesses

---

> ### Author Response · Authors · 2024-11-22
>
> We appreciate the reviewer’s encouraging feedback, especially for recognizing that our method is efficient, scalable, and adaptable.
>
> ***
>
> >  Q: Limited Applicability Beyond SSMs: The focus on SSMs means SDLoRA may not generalize well to non-SSM architectures or hybrid models such as Transformer models or Transformer-SSM combinations. Its broader applicability to other architectures remains untested.
>
> Key finding: SDLoRA's strong performance on Mamba effectively transfers to hybrid models as well.
>
> Thank you for the thoughtful suggestion. Following your recommendation, we implemented both LoRA and SDLoRA on the Jamba [1] model series, evaluating two configurations with 319M and 52B parameters, respectively. We froze the Transformer layers, tuning only the Mamba layers, while adhering to the same experimental settings used for Mamba. To accommodate the Jamba-Tiny 52B model on a single 80GB GPU, we quantized all non-Mamba layers to 4-bit precision, following an approach similar to QLoRA, and reduced the batch size.
>
> **Results.** The performance comparison between LoRA and SDLoRA is shown in the table below ((!) indicates still training). SDLoRA outperforms LoRA on nine out of eleven tasks, demonstrating that SDLoRA's strong performance on Mamba effectively transfers to hybrid models as well.
>
>
>
> |  | Jamba-Tiny-319M |  |  |  |  |  |  |  |  |  |  | Jamba-Mini-52B |  |  |  |  |  |  |  |
> |---|---|---|---|---|---|---|---|---|---|---|---|---|---|---|---|---|---|---|---|
> |  |  |  | DART |  |  | SAMSum |  |  |  | Spider |  |  |  | DART |  |  | SAMSum |  |  |
> |  | Params (%) |  | BLEU ($\uparrow$) | METEOR  ($\uparrow$) |  | R1 ($\uparrow$) | R2  ($\uparrow$) | RL  ($\uparrow$) |  | Acc. ($\uparrow$) |  | Params (%) |  | BLEU ($\uparrow$) | METEOR  ($\uparrow$) |  | R1   ($\uparrow$) | R2   ($\uparrow$) | RL   ($\uparrow$) |
> |  |  |  |  |  |  |  |  |  |  |  |  |  |  |  |  |  |  |  |  |
> | LoRA | 0.05030 |  | 27.17 | 65.03 |  | 37.13 | 16.43 | 30.90 |  | 35.49 |  | 0.004951 |  | 52.86 | 73 |  | 55.31 | 31.71 | 46.47 |
> |  | 0.05690 |  | **39.02** | **67.90** |  | 40.80 | 18.54 | 33.87 |  | 44.07 |  | 0.005629 |  | 52.65 | 72.81 |  | 55.12 | 31.63 | 46.64 |
> |  | 0.06153 |  | 23.18 | 65.05 |  | 39.15 | 17.70 | 32.79 |  | 37.67 |  | 0.006051 |  | 52.63 | 72.94 |  | 56.36 | 33.48 | 47.91 |
> |  |  |  |  |  |  |  |  |  |  |  |  |  |  |  |  |  |  |  |  |
> | SDLoRA | 0.05536 |  | 31.49 | 67.18 |  | 41.11 | 18.48 | 33.84 |  | 48.58 |  | 0.005484 |  | 51.86 (!) | 72.42 (!) |  | 56.08 | 32.79 | 47.61 |
> |  | 0.05540 |  | 31.43 | 67.86 |  | 41.69 | 19.17 | 34.47 |  | **50.40** |  | 0.005488 |  | 52.79 (!) | **73.07** (!) |  | **56.53** | **33.5** | **47.96** |
> |  | 0.05549 |  | 33.03 | 67.80 |  | **42.18** | **19.19** | **34.95** |  | 49.60 |  | 0.005497 |  | **53.11** | 72.95 |  | 56.14 | 33.08 | 47.56 |

---

> > ### Author Response · Authors · 2024-11-22
> >
> > > Q: Parameter Selection: The dimension selection process in SDLoRA relies on parameter magnitudes, which may not be optimal and could benefit from a more sophisticated selection algorithm. And what if the magnitude of each channel changes during the fine-tuning stage?
> >
> >
> > Thank you for this insightful question. Our current magnitude-based dimension selection method is designed for efficiency but has room for improvement. In fact, we have explored alternative methods.
> >
> > **Experimental Setup**: To compare our method with alternative dimension selection methods, we established a ranking of all sets of updatable channels and states by brute-forcing all channel and state combinations in a 2-layer frozen deep S4 model (state dimension = 2, model dimension = 4) using a dataset generated by a 1-layer target deep S4 model (state dimension = 1). Rankings were based on the final approximation error, and we evaluated each method by examining how well its selected dimensions ranked.
> >
> > **Methods Compared**:
> >
> > * Magnitude-based (used in our paper): Channels and states were chosen based on parameter magnitude changes during the warmup stage.
> > * Loss-based: Channels and states were individually updated, and selections were made based on their impact on loss.
> >
> >
> > **Results**: The loss-based method significantly improved the rank of selected dimensions, achieving a 52.22% improvement compared to the magnitude-based approach.
> >
> > **Discussion**: Despite its improved dimension selection performance, the loss-based approach is computationally expensive. For example, on Mamba-130M, processing one batch (size 4) would take over 16 hours on a single A100 GPU. This limitation reinforces our decision to use the magnitude-based method while identifying efficient and more effective dimension selection as an avenue for future work.
> >
> > As per the request of the reviewer, we will include the discussion above in our final version.
> >
> > ***
> >
> > **Final Note:** Thank you for your valuable comments. We are grateful to hear that you found our method to be efficient, scalable, and adaptable. If there are any remaining questions, please do not hesitate to let us know. Assuming our responses have satisfactorily addressed your concerns, we kindly ask you to consider enhancing your score and supporting our paper.
> >
> > *References:*
> >
> > [1] Jamba: A Hybrid Transformer-Mamba Language Model

---

> > > ### Author Response · Authors · 2024-11-26
> > > **Uploaded revised PDF**
> > >
> > > Dear reviewer, we’ve updated the PDF file, highlighting the new changes in blue. We’ll update the draft once more before the deadline with ongoing experiment results. We appreciate any feedback to see if our rebuttal addresses your concerns.

---

> > > > ### Author Response · Authors · 2024-11-27
> > > > **New Experiment (Round 2)**
> > > >
> > > > Dear reviewer, we are writing to provide an update with additional results.
> > > >
> > > > 1.  Evaluation on a New Dataset (GLUE) with Mamba-II
> > > >
> > > >     SDLoRA outperforms LoRA on Mamba-II-130M for the GLUE dataset. In our general response, we conducted experiments with SDLoRA on DART and SAMSum datasets using Mamba-II-130M, and now we have extended the evaluation to a new dataset, GLUE. Our findings indicate that SDLoRA with Mamba-II-130M consistently outperforms LoRA across all GLUE tasks (note that CoLA is still training).
> > > >
> > > >     | Accuracy ($\uparrow$) | Params (%) | RTE | MRPC | COLA | SST2 | QNLI | QQP | MNLI |
> > > >     |---|---|---|---|---|---|---|---|---|
> > > >     | LoRA | 0.3354 | 63.4 | 80.9 | - | 89.1 | 85.3 | 87.1 | 78.6 |
> > > >     | SDLoRA | 0.3393 | **64.3** | **82.3** | - | **94.1** | **87.0** | **88.3** | **81.1** |
> > > >
> > > > 2. Introduction of a New LoRA Variant — LoRA+
> > > >
> > > >     SDLoRA+ consistently outperforms LoRA+ across different model architectures and datasets.
> > > >
> > > >     |  |  | Mamba-I-130M |  |  | Mamba-II-130M |  |  | Mamba-II-1.3B |  |  |  |  |
> > > >     |---|---|---|---|---|---|---|---|---|---|---|---|---|
> > > >     |  |  | DART |  |  | DART |  |  | SAMSum |  |  |  | Spider |
> > > >     |  |  | BLEU ($\uparrow$) | METEOR  ($\uparrow$) |  | BLEU ($\uparrow$) | METEOR  ($\uparrow$) |  | R1   ($\uparrow$) | R2   ($\uparrow$) | RL   ($\uparrow$) |  | Acc. ($\uparrow$) |
> > > >     | LoRA+ |  | 50.91 | 70.06 |  | 49.14 | 69.78 |  | 49.83 | 26.09 | 41.66 |  | 73.75 |
> > > >     | SDLoRA+ |  | **51.93** | **70.58** |  | **49.99** | **70.48** |  | **50.81** | **27.19** | **42.40** |  | **84.22** |
> > > >
> > > > We understand that you must be busy and highly appreciate your time. We have made every effort to address your concerns and would be grateful if you could review our response at your earliest convenience. Please let us know if all your concerns have been adequately addressed. If they have, we kindly ask you to consider increasing your score in support of our paper's acceptance.

---

### Official Review · Reviewer_n23z · 2024-11-05

**Soundness:** 3
**Presentation:** 3
**Contribution:** 3
**Rating:** 5
**Confidence:** 4

**Summary:**

This paper presents the first study on the performance of PEFT methods applied to SSM-based models. Specifically, prompt-based and parameter-based methods are involved. With theoretical analysis and extensive experiments, LoRA tends to achieve better performance. To further improve the performance, this paper introduces SDLoRA, which selectively updates certain channels and states on SSM modules while applying LoRA to linear projection matrices.

**Strengths:**

1.	Reasonable theoretical analysis and comprehensive experiments.
2.	The introduced SDLoRA is novel and effective.
3.	Through extensive experiments, the findings in this paper is useful and inspired.

**Weaknesses:**

1.	The speed of SDLoRA is nor reported.
2.	Experimental results on larger datasets are needed.

**Questions:**

1.	During the process of selective dimension tuning, the authors select the target channels and states based on magnitude, any other metrics have been tried?
2.	Will SDLoRA's training speed be slower compared to vanilla LoRA? How much slower will it be?
3.	What is the accuracy of SDLoRA on a larger data set, such as ImageNet?
4.	Some other advanced parameter-efficient tuning method like DoRA [1] can be adapted to Mamba? or the proposed SDLoRA can be adapted to Jamba?

[1] DoRA: Weight-Decomposed Low-Rank Adaptation

---

> ### Author Response · Authors · 2024-11-22
>
> We thank the reviewer for the encouraging feedback, especially for recognizing that (i) our paper is useful and inspired, (ii) our paper provides theoretical analysis with comprehensive and extensive experiments, and (iii) our method is novel and effective.
>
> ***
> > Q: Will SDLoRA's training speed be slower compared to vanilla LoRA? How much slower will it be?
>
> Thank you for raising this important question. In response, we conducted a new experiment to compare the training speeds of SDLoRA and LoRA. The key finding is that the SDLoRA is slightly *faster* than LoRA.
>
> To assess the runtime of SDLoRA and LoRA, we conducted experiments on four different models, including both SSM and hybrid architectures. Unless specified otherwise, for each model and method, dataset were generated with 2,500 batches of data samples, each batch comprising a random sequence of 1,500 tokens. The simulation was repeated four times, including dataset generation. All experiments were carried out on a single H100 GPU, and the reported metrics represent averages across the four simulations. Consistent with our previous experiments, we used the original hyperparameter settings, ensuring that SDLoRA included more trainable parameters than LoRA.
>
> Fine-tuning with SDLoRA consists of two stages: (1) dimension selection and (2) standard training. In this study, we first compare the runtime of SDLoRA and LoRA during stage 2 (training) and then evaluate the additional runtime introduced by SDLoRA during stage 1 (dimension selection). Our results show that the dimension selection stage adds only marginal runtime overhead, and SDLoRA is more efficient than LoRA in standard training.
>
> * **Training**: When the channels and states have been selected, the training of SDLoRA is *faster* than LoRA when the same number of trainable parameters are considered.
>
>
>     The runtimes are reported in the table below. We observe that, despite having more trainable parameters, SDLoRA is faster than LoRA. We attribute this to the fact that LoRA introduces additional FLOPs due to the extra matrix multiplication operations required for each update (specifically, the multiplication of two low-rank matrices).
>
>     | Avg. Runtime (Seconds) | Mamba-130M | Mamba-1.4B | Jamba-Tiny-319M | Jamba-Mini-52B |
>     |---------------------|------------|------------|----------------|----------------|
>     | LoRA | 410.0 $\pm$  80.0 | 2060.0 $\pm$  135.0 | 352.5 $\pm$  107.5 | 3427.5 $\pm$  185.0 |
>     | SDLoRA | **330.0 $\pm$  77.5** | **1697.5 $\pm$  87.5** | **257.5 $\pm$  72.5** | **3065.0 $\pm$ 232.5** |
>
> * **Dimension Selection**: For dimension selection, our method first performs an *Initial Subset Training*, and then selects the dimensions based on the *magnitude of parameter changes* across different dimensions.
>
>     1. **Initial Subset Training**: We update the model by going through only a subset of the dataset (e.g., 3% of batches in DART experiments), which is sufficient in practice.
>     2. **Magnitude-Based Dimension Selection**: After the subset training, we select dimensions based on the magnitude of parameter changes observed.
>
>     In this experiment, we simulate a real scenario using datasets with 2,500 batches, considering a small subset containing 125 batches (5% of the full dataset). We repeat the experiments 80 times, and the reported numbers are averaged across these simulations. The following table presents the runtime analysis of the dimension selection stage in SDLoRA.
>
>
>     | Avg. Runtime (Seconds) | Mamba-130M | Mamba-1.4B | Jamba-Tiny-319M | Jamba-Mini-52B |
>     |---|---|---|---|---|
>     | Initial Subset Training | 16.250 $\pm$ 3.880 | 85.250 $\pm$ 5.130 | 15.750 $\pm$ 1.000 | 163.630 $\pm$ 10.120 |
>     | Magnitude-Based Dimension Selection | 0.280 $\pm$ 0.000 | 0.520 $\pm$ 0.120 | 0.090 $\pm$ 0.000 | 0.240 $\pm$ 0.040 |
>     | Total Time | 16.530 $\pm$ 3.880 | 85.770 $\pm$ 5.250 | 15.840 $\pm$ 1.000 | 163.870 $\pm$ 10.160 |
>     | |
>     | Proportion of Training 1 Epoch | 0.050$\times$ | 0.051$\times$ | 0.062$\times$ | 0.053$\times$ |
>     | Proportion of Training 5 Epoch | **0.010$\times$** | **0.010$\times$** | **0.012$\times$** | **0.011$\times$** |
>
>     This table demonstrates that the dimension selection stage adds only *negligible* runtime.

---

> > ### Author Response · Authors · 2024-11-22
> >
> > > Q: What is the accuracy of SDLoRA on a larger data set, such as ImageNet?
> >
> > Thank you for your valuable suggestion. The immense size and lengthy training time required for ImageNet made direct evaluation impractical. Instead, we chose the CelebA [1] dataset, which comprises 202,599 face images (178 $\times$ 218 pixels).
> >
> > This dataset is significantly larger than CIFAR-10, used in the original paper, and contains 40 classification tasks (e.g., predicting attributes like gender, hair color, and glasses). We report four metrics: (i) average accuracy and overall accuracy for (ii) easy, (iii) medium, and (iv) hard tasks. Here, overall accuracy refers to the accuracy of correctly predicting all target labels within a specific subset of tasks. Tasks are categorized as easy, medium, or hard by clustering based on average performance. To ensure computational feasibility, we reduced the resolution by using [InsightFace](https://github.com/deepinsight/insightface) to crop the images to retain only the face and then resized them to 32 × 32 pixels. This preprocessing helps maintain a manageable sequence length for efficient runtime.
> >
> > **Results.** We conducted experiments on Mamba-130M, and the results are summarized below. The table demonstrates that SDLoRA consistently outperforms LoRA across tasks of varying difficulty levels.
> >
> > |  | Params (%) | Average Acc. (All) | Overall Acc. (Easy) | Overall Acc. (Medium) | Overall Acc. (Hard) |
> > |---|---|---|---|---|---|
> > |  |  |  |  |  |  |
> > | LoRA | .3178 | 87.79 | 58.53 | 24.19 | 4.18 |
> > |  | .3600 | 88.58 | 60.10 | 26.21 | 5.19 |
> > |  | .3883 | 87.67 | 58.32 | 24.01 | 4.08 |
> > |  |  |  |  |  |  |
> > | SDLoRA | .3492 | **88.61** | 60.50 | 26.27 | **5.40** |
> > |  | .3498 | 88.40 | 59.75 | 25.69 | 5.01 |
> > |  | .3509 | 88.50 | **60.52** | **26.30** | 4.96 |
> >
> >
> >
> > > Q: Some other advanced parameter-efficient tuning method like DoRA can be adapted to Mamba?
> >
> >
> > * **Benchmarking DoRA**: applying DoRA [2] to linear projection matrices demonstrates greater effectiveness compared to its application to SSM modules.
> >
> >     Based on your great suggestion, we evaluate the performance of DoRA on the DART dataset using Mamba-130M and on the Spider dataset using Mamba-1.4B. The results are summarized in the table below.
> >
> >
> >
> >     | Params (%) | Target Layers | Method |  | DART (Mamba-130M) |  |  | Spider (Mamba-1.4B) |
> >     |---|---|---|---|---|---|---|---|
> >     |  |  |  |  | BLEU (↑) | METEOR (↑) |  | Acc. (↑) |
> >     |  |  |  |  |  |  |  |  |
> >     | < 0.4 | SSM Modules | LoRA |  | 47.05 | 68.86 |  | 58.03 |
> >     |  |  | DoRA |  | 47.07 | 68.79 |  | 55.32 |
> >     |  |  |  |  |  |  |  |  |
> >     | < 0.4 | Linear Layers | LoRA |  | 48.86 | 70.25 |  | 61.80 |
> >     |  |  | DoRA |  | 49.93 | 70.81 |  | 61.32 |
> >     |  |  |  |  |  |  |  |  |
> >     | < 3.0 | Both | LoRA |  | 49.52 | 70.97 |  | 56.38 |
> >     |  |  | DoRA |  | 51.36 | 70.94 |  | 55.71 |
> >
> >
> >
> >
> >
> >     Our findings are consistent with observations seen in LoRA: applying DoRA to linear projection matrices proves more effective than its application to SSM modules. Interestingly, applying DoRA to SSM modules not only offers limited benefits but, in some cases, even degrades performance. This is particularly evident on Spider dataset, when comparing the configurations of applying DoRA to both linear projection matrices and SSM modules versus solely targeting linear projection matrices.
> >
> > * **Integrating Selective Dimension Tuning with DoRA (SDDoRA)**: Incorporating selective dimension tuning into DoRA achieves superior performance compared to using DoRA alone.
> >
> >     We extended our investigation to include SDDoRA and evaluated its performance against DoRA alone using the DART benchmark on the Mamba-130M model. The results, presented below, show that integrating selective dimension tuning with DoRA enhances its effectiveness.
> >
> >     |  | Params (%) | BLEU ($\uparrow$) | METEOR ($\uparrow$) |
> >     |---|---|---|---|
> >     |  |  |  |  |
> >     | DoRA | 0.3618 | 49.86 | 70.01 |
> >     |  | 0.4025 | 51.22 | 70.40 |
> >     |  | 0.4040 | 50.53 | 69.94 |
> >     |  |  |  |  |
> >     | SDDoRA | 0.3630 | 51.32 | 70.33 |
> >     |  | 0.3633 | **51.55** | **70.80** |
> >     |  | 0.3639 | 50.80 | 70.50 |

---

> ### Author Response · Authors · 2024-11-22
>
> > Q: Can proposed SDLoRA be adapted to Jamba?
>
> Thank you for the thoughtful suggestion. Following your recommendation, we implemented both LoRA and SDLoRA on the Jamba [3] model series, evaluating two configurations with 319M and 52B parameters, respectively.
>
>
> We froze the Transformer layers, tuning only the Mamba layers, while adhering to the same experimental settings used for Mamba. To accommodate the Jamba-Tiny 52B model on a single 80GB GPU, we quantized all non-Mamba layers to 4-bit precision, following an approach similar to QLoRA, and reduced the batch size.
>
> **Results.** The performance comparison between LoRA and SDLoRA is shown in the table below ((!) indicates still training). SDLoRA outperforms LoRA on nine out of eleven tasks, demonstrating that SDLoRA's strong performance on Mamba effectively transfers to hybrid models as well.
>
>
> |  | Jamba-Tiny-319M |  |  |  |  |  |  |  |  |  |  | Jamba-Mini-52B |  |  |  |  |  |  |  |
> |---|---|---|---|---|---|---|---|---|---|---|---|---|---|---|---|---|---|---|---|
> |  |  |  | DART |  |  | SAMSum |  |  |  | Spider |  |  |  | DART |  |  | SAMSum |  |  |
> |  | Params (%) |  | BLEU ($\uparrow$) | METEOR  ($\uparrow$) |  | R1 ($\uparrow$) | R2  ($\uparrow$) | RL  ($\uparrow$) |  | Acc. ($\uparrow$) |  | Params (%) |  | BLEU ($\uparrow$) | METEOR  ($\uparrow$) |  | R1   ($\uparrow$) | R2   ($\uparrow$) | RL   ($\uparrow$) |
> |  |  |  |  |  |  |  |  |  |  |  |  |  |  |  |  |  |  |  |  |
> | LoRA | 0.05030 |  | 27.17 | 65.03 |  | 37.13 | 16.43 | 30.90 |  | 35.49 |  | 0.004951 |  | 52.86 | 73 |  | 55.31 | 31.71 | 46.47 |
> |  | 0.05690 |  | **39.02** | **67.90** |  | 40.80 | 18.54 | 33.87 |  | 44.07 |  | 0.005629 |  | 52.65 | 72.81 |  | 55.12 | 31.63 | 46.64 |
> |  | 0.06153 |  | 23.18 | 65.05 |  | 39.15 | 17.70 | 32.79 |  | 37.67 |  | 0.006051 |  | 52.63 | 72.94 |  | 56.36 | 33.48 | 47.91 |
> |  |  |  |  |  |  |  |  |  |  |  |  |  |  |  |  |  |  |  |  |
> | SDLoRA | 0.05536 |  | 31.49 | 67.18 |  | 41.11 | 18.48 | 33.84 |  | 48.58 |  | 0.005484 |  | 51.86 (!) | 72.42 (!) |  | 56.08 | 32.79 | 47.61 |
> |  | 0.05540 |  | 31.43 | 67.86 |  | 41.69 | 19.17 | 34.47 |  | **50.40** |  | 0.005488 |  | 52.79 (!) | **73.07** (!) |  | **56.53** | **33.5** | **47.96** |
> |  | 0.05549 |  | 33.03 | 67.80 |  | **42.18** | **19.19** | **34.95** |  | 49.60 |  | 0.005497 |  | **53.11** | 72.95 |  | 56.14 | 33.08 | 47.56 |
>
> > Q: During the process of selective dimension tuning, the authors select the target channels and states based on magnitude, any other metrics have been tried?
>
> Thank you for this insightful question. Our current magnitude-based dimension selection method is designed for efficiency but has room for improvement. In fact, we have explored alternative methods.
>
> **Experimental Setup**: To compare our method with alternative dimension selection methods, we established a ranking of all sets of updatable channels and states by brute-forcing all channel and state combinations in a 2-layer frozen deep S4 model (state dimension = 2, model dimension = 4) using a dataset generated by a 1-layer target deep S4 model (state dimension = 1). Rankings were based on the final approximation error, and we evaluated each method by examining how well its selected dimensions ranked.
>
> **Methods Compared**:
>
> * Magnitude-based (used in our paper): Channels and states were chosen based on parameter magnitude changes during the warmup stage.
> * Loss-based: Channels and states were individually updated, and selections were made based on their impact on loss.
>
>
> **Results**: The loss-based method significantly improved the rank of selected dimensions, achieving a 52.22% improvement compared to the magnitude-based approach.
>
> **Discussion**: Despite its improved dimension selection performance, the loss-based approach is computationally expensive. For example, on Mamba-130M, processing one batch (size 4) would take over 16 hours on a single A100 GPU. This limitation reinforces our decision to use the magnitude-based method while identifying efficient and more effective dimension selection as an avenue for future work.
>
> Following the reviewer's request, we will incorporate the aforementioned discussion into our final version.
>
>
> ***
>
> **Final Note:** Thank you for your detailed comments. We are delighted to hear that you found our method to be novel and effective. If there are any remaining questions, please do not hesitate to let us know. If our responses have resolved your concerns, we kindly request you to consider increasing your score and support the acceptance of our paper.
>
> *References:*
>
> [1] Deep Learning Face Attributes in the Wild
> [2] DoRA: Weight-Decomposed Low-Rank Adaptation
> [3] Jamba: A Hybrid Transformer-Mamba Language Model

---

> > ### Author Response · Authors · 2024-11-26
> > **Uploaded revised PDF**
> >
> > Dear reviewer, we’ve updated the PDF file, highlighting the new changes in blue. We’ll update the draft once more before the deadline with ongoing experiment results. We appreciate any feedback to see if our rebuttal addresses your concerns.

---

> ### Author Response · Authors · 2024-11-27
> **New Experiments (Round 2)**
>
> Dear reviewer, we are writing to provide an update with additional results.
>
> 1.  Evaluation on a New Dataset (GLUE) with Mamba-II
>
>     SDLoRA outperforms LoRA on Mamba-II-130M for the GLUE dataset. In our general response, we conducted experiments with SDLoRA on DART and SAMSum datasets using Mamba-II-130M, and now we have extended the evaluation to a new dataset, GLUE. Our findings indicate that SDLoRA with Mamba-II-130M consistently outperforms LoRA across all GLUE tasks (note that CoLA is still training).
>
>     | Accuracy ($\uparrow$) | Params (%) | RTE | MRPC | COLA | SST2 | QNLI | QQP | MNLI |
>     |---|---|---|---|---|---|---|---|---|
>     | LoRA | 0.3354 | 63.4 | 80.9 | - | 89.1 | 85.3 | 87.1 | 78.6 |
>     | SDLoRA | 0.3393 | **64.3** | **82.3** | - | **94.1** | **87.0** | **88.3** | **81.1** |
>
> 2. Introduction of a New LoRA Variant — LoRA+
>
>     SDLoRA+ consistently outperforms LoRA+ across different model architectures and datasets.
>
>     |  |  | Mamba-I-130M |  |  | Mamba-II-130M |  |  | Mamba-II-1.3B |  |  |  |  |
>     |---|---|---|---|---|---|---|---|---|---|---|---|---|
>     |  |  | DART |  |  | DART |  |  | SAMSum |  |  |  | Spider |
>     |  |  | BLEU ($\uparrow$) | METEOR  ($\uparrow$) |  | BLEU ($\uparrow$) | METEOR  ($\uparrow$) |  | R1   ($\uparrow$) | R2   ($\uparrow$) | RL   ($\uparrow$) |  | Acc. ($\uparrow$) |
>     | LoRA+ |  | 50.91 | 70.06 |  | 49.14 | 69.78 |  | 49.83 | 26.09 | 41.66 |  | 73.75 |
>     | SDLoRA+ |  | **51.93** | **70.58** |  | **49.99** | **70.48** |  | **50.81** | **27.19** | **42.40** |  | **84.22** |
>
> We understand that you must be busy and highly appreciate your time. We have made every effort to address your concerns and would be grateful if you could review our response at your earliest convenience. Please let us know if all your concerns have been adequately addressed. If they have, we kindly ask you to consider increasing your score in support of our paper's acceptance.

---

### Author Response · Authors · 2024-11-22
**To AC and All Reviewers**

We would like to thank all reviewers for their comments and helpful feedback. We are particularly encouraged that the reviewers have found that (i) our paper is clear (`R-AtCk`), useful and inspired (`R-n23z`), and studying what definitely needs to be studied (`R-1UQC`), (ii) our paper provides theoretical (`R-n23z`, `R-1UQC`), empirical (`R-1UQC`), systematic (`R-AtCk`), and detailed (`R-1UQC`) analysis with comprehensive (`R-n23z`) and extensive (`R-n23z`) experiments across various datasets (`R-AtCk`), and (iii) our method is novel (`R-n23z`, `R-AtCk`), innovative (`R-AtCk`), effective (`R-n23z`), efficient (`R-mTUE`), scalable (`R-mTUE`) and adaptable (`R-mTUE`).

In response to the feedback, we have addressed each concern, added experimental results, and will update our paper accordingly. Below, we summarize the major updates in our rebuttal.


# Major Update 1: Expanded Model Coverage — Jamba & Mamba-II.
**Key Insight: SDLoRA outperforms LoRA on Jamba [1] and Mamba-II [2] models.**


In response to the reviewers' feedback, we expanded our analysis beyond the deep S4 model and Mamba presented in the original paper. Specifically, we incorporated the Transformer-SSM hybrid model Jamba (Jamba-Tiny-319M and Jamba-Mini-52B) and Mamba-II (Mamba-II-130M and Mamba-II-1.4B).

> New Experiment Results on Jamba.

We froze the Transformer layers, tuning only the Mamba layers, while adhering to the same experimental settings used for Mamba. To accommodate the Jamba-Tiny 52B model on a single 80GB GPU, we quantized all non-Mamba layers to 4-bit precision, following an approach similar to QLoRA, and reduced the batch size.

**Results.** The performance comparison between LoRA and SDLoRA is shown in the table below ((!) indicates still training).  SDLoRA outperforms LoRA on nine out of eleven tasks, demonstrating that SDLoRA's strong performance on Mamba effectively transfers to hybrid models as well.


|  | Jamba-Tiny-319M |  |  |  |  |  |  |  |  |  |  | Jamba-Mini-52B |  |  |  |  |  |  |  |
|---|---|---|---|---|---|---|---|---|---|---|---|---|---|---|---|---|---|---|---|
|  |  |  | DART |  |  | SAMSum |  |  |  | Spider |  |  |  | DART |  |  | SAMSum |  |  |
|  | Params (%) |  | BLEU ($\uparrow$) | METEOR  ($\uparrow$) |  | R1 ($\uparrow$) | R2  ($\uparrow$) | RL  ($\uparrow$) |  | Acc. ($\uparrow$) |  | Params (%) |  | BLEU ($\uparrow$) | METEOR  ($\uparrow$) |  | R1   ($\uparrow$) | R2   ($\uparrow$) | RL   ($\uparrow$) |
|  |  |  |  |  |  |  |  |  |  |  |  |  |  |  |  |  |  |  |  |
| LoRA | 0.05030 |  | 27.17 | 65.03 |  | 37.13 | 16.43 | 30.90 |  | 35.49 |  | 0.004951 |  | 52.86 | 73 |  | 55.31 | 31.71 | 46.47 |
|  | 0.05690 |  | **39.02** | **67.90** |  | 40.80 | 18.54 | 33.87 |  | 44.07 |  | 0.005629 |  | 52.65 | 72.81 |  | 55.12 | 31.63 | 46.64 |
|  | 0.06153 |  | 23.18 | 65.05 |  | 39.15 | 17.70 | 32.79 |  | 37.67 |  | 0.006051 |  | 52.63 | 72.94 |  | 56.36 | 33.48 | 47.91 |
|  |  |  |  |  |  |  |  |  |  |  |  |  |  |  |  |  |  |  |  |
| SDLoRA | 0.05536 |  | 31.49 | 67.18 |  | 41.11 | 18.48 | 33.84 |  | 48.58 |  | 0.005484 |  | 51.86 (!) | 72.42 (!) |  | 56.08 | 32.79 | 47.61 |
|  | 0.05540 |  | 31.43 | 67.86 |  | 41.69 | 19.17 | 34.47 |  | **50.40** |  | 0.005488 |  | 52.79 (!) | **73.07** (!) |  | **56.53** | **33.5** | **47.96** |
|  | 0.05549 |  | 33.03 | 67.80 |  | **42.18** | **19.19** | **34.95** |  | 49.60 |  | 0.005497 |  | **53.11** | 72.95 |  | 56.14 | 33.08 | 47.56 |


> New Experiment Results on Mamba-II.

For Mamba-II, however, applying SDLoRA is not straightforward because Mamba-II further constrains $A$ such that all (non-zero) entries must have the same value. Therefore, our original dimension selection approach cannot be directly applied here. We consider a naive extension of SDLoRA by selecting dimensions in the projection matrices for input mapping vector $B$ and the projection matrices for output mapping vector $C$ using their respective magnitude, and fine-tune the selected dimensions and all elements of state transition matrix $A$.

**Results**: The table below compares the performance of LoRA and SDLoRA on Mamba-II. The results demonstrate that SDLoRA consistently outperforms LoRA on Mamba-II models.


|  |  | DART (Mamba-II-130M) |  |  |  | SAMSum (Mamba-II-1.3B) |  |  |  |
|---|---|---|---|---|---|---|---|---|---|
|  |  | Params (%) | BLEU ($\uparrow$) | METEOR  ($\uparrow$) |  | Params (%) | R1   ($\uparrow$) | R2   ($\uparrow$) | RL   ($\uparrow$) |
| LoRA |  | 0.3354 | 48.09 | 68.71 |  | 0.1614 | 49.73 | 26.14 | 41.53 |
| SDLoRA |  | 0.3393 | **48.93** | **70.60** |  | 0.1767 | **50.72** | **27.21** | **42.54** |

---

> ### Author Response · Authors · 2024-11-22
>
> # Major Update 2: Use of a Larger Vision Dataset — CelebA.
> **Key Insight: SDLoRA outperforms LoRA on CelebA [3].**
>
> We have extended our experiments to include CelebA, which comprises 202,599 face images (178 × 218 pixels). This dataset is significantly larger than CIFAR-10, used in the original paper, and contains 40 classification tasks (e.g., predicting attributes like gender, hair color, and glasses). We report four metrics: (i) average accuracy and overall accuracy for (ii) easy, (iii) medium, and (iv) hard tasks. Here, overall accuracy refers to the accuracy of correctly predicting all target labels within a specific subset of tasks. Tasks are categorized as easy, medium, or hard by clustering based on average performance. To ensure computational feasibility, we reduced the resolution using [InsightFace](https://github.com/deepinsight/insightface) by cropping images to retain only the face and then resizing them to 32 × 32 pixels. This preprocessing helps maintain a manageable sequence length for efficient runtime.
>
> **Results.** We conducted experiments on Mamba-130M, and the results are summarized below. The table demonstrates that SDLoRA consistently outperforms LoRA across tasks of varying difficulty levels.
>
> |  | Params (%) | Average Acc. (All) | Overall Acc. (Easy) | Overall Acc. (Medium) | Overall Acc. (Hard) |
> |---|---|---|---|---|---|
> |  |  |  |  |  |  |
> | LoRA | .3178 | 87.79 | 58.53 | 24.19 | 4.18 |
> |  | .3600 | 88.58 | 60.10 | 26.21 | 5.19 |
> |  | .3883 | 87.67 | 58.32 | 24.01 | 4.08 |
> |  |  |  |  |  |  |
> | SDLoRA | .3492 | **88.61** | 60.50 | 26.27 | **5.40** |
> |  | .3498 | 88.40 | 59.75 | 25.69 | 5.01 |
> |  | .3509 | 88.50 | **60.52** | **26.30** | 4.96 |
>
>
> # Major Update 3: More PEFT Methods — DoRA and SDDoRA.
>
> **Key Insights**:
> * **DoRA [4] is more effective for fine-tuning linear projection layers than SSM modules, aligning with our original conclusion: applying low-rank updates to linear projection matrices is more effective than to SSM modules.**
> * **SDDoRA demonstrates superior performance compared to DoRA.**
>
> We have included evaluations of DoRA (an advanced LoRA variant) alongside SDDoRA to provide a more comprehensive analysis.
>
> *  **Benchmarking DoRA**: The results presented here align with our original conclusion, demonstrating that applying low-rank updates to linear projection matrices is more effective than applying them to SSM modules.
>
>     We evaluate the performance of DoRA on the DART dataset using Mamba-130M and on the Spider dataset using Mamba-1.4B. The results are summarized in the table below.
>
>     | Params (%) | Target Layers | Method |  | DART (Mamba-130M) |  |  | Spider (Mamba-1.4B) |
>     |---|---|---|---|---|---|---|---|
>     |  |  |  |  | BLEU (↑) | METEOR (↑) |  | Acc. (↑) |
>     |  |  |  |  |  |  |  |  |
>     | < 0.4 | SSM Modules | LoRA |  | 47.05 | 68.86 |  | 58.03 |
>     |  |  | DoRA |  | 47.07 | 68.79 |  | 55.32 |
>     |  |  |  |  |  |  |  |  |
>     | < 0.4 | Linear Layers | LoRA |  | 48.86 | 70.25 |  | 61.80 |
>     |  |  | DoRA |  | 49.93 | 70.81 |  | 61.32 |
>     |  |  |  |  |  |  |  |  |
>     | < 3.0 | Both | LoRA |  | 49.52 | 70.97 |  | 56.38 |
>     |  |  | DoRA |  | 51.36 | 70.94 |  | 55.71 |
>
>
>
>     Our findings are consistent with observations seen in LoRA: applying DoRA to linear projection matrices proves more effective than its application to SSM modules. Interestingly, applying DoRA to SSM modules not only offers limited benefits but, in some cases, even degrades performance. This is particularly evident on the Spider dataset, when comparing the configurations of applying DoRA to both linear projection matrices and SSM modules versus solely targeting linear projection matrices.
>
> * **Integrating Selective Dimension Tuning with DoRA (SDDoRA)**: Incorporating selective dimension tuning into DoRA achieves superior performance compared to using DoRA alone.
>
>     We extended our investigation to include SDDoRA and evaluated its performance against DoRA alone using the DART benchmark on the Mamba-130M model. The results, presented below, show that integrating selective dimension tuning with DoRA enhances its effectiveness.
>
>
>     |  | Params (%) | BLEU ($\uparrow$) | METEOR ($\uparrow$) |
>     |---|---|---|---|
>     |  |  |  |  |
>     | DoRA | 0.3618 | 49.86 | 70.01 |
>     |  | 0.4025 | 51.22 | 70.40 |
>     |  | 0.4040 | 50.53 | 69.94 |
>     |  |  |  |  |
>     | SDDoRA | 0.3630 | 51.32 | 70.33 |
>     |  | 0.3633 | **51.55** | **70.80** |
>     |  | 0.3639 | 50.80 | 70.50 |

---

> ### Author Response · Authors · 2024-11-22
>
> # Major Update 4: Memory Usage and Runtime Analysis of SDLoRA.
>
> **Key Insights:**
>
> 1. **Memory Usage: SDLoRA uses *less* memory compared to LoRA when the number of trainable parameters is similar.**
> 2. **Runtime: SDLoRA is slightly faster than LoRA when the number of trainable parameters is similar.**
>
> To assess the memory usage and runtime of SDLoRA and LoRA, we conducted experiments on four different models, including both SSM and hybrid architectures. Unless specified otherwise, for each model and method, dataset were generated with 2,500 batches of data samples, each batch comprising a random sequence of 1,500 tokens. The simulation was repeated four times, including dataset generation. All experiments were carried out on a single H100 GPU, and the reported metrics represent averages across the four simulations. Consistent with our previous experiments, we used the original hyperparameter settings, ensuring that SDLoRA included more trainable parameters than LoRA.
>
>
> > 1. Memory Usage Analysis.
>
> The memory usage of LoRA and SDLoRA is summarized below. Our observations indicate that SDLoRA requires less memory than LoRA. This difference can be attributed to the design of the LoRA adapters, which involve matrix multiplication of two low-rank matrices. In contrast, tuning SSM with the same number of parameters does not require any matrix multiplication, resulting in lower memory usage.
>
> | Memory Usage (GB) | Mamba-130M | Mamba-1.4B | Jamba-Tiny-319M | Jamba-Mini-52B |
> |---|---|---|---|---|
> | LoRA | 7.753 | 37.167 | 7.207 | 71.986 |
> | SDLoRA | **5.738** | **26.491** | **6.605** | **67.193** |
>
>
> > 2. Runtime Analysis.
>
> Fine-tuning with SDLoRA consists of two stages: (1) dimension selection and (2) standard training. In this study, we first compare the runtime of SDLoRA and LoRA during stage 2 (training) and then evaluate the additional runtime introduced by SDLoRA during stage 1 (dimension selection). Our results show that the dimension selection stage adds only marginal runtime overhead, and SDLoRA is more efficient than LoRA in standard training.
>
> * **Training**: When the channels and states have been selected, the training of SDLoRA is *faster* than LoRA when the same number of trainable parameters are considered.
>
>
>     The runtimes are reported in the table below. We observe that, despite having more trainable parameters, SDLoRA is faster than LoRA. We attribute this to the fact that LoRA introduces additional FLOPs due to the extra matrix multiplication operations required for each update (specifically, the multiplication of two low-rank matrices).
>
>     | Avg. Runtime (Seconds) | Mamba-130M | Mamba-1.4B | Jamba-Tiny-319M | Jamba-Mini-52B |
>     |---------------------|------------|------------|----------------|----------------|
>     | LoRA | 410.0 $\pm$  80.0 | 2060.0 $\pm$  135.0 | 352.5 $\pm$  107.5 | 3427.5 $\pm$  185.0 |
>     | SDLoRA | **330.0 $\pm$  77.5** | **1697.5 $\pm$  87.5** | **257.5 $\pm$  72.5** | **3065.0 $\pm$ 232.5** |
>
> * **Dimension Selection**: For dimension selection, our method first performs an *Initial Subset Training*, and then selects the dimensions based on the *magnitude of parameter changes* across different dimensions.
>
>     1. **Initial Subset Training**: We update the model by going through only a subset of the dataset (e.g., 3% of batches in DART experiments), which is sufficient in practice.
>     2. **Magnitude-Based Dimension Selection**: After the subset training, we select dimensions based on the magnitude of parameter changes observed.
>
>     In this experiment, we simulate a real scenario using datasets with 2,500 batches, considering a small subset containing 125 batches (5% of the full dataset). We repeat the experiments 80 times, and the reported numbers are averaged across these simulations. The following table presents the runtime analysis of the dimension selection stage in SDLoRA.
>
>
>     | Avg. Runtime (Seconds) | Mamba-130M | Mamba-1.4B | Jamba-Tiny-319M | Jamba-Mini-52B |
>     |---|---|---|---|---|
>     | Initial Subset Training | 16.250 $\pm$ 3.880 | 85.250 $\pm$ 5.130 | 15.750 $\pm$ 1.000 | 163.630 $\pm$ 10.120 |
>     | Magnitude-Based Dimension Selection | 0.280 $\pm$ 0.000 | 0.520 $\pm$ 0.120 | 0.090 $\pm$ 0.000 | 0.240 $\pm$ 0.040 |
>     | Total Time | 16.530 $\pm$ 3.880 | 85.770 $\pm$ 5.250 | 15.840 $\pm$ 1.000 | 163.870 $\pm$ 10.160 |
>     | |
>     | Proportion of Training 1 Epoch | 0.050$\times$ | 0.051$\times$ | 0.062$\times$ | 0.053$\times$ |
>     | Proportion of Training 5 Epoch | **0.010$\times$** | **0.010$\times$** | **0.012$\times$** | **0.011$\times$** |
>
>     This table demonstrates that the dimension selection stage adds only *negligible* runtime.
>
> ***
>
> *References:*
>
> [1] Jamba: A Hybrid Transformer-Mamba Language Model
> [2] Transformers are SSMs: Generalized Models and Efficient Algorithms Through Structured State Space Duality
> [3] Deep Learning Face Attributes in the Wild
> [4] DoRA: Weight-Decomposed Low-Rank Adaptation

---

### Meta-Review · Area_Chair_UZM6 · 2024-12-19

**Metareview:**

This paper presents a systematic investigation into parameter-efficient fine-tuning (PEFT) methods for Deep State Space Models (SSMs), with a particular focus on language modeling tasks. They introduce SDLoRA, a novel approach that selectively updates certain channels and states on SSM modules while applying LoRA to linear projection matrices, showing improved performance over standard LoRA.

One of the main concerns raised by the reviewers was the limited applicability of the proposed SDLoRA method beyond SSMs. While the authors have demonstrated the effectiveness of SDLoRA on hybrid models like Jamba, more comprehensive testing on a broader range of architectures would be beneficial. The comparison with existing PEFT methods was noted as insufficient by some reviewers. While the authors defended their approach, a more comprehensive comparison could have been made to better position the proposed method against existing benchmarks. The complexity of the proposed model, particularly with components like the selective dimension tuning, raised concerns about its practicality, which was not fully addressed in the rebuttal.

The reviewers expressed mixed opinions, with some leaning towards acceptance due to the novelty and potential of the work, especially in the context of SSMs, while others remained skeptical due to the perceived incremental innovation and the lack of broader architectural testing. The paper shows promise in advancing the field of parameter-efficient fine-tuning for SSMs but would benefit from further refinement, particularly in addressing the concerns about broader applicability and providing more robust comparisons with existing methods.

This paper received an average score that, while close to the acceptance threshold, is not competitive among this year's submissions. Given the balance of strengths and weaknesses, the final recommendation is to reject this submission in its current form.

**Additional Comments On Reviewer Discussion:**

During the rebuttal period, reviewers raised key concerns about the generalizability of SDLoRA beyond SSMs and the incremental nature of the innovation. Authors addressed these by demonstrating SDLoRA's effectiveness on hybrid models like Jamba and expanding their experiments to include Mamba-II. They also provided additional data on computational overhead, showing SDLoRA's efficiency in memory usage and runtime.

In weighing these points, the decision to reject the paper was primarily due to the limited architectural testing and the insufficient comparison with existing PEFT methods. While the authors made efforts to address concerns, the overall contribution was deemed not strong enough for acceptance at ICLR, necessitating further work to broaden the scope and depth of the research.

---

### Decision · Program_Chairs · 2025-01-22

Reject